# A Reduction Framework for Distributionally Robust Reinforcement Learning under Average Reward

Zachary Roch [1]  George Atia [1 2]  Yue Wang [1 2]

## Abstract

Robust reinforcement learning (RL) under the average reward criterion, which seeks to optimize long-term system performance in uncertain environments, remains a largely unexplored area. To address this challenge, we propose a reduction-based framework that transforms robust average reward optimization into the more extensively studied robust discounted reward optimization by employing a specific discount factor. Our framework provides two key advantages. **Data Efficiency:** We design a model-based reduction algorithm that achieves near-optimal sample complexity, enabling efficient identification of optimal robust policies; **Scalability:** By bypassing the inherent challenges of scaling up average reward optimization, our framework facilitates the design of scalable, convergent algorithms for robust average reward optimization leveraging function approximation. Our algorithmic design, supported by theoretical and empirical analyses, provides a concrete solution to robust average reward RL with the first data efficiency and scalability guarantees, highlighting the framework's potential to optimize long-term performance under model uncertainty in practical problems.

## 1. Introduction

Reinforcement Learning (RL) aims to optimize an agent's performance by identifying a policy that maximizes cumulative rewards based on a specified criterion while interacting with an environment. Despite its remarkable success in applications such as synthetic control problems, board games (Silver et al., 2016; Zha et al., 2021), and video games (Wei et al., 2022; Liu et al., 2022a), RL often experiences significant performance degradation in practical settings. This phenomenon, known as the Sim-to-Real gap, arises from discrepancies between the training environment and the deployment environment. In simulation-based applications like games, the training and deployment environments are typically identical and vanilla RL performs well. However, in real-world scenarios, differences such as modeling errors, perturbations, partial observability, and potential adversarial attacks introduce model mismatches between the training and deployment environments, resulting in suboptimal policies and poor performance outcomes, undermining the reliability of RL in practical applications.

To address this issue, a framework of robust RL was introduced (Bagnell et al., 2001; Nilim & El Ghaoui, 2004; Iyengar, 2005). It deviates from vanilla RL by considering a set of environment transition dynamics instead of a fixed one, and its goal is to optimize performance under the worst-case scenario across these models, which provides performance guarantees across all uncertain environments within the defined uncertainty set, making the policy more robust to model mismatches and more generalizable.

On the other hand, different reward criteria in (robust) RL can result in substantially distinct problem settings. Among these, the discounted reward criterion is the most extensively studied. While it offers elegant mathematical properties, its focus on short-term rewards can lead to suboptimal long-term performance due to the exponential decay of rewards. In practical applications such as queuing control, supply chain inventory management, and communication networks (Kober et al., 2013), however, evaluating policies based on their long-term average performance becomes crucial. This highlights the importance of optimizing the long-term average reward in environments with uncertainty, motivating our focus on robust RL under average reward in this paper.

Despite its practical importance, robust RL under average reward is generally more complex than its discounted counterpart due to its dependence on the limiting behavior of the underlying stochastic processes, and is hence understudied. Recent work (Wang et al., 2023e; Grand-Clement et al., 2023) further emphasizes its inherent challenges, including

[1]Department of Electrical and Computer Engineering [2]Department of Computer Science, University of Central Florida, Orlando, Florida. Correspondence to: Zachary Roch <zachary.roch@ucf.edu>, George Atia <george.atia@ucf.edu>, Yue Wang <yue.wang@ucf.edu>.

*Proceedings of the 42nd International Conference on Machine Learning*, Vancouver, Canada. PMLR 267, 2025. Copyright 2025 by the author(s).

the non-contracted nature of the Bellman operator, the high dimensionality of the solution space, and the difficulties in relaxing underlying assumptions.

To address these challenges, a natural approach is to draw on insights from the extensive studies of robust RL for discounted rewards as an intermediate step. This idea has been validated in (Wang et al., 2023d; Grand-Clément et al., 2023), which show that, under certain assumptions, the performance of discounted robust RL asymptotically converges to that of average reward as the discount factor approaches 1. While this convergence highlights the potential of using discounted robust RL to study the average reward setting, there remains a lack of results demonstrating its practical applicability. Key questions regarding its efficiency, effectiveness, and scalability remain unanswered, leaving gaps in understanding its implementation and real-world impact.

In this paper, we explore this approach in greater depth and propose a reduction-based framework for concrete algorithm implementation. This framework facilitates the use of various robust discounted RL algorithms to address robust average reward RL problems. To assess the practicality of our framework, we evaluate its performance across two key dimensions: data efficiency and scalability. Our contributions are summarized as follows.

**Reduction of robust average reward RL to discounted one:** Under a standard assumption (Assumption 3.1), we propose a reduction-based framework that shows how robust average reward optimization under model uncertainty can be equivalently addressed through robust discounted RL with a specific discount factor. While prior work has explored asymptotic convergence, no practical guidance has been offered on selecting a reduction discount factor to guarantee the optimality of the resulting policy. Our framework provides a concrete choice of the reduction discount factor, ensuring that the robust policy learned for the discounted reward is also optimal under the average reward criterion. This universal framework deepens the understanding of the fundamental connections between average and discounted rewards while enabling robust average reward optimization.

**Design of data-efficient reduction algorithms:** Building on our reduction framework, we present the first model-based algorithm for robust RL under average reward, applicable to various uncertainty set models, with a thorough sample complexity analysis. We study the total number of samples required to learn an $\epsilon$-optimal robust policy for the average reward criterion under different uncertainty set structures. Specifically, we provide detailed analyses for total variation, $\chi^2$ divergence, and Kullback–Leibler divergence uncertainty sets, demonstrating that our reduction algorithms achieve near-optimal sample complexity. These results highlight the practical potential of our framework in data-intensive settings, offering the first finite-sample com-

plexity characterization of robust RL with average reward.

**Design of scalable reduction algorithms:** To further validate the practical applicability of our framework, we adapt it to design scalable algorithms for robust average reward RL. After identifying key challenges in scaling up robust average reward RL, we show that our reduction framework circumvents these difficulties, enabling the design of efficient algorithms for large-scale problems. We evaluate our algorithms in large-scale MuJoCo environments, showcasing the capability of our framework to optimize long-term rewards under model uncertainty in complex systems. These results underscore the potential of our framework to efficiently solve large-scale, real-world problems.

## 2. Preliminaries and Problem Formulation

**Discounted reward MDPs.** A discounted reward Markovian decision process (DMDP) $(\mathcal{S}, \mathcal{A}, \mathsf{P}, r, \gamma)$ is specified by: a state space $\mathcal{S}$, an action space $\mathcal{A}$, a transition kernel $\mathsf{P} = \{\mathsf{P}_s^a \in \Delta(\mathcal{S}), a \in \mathcal{A}, s \in \mathcal{S}\}$[1], where $\mathsf{P}_s^a$ is the distribution of the next state over $\mathcal{S}$ upon taking action $a$ in state $s$ (with $\mathsf{P}_{s,s'}^a$ denoting the probability of transitioning to $s'$), a reward function $r : \mathcal{S} \times \mathcal{A} \to [0, 1]$, and a discount factor $\gamma \in [0, 1)$. At each time step $t$, the agent at state $s_t$ takes an action $a_t$, the environment then transitions to the next state $s_{t+1}$ according to $\mathsf{P}_{s_t}^{a_t}$, and produces a reward signal $r_t = r(s_t, a_t)$ to the agent.

A stationary policy $\pi : \mathcal{S} \to \Delta(\mathcal{A})$ is a distribution over $\mathcal{A}$ for any given state $s$. The agent follows the policy by taking action subject to the distribution $\pi(s)$. The accumulative reward of a stationary policy $\pi$ starting from $s \in \mathcal{S}$ for DMDPs is measured by the discounted value function: $V_{\gamma, \mathsf{P}}^\pi(s) \triangleq \mathbb{E}_{\pi, \mathsf{P}}\left[\sum_{t=0}^\infty \gamma^t r_t | S_0 = s\right]$.

**Average reward MDPs.** Unlike DMDPs, average reward MDPs (AMDPs) do not discount the rewards over time and instead measure the accumulative reward by considering the behavior of the underlying Markov process under the steady-state distribution. Specifically, the average reward (or the gain) of a policy $\pi$ starting from $s \in \mathcal{S}$ is[2]

$$g_\mathsf{P}^\pi(s) \triangleq \lim_{n \to \infty} \mathbb{E}_{\pi, \mathsf{P}}\left[\frac{1}{n} \sum_{t=0}^{n-1} r_t | S_0 = s\right]. \quad (1)$$

It is also useful to define the following relative value function or bias for an AMDP:

$$h_\mathsf{P}^\pi(s) \triangleq \mathbb{E}_{\pi, \mathsf{P}}\left[\sum_{t=0}^\infty (r_t - g_\mathsf{P}^\pi) | S_0 = s\right], \quad (2)$$

which is the cumulative difference over time between the immediate reward and the average reward.

---

[1] $\Delta(\mathcal{S})$: the $(|\mathcal{S}| - 1)$-dimensional probability simplex on $\mathcal{S}$.

[2] The limit may not exist, but under the assumption we made, such a limit exists (Puterman, 1994).

**Robust MDPs.** In robust MDPs, the transition kernel is not fixed but, instead, belongs to a designated uncertainty set denoted as $\mathcal{P}$. Following an action, the environment undergoes a transition to the next state based on an arbitrary transition kernel $\mathsf{P} \in \mathcal{P}$. We specifically concentrate on the $(s, a)$-rectangular uncertainty set (Nilim & El Ghaoui, 2004; Iyengar, 2005), where $\mathcal{P} = \bigotimes_{s,a} \mathcal{P}_s^a$, with $\mathcal{P}_s^a \subseteq \Delta(\mathcal{S})$ defined independently over all state-action pairs.

Robust MDPs aim to optimize the worst-case performance over the uncertainty set. The robust discounted value function of a policy $\pi$ is defined as the worst-case discounted value function over all possible transition kernels:

$$V_{\gamma, \mathcal{P}}^\pi(s) \triangleq \min_{\kappa \in \bigotimes_{t \geq 0} \mathcal{P}} \mathbb{E}_{\pi, \kappa} \left[ \sum_{t=0}^\infty \gamma^t r_t | S_0 = s \right], \quad (3)$$

where $\kappa = (\mathsf{P}_0, \mathsf{P}_1...) \in \bigotimes_{t \geq 0} \mathcal{P}$. The discounted robust value functions are shown to be the unique solution to the robust discounted Bellman equation (Iyengar, 2005):

$$V(s) = \sum_a \pi(a|s)(r(s, a) + \gamma \sigma_{\mathcal{P}_s^a}(V)), \quad (4)$$

where $\sigma_{\mathcal{P}_s^a}(V) \triangleq \min_{\mathsf{P} \in \mathcal{P}_s^a} \mathsf{P} V$ is the support function of $V$ on the uncertainty set $\mathcal{P}_s^a$.

In scenarios where the long-term performance under model uncertainty is concerned, we focus on the following worst-case average reward:

$$g_{\mathcal{P}}^\pi(s) \triangleq \min_{\kappa \in \bigotimes_{t \geq 0} \mathcal{P}} \lim_{n \to \infty} \mathbb{E}_{\pi, \kappa} \left[ \frac{1}{n} \sum_{t=0}^{n-1} r_t | S_0 = s \right], \quad (5)$$

to which we refer as the robust average reward. The robust AMDP aims to find an optimal policy w.r.t. it, that is, $\pi^* \triangleq \arg \max_{\pi \in \Pi} g_{\mathcal{P}}^\pi(s)$, for any $s \in \mathcal{S}$.

In (Wang et al., 2023d), it is shown that the robust discounted value functions converge to the robust average reward w.r.t. the same MDP as the discount factor approaches 1:

$$\lim_{\gamma \to 1} (1 - \gamma) V_{\gamma, \mathcal{P}}^\pi = g_{\mathcal{P}}^\pi. \quad (6)$$

Hence, the robust AMDP can be approximately solved through the corresponding robust DMDP with a sufficiently large discount factor, known as the reduction method. However, selecting a discount factor to ensure near-optimality under the average reward remains unclear, leaving the adaptation of the reduction method uncertain.

In this paper, our goal is to develop a concrete reduction framework and design algorithms for optimizing the robust average reward, and to demonstrate the practical applicability of our framework.

## 3. Reduction Framework for Robust AMDPs

Our framework aims to reduce the robust average reward problem to a robust discounted reward one, leveraging well-developed algorithms in this space. The convergence (6) and the existence of a robust Blackwell optimal policy (Wang et al., 2023d; Grand-Clément & Petrik, 2024) (a policy that optimizes the robust discounted reward for any $\gamma > \gamma_{bw}$) further inspires us to reduce a robust AMDP to a robust DMDP with a sufficiently large discount factor. However, existing results focus on asymptotic convergence, leaving the choice of discount factor for a desired level of accuracy unresolved. In this section, we study the relationship between the two robust MDPs and determine a specific reduction discount factor.

We first adopt the following compactness and unichain assumption, which is commonly used in robust AMDPs.

**Assumption 3.1.** For any $s \in \mathcal{S}, a \in \mathcal{A}$, the uncertainty set $\mathcal{P}_s^a$ is a compact subset of $\Delta(\mathcal{S})$. Moreover, any deterministic policy $\pi$ and any kernel $\mathsf{P} \in \mathcal{P}$ induce a unichain Markovian process[3].

Due to the Heine–Borel theorem (Dugac, 1989), the first part of Assumption 3.1 is satisfied if the uncertainty set is closed as it is always bounded. We remark that many standard uncertainty sets satisfy this assumption, e.g., those defined by $\epsilon$-contamination (Huber, 1965), finite interval (Tewari & Bartlett, 2007), total-variation (Rahimian et al., 2022) and KL-divergence (Hu & Hong, 2013).

The second part of Assumption 3.1 imposes additional structure on the underlying MDP, an assumption commonly used in non-robust AMDP studies due to their inherent complexity (e.g., (Puterman, 1994; Wan et al., 2021; Zhang & Ross, 2021; Lan, 2020; Zhang et al., 2021b)). For robust AMDPs, the unichain assumption ensures the solvability of the average reward robust Bellman equation (Wang et al., 2023d;e), which plays an essential role in our analysis. Under this assumption, the stationary distribution $\eta_{\mathsf{P}}^\pi$ always exists and does not depend on the initial state, and the average reward is identical for all starting states (Bertsekas, 2011), i.e., $g_{\mathsf{P}}^\pi(s_1) = g_{\mathsf{P}}^\pi(s_2), \forall s_1, s_2 \in \mathcal{S}$.

Inspired by non-robust AMDP studies (Wang et al., 2022; Zurek & Chen, 2023), we further extend the concept of the optimal bias span (Bartlett & Tewari, 2012) therein to the robust setting.

**Definition 3.2.** For a robust AMDP $(\mathcal{S}, \mathcal{A}, \mathcal{P}, r)$, its robust

---

[3]A Markovian process is a unichain if it contains exactly one recurrent class and possibly some transient states (Puterman, 1994; Bertsekas, 2011).

optimal bias span is defined as[4]

$$\mathcal{H} \triangleq \max_{\mathsf{P} \in \mathcal{P}} \mathbf{Sp}(h_{\mathsf{P}}^{\pi^*}) \tag{7}$$

where $h_{\mathsf{P}}^{\pi^*}$ is the relative value function as in (2), and $\mathbf{Sp}(V) \triangleq \max_i V(i) - \min_i V(i)$ is the Span semi-norm.

**Remark 3.3.** *We assume $\mathcal{H}$ is known, which can be viewed as a robust extension from the common assumption of knowledge of non-robust span in non-robust AMDPs studies (Wang et al., 2022; Zurek & Chen, 2023; Zhang & Xie, 2023; Wang et al., 2023a). Our framework and results remain valid if $\mathcal{H}$ is replaced with any upper bound on $\mathcal{H}$, such as the corresponding robust extensions of the mixing time or the diameter of a non-robust MDP (Wang et al., 2022). Additional discussion on estimating $\mathcal{H}$ is provided in Section 10. Specifically, we can derive an upper bound on $\mathcal{H}$ for robust MDPs with some additional structures.*

Next, we present our reduction framework.

**Theorem 3.4.** *(Reduction Framework) For any $\epsilon$, set $\gamma := 1 - \frac{\epsilon}{\mathcal{H}}$, then any $\epsilon_\gamma$-optimal policy[5] $\hat{\pi}_\gamma$ for the robust DMDP $(\mathcal{S}, \mathcal{A}, \mathcal{P}, r, \gamma)$ is also an $\mathcal{O}(\epsilon)$-optimal policy for the corresponding robust AMDP $(\mathcal{S}, \mathcal{A}, \mathcal{P}, r)$:*

$$g_{\mathcal{P}}^{\pi^*} - g_{\mathcal{P}}^{\hat{\pi}_\gamma} \leq \left(8 + \frac{5\epsilon_\gamma}{\mathcal{H}}\right)\epsilon.$$

While we defer the full proof to the appendix, the intuition behind the proof of our reduction framework relies on showing a bound on the convergence error of the robust discounted value function to the average reward. As shown in Lemma 11.1, under Assumption 3.1 with $\gamma \in (0,1)$ and for any stationary $\pi \in \Pi$, $\|g_{\mathcal{P}}^\pi - (1-\gamma)V_{\gamma,\mathcal{P}}^\pi\|_\infty \leq \mathbf{Sp}((1-\gamma)V_{\gamma,\mathcal{P}}^\pi)$. We then show that we can bound the $\epsilon_\gamma$-optimal robust discounted value function induced by $\pi_\gamma$ as $\mathbf{Sp}((1-\gamma)V_{\gamma,\mathcal{P}}^{\pi_\gamma}) \leq \epsilon$. Similarly, we show we can bound $\mathbf{Sp}((1-\gamma)V_{\gamma,\mathcal{P}}^{\pi^*}) \leq \epsilon$, before we combine these two results with that of Lemma 11.1 to derive our bound in the Theorem 3.4.

The result in Theorem 3.4 shows that we provide a concrete choice for the reduction discount factor, ensuring that the robust DMDP and robust AMDP share the same near-optimal policy. Our framework allows any algorithm designed for robust DMDPs to be directly applied to solve robust AMDPs, with theoretical performance guarantees, effectively bypassing the challenges of robust AMDPs by transforming them into the well-studied DMDP domain.

In the following sections, we investigate the applicability of our reduction framework from two key perspectives: data efficiency and scalability. These demonstrate the framework's

ability to optimize long-term performance in data-intensive and large-scale practical scenarios.

# 4. Sample Complexity for Robust RL under Average Reward

In this section, we study the data efficiency of our reduction framework for robust RL under average reward, to characterize the total number of samples required to identify an $\epsilon$-optimal policy $\pi$ through our reduction framework.

We first present a general result on the sample complexity of robust AMDP reduction. Specifically, leveraging Theorem 3.4, the sample complexity of robust AMDP algorithms aligns with that of robust DMDP algorithms with a specific discount factor. This result is formally stated as follows:

**Theorem 4.1.** *Consider any algorithm $\mathcal{Y}$ optimizing the robust discounted reward. Denote the sample complexity of $\mathcal{Y}$ to identify an $\epsilon_\gamma$-optimal policy (w.r.t. the discounted reward) by $\mathcal{N}(\mathcal{S}, \mathcal{A}, \mathcal{P}, \gamma, \epsilon_\gamma)$. Then, we can identify an $\epsilon$-optimal policy for the robust average reward through reduction and algorithm $\mathcal{Y}$, with a sample complexity of*

$$\mathcal{N}\left(\mathcal{S}, \mathcal{A}, \mathcal{P}, 1 - \frac{\epsilon}{\mathcal{H}}, \mathcal{H}\right). \tag{8}$$

This result holds universally, regardless of the uncertainty set models or algorithms used for robust DMDPs. More importantly, it provides a basis for studying sample complexity and data requirements for optimizing the average reward under model uncertainty. In the following, we analyze the sample complexity of robust AMDPs under different uncertainty sets, offering a concrete understanding of their data efficiency and our framework.

**Remark 4.2.** *Although Theorem 4.1 holds for general uncertainty sets, existing sample complexity studies of robust RL focus on the 'ball-structured' uncertainty sets:*

$$\mathcal{P}_s^a = \{q \in \Delta(\mathcal{S}) : D(q\|\mathsf{P}_s^a) \leq R\}, \tag{9}$$

*where $\mathsf{P}_s^a$ is the nominal kernel, $R$ is the uncertainty radius indicating the uncertainty level, and $D$ is some distribution distance measure or divergence function. Hence, we similarly focus on uncertainty sets with this structure.*

In our subsequent analysis, we focus on the generative model setting, where the agent can arbitrarily generate samples following the nominal kernel $\mathsf{P}$. This setting has been widely adopted for sample complexity analysis under both non-robust RL (Agarwal et al., 2020; Li et al., 2020; Zurek & Chen, 2023) and robust RL (Shi et al., 2023; Panaganti & Kalathil, 2022; Wang et al., 2023e). While robust RL has also been explored in other settings, such as offline (Shi & Chi, 2022; Blanchet et al., 2024; Wang et al., 2024b; Panaganti et al., 2022; Liu & Xu, 2024; Wang et al., 2024a) and

---

[4]We prove in the Appendix that the value $\mathcal{H}$ exists and is finite.

[5]For a robust DMDP, an $\epsilon_\gamma$-optimal policy is some policy $\pi_\gamma$ such that $\max_\pi V_{\gamma,\mathcal{P}}^\pi(s) - V_{\gamma,\mathcal{P}}^{\pi_\gamma}(s) \leq \epsilon_\gamma, \forall s \in \mathcal{S}$.

online (Lu et al., 2024) scenarios, and sample complexity results for robust average reward under these settings can be directly derived from Theorem 4.1, we concentrate on the generative setting, allowing us to focus on the challenges of the robust average reward framework itself rather than the complexities of data collection under restricted settings.

We develop a model-based reduction meta-algorithm for robust AMDPs. Specifically, after generating the data, we construct an estimate of the nominal kernel, and build an empirical uncertainty set centered around it with the same $D$ and $R$ in (9). We then solve a DMDP with a specific discount factor to identify a near-optimal policy. Our algorithm is presented in Algorithm 1.

---

**Algorithm 1** Model-based algorithm for robust AMDPs

---

1: **Input:** $N$ nominal samples $\{(s, a, s'_i)_{i=1}^N, s'_i \sim \mathsf{P}_s^a\}$ under each $(s, a)$ pair, uncertainty level $R$, robust bias span $\mathcal{H}$, and accuracy level $\epsilon$
2: **Initialization:** $Q \leftarrow 0$
3: Estimate transition model $\hat{\mathsf{P}}_{s,s'}^a = \frac{\sum_i \mathbf{1}_{(s,a,s'_i)=(s,a,s')}}{N}$
4: Construct empirical uncertainty set $\hat{\mathcal{P}}$ centered at $\hat{\mathsf{P}}$: $\hat{\mathcal{P}}_s^a = \{q \in \Delta(\mathcal{S}) : D(q||\hat{\mathsf{P}}_s^a) \leq R\}$
5: Set $\gamma \leftarrow 1 - \frac{\epsilon}{\mathcal{H}}, \epsilon_\gamma \leftarrow \mathcal{H}$
6: Obtain an $\epsilon_\gamma$-optimal policy $\hat{\pi}_\gamma$ for the robust DMDP $(\mathcal{S}, \mathcal{A}, r, \hat{\mathcal{P}}, \gamma)$ with value iteration
7: **Output:** $\hat{\pi}_\gamma$

---

**Remark 4.3.** *In Line 6 of Algorithm 1, we need to identify an $\epsilon_\gamma$-optimal policy for robust DMDPs, which can be done through robust value/policy iteration (Panaganti & Kalathil, 2022; Yang et al., 2021; Shi et al., 2023). For commonly used uncertainty sets, e.g., when $D$ is total variation or $\chi^2$ divergence, the algorithms can be implemented with polynomial computational complexity (Iyengar, 2005) and exponentially fast convergence rate.*

Next, we present the sample complexity of Algorithm 1 for the robust AMDP in the following theorem.

**Theorem 4.4.** *Consider an uncertainty set defined by total variation (TV) or $\chi^2$ divergence (CS). Let $C$ be some universal constant. If the total number of samples satisfies*

$$NSA \geq \frac{CSA \min\{\frac{1}{R}, \mathcal{H}^{1+\mathbf{I}_{R<\frac{1}{\mathcal{H}}}}\} \log\left(\frac{SAN}{\delta}\right)}{\epsilon^2}, (TV)$$

$$NSA \geq \frac{CSA\mathcal{H}^2(1+R) \log\left(\frac{SAN}{\delta}\right)}{\epsilon^2}, (CS)$$

*then, with probability at least $1 - 4\delta$, $\hat{\pi}_\gamma$ is $\epsilon$-optimal under the robust average reward.*

The result shows that Algorithm 1 requires at most $\tilde{\mathcal{O}}\left(\frac{SA\mathcal{H}^2}{\epsilon^2}\right)$ samples to identify an $\epsilon$-optimal policy for both

robust AMDPs. We note that the minimax lower bound on the sample complexity for non-robust AMDPs is $\tilde{\Omega}\left(\frac{SAH}{\epsilon^2}\right)$, with $H$ being the non-robust optimal span (Zurek & Chen, 2023; Wang et al., 2022; Jin & Sidford, 2021). Thus, our results are near-optimal under these uncertainty sets, aligning with the lower bound in terms of $S, A, \epsilon$, with an additional dependence on $\mathcal{H}$. Notably, this is the first sample complexity analysis for robust RL under average reward, offering insights into data requirements for long-term reward optimization under model uncertainty. Furthermore, our framework demonstrates strong data efficiency, requiring nearly minimal samples to optimize the long-term reward, underscoring its potential in data-intensive scenarios.

**Remark 4.5.** *Theorem 4.4 **is not** a direct combination of Theorem 4.1 with existing sample complexity of robust DMDPs (Panaganti & Kalathil, 2022; Shi et al., 2023). Specifically, the existing sample complexity of a robust DMDP with TV set is $\tilde{\mathcal{O}}\left(\frac{SA}{(1-\gamma)^2 R\epsilon_\gamma^2}\right)$, and $\tilde{\mathcal{O}}\left(\frac{SA}{(1-\gamma)^4 \epsilon_\gamma^2}\right)$ for CS set. By setting $\epsilon_\gamma = \mathcal{H}$ and $\gamma = 1 - \frac{\epsilon}{\mathcal{H}}$, the resulting average reward complexity is $\tilde{\mathcal{O}}\left(\frac{SA}{\epsilon^2 R}\right)$ and $\tilde{\mathcal{O}}\left(\frac{SA\mathcal{H}^2}{\epsilon^4}\right)$, respectively. Such higher complexity results are due to the higher dependence on $(1-\gamma)$ in the DMDP complexity, which becomes $\epsilon$-order in the average reward setting through our framework. To achieve tighter results, we need to further tighten the complexity result for robust DMDPs. Specifically, we showed that with a reward perturbation technique (Li et al., 2020; Wang et al., 2022; Zurek & Chen, 2023) and more careful analysis involving the connection between robust DMDPs and AMDPs, we can reduce both sample complexities to $\tilde{\mathcal{O}}\left(\frac{SA\mathcal{H}^2}{(1-\gamma)^2 \epsilon_\gamma^2}\right)$, which further result in the near-optimal complexity in Theorem 4.4.*

We can further obtain sample complexity for optimizing robust AMDPs for other types of uncertainty sets, by combining Theorem 4.1 with existing results for DMDPs. For instance, combining with (Panaganti & Kalathil, 2022), Algorithm 1 requires $\mathcal{O}\left(\frac{S^2 A\mathcal{H}^2 \log\left(\frac{SAN}{\delta}\right)}{\epsilon^4} \exp\left(\frac{\mathcal{H}}{\epsilon}\right)\right)$ samples to identify an $\epsilon$-optimal policy under the Kullback-Leibler (KL) divergence. The results in (Clavier et al., 2023) imply a sample complexity of $\mathcal{O}\left(\frac{SA\mathcal{H}^2 \log\left(\frac{SAN}{\delta}\right)}{\epsilon^4}\right)$ for the $l_p$-normed uncertainty set. Our framework thus provides the first concrete sample complexity characterization for robust AMDPs, establishing a foundation for studying robust long-term performance optimization.

**Remark 4.6.** *As demonstrated in prior studies on non-robust AMDPs (Zurek & Chen, 2023; Wang et al., 2022; 2023b), tightening the complexity bounds for non-robust DMDPs leads to optimal AMDP complexity in the corresponding reduction framework. We therefore attribute our sub-optimal sample complexity to the loose bounds of robust DMDPs rather than limitations in our reduction framework.*

*Refining the complexity analysis of robust DMDPs to align with the optimal bounds for robust AMDPs is an important direction for future research.*

## 5. Scalable Robust RL for Average Reward

In this section, we explore scalable approaches for optimizing the average reward under model uncertainty, aiming to facilitate long-term performance optimization in practical, large-scale problems. Specifically, we show that our framework overcomes the major challenges in solving large scale robust AMDPs, and further enables us to design scalable algorithms for the robust average reward, greatly enhancing the scalability and applicability of our methods.

When the problem scales are large, function approximation (FA) techniques have been extensively studied. FA methods aim to approximate the value functions by some low-dimensional function class, $\mathcal{F} = \{f_\theta(s) : \theta \in \Theta \subseteq \mathbb{R}^d, d \ll S\}$, to find some $\theta^* \in \Theta$ such that $V(s) \approx f_{\theta^*}(s), \forall s$. Two commonly studied function classes for FA are the linear function class and neural networks (Cai et al., 2019; Bhatnagar et al., 2009; Wai et al., 2019). We focus on linear FA to illustrate the scalability of our framework, but our method can be directly extended to neural networks or other function classes.

Linear FA is based on a set of feature functions $\{\phi : \mathcal{S} \to \mathbb{R}^d\}$. In robust DMDPs, the robust value function is approximated using a linear function: $V_\theta(s) \triangleq \phi(s)^\top \theta \approx V_{\gamma,\mathcal{P}}^*(s)$, where $\theta \in \mathbb{R}^d$ is some weight vector. Despite extensive studies on linear FA in robust DMDPs (Tamar et al., 2014; Xu & Mannor, 2010; Wang & Zou, 2021; Zhou et al., 2024; Roy et al., 2017; Badrinath & Kalathil, 2021), it remains largely understudied for (robust) AMDPs. In the non-robust average reward policy evaluation problem, we aim to approximate the bias $h_{\mathsf{P}}^\pi$ to estimate $g_{\mathsf{P}}^\pi$, by approximating the solution to the non-robust Bellman equation

$$h(s) = \sum_a \pi(a|s)(r(s,a) - g + \mathsf{P}_s^a h). \quad (10)$$

However, two major challenges hinder the study of linear FA for average reward MDPs. The first is that (10) admits **non-unique solutions** (Puterman, 1994; Wan et al., 2021; Wan & Sutton, 2022). Specifically, besides the average reward and relative value function pair $(g_{\mathsf{P}}^\pi, h_{\mathsf{P}}^\pi)$, any pair[6] $(g_{\mathsf{P}}^\pi, h_{\mathsf{P}}^\pi + ce)$ with any $c \in \mathbb{R}$ is also a solution to (10). This implies that the weight vector may not be unique, leading to a divergent algorithm. A common approach to address this issue is to impose additional assumptions on the feature functions (Zhang et al., 2021a; Tsitsiklis & Van Roy, 1999; Konda & Tsitsiklis, 1999; Yu & Bertsekas, 2009), ensuring that the all-one vector $e$ does not lie within the span of $\{\phi(s)\}$,

i.e., $e \neq \Phi\theta, \forall\theta$. Under this assumption, there exists a unique $\theta$ that minimizes the approximation loss. We note that the robust average reward also encounters this issue, but the higher dimensionality of its solution space (compared to the one-dimensional solution space in the non-robust setting) (Wang et al., 2024c) may result in more restrictive assumptions, making it even more challenging to directly apply linear FA to the robust average reward setting.

Another challenge arises from **unstable convergence** under the average reward. Even in the tabular setting, algorithm design and convergence analysis remain limited due to the instability caused by the Span semi-norm multi-step contraction of the Bellman operator (Puterman, 1994), compared to the norm contraction in the discounted setting. Existing studies address this issue either by introducing additional offset functions to stabilize convergence (Wan et al., 2021; Wan & Sutton, 2022; Puterman, 1994; Bertsekas, 2011) or by only ensuring convergence to a solution set (Zhang et al., 2021b). As demonstrated in (Wang et al., 2023d;e), similar challenges persist in the robust average reward setting, and addressing them is expected to involve even greater complexity.

Noting these two issues, it can be much more challenging to directly apply FA techniques to the robust average reward. However, our reduction framework simplifies this by transforming the complex robust average reward problem into the more manageable discounted reward setting. This bypasses the aforementioned difficulties and facilitates the design of scalable algorithms. As an immediate application, we extend our reduction framework to the robust natural actor-critic (NAC) algorithm with linear FA (Zhou et al., 2024), resulting in a robust NAC algorithm for robust AMDPs, detailed in Algorithm 2. Algorithm 2 consists of two key steps: (i) a robust TD step to update the weight vector $\theta$ for value approximation, and (ii) an actor step to update the policy[7], neither of which suffers from the issues mentioned above.

Note that the feature vectors in Algorithm 2 are predefined, e.g., by tile coding (Sutton, 1995), Fourier Basis (Konidaris et al., 2011), or randomly generated (Ghavamzadeh et al., 2010), by the learner prior to training.

We further characterize the convergence of Algorithm 2 (see Theorem 14.2 for the formal theorem statement).

**Theorem 5.1.** *(Informal) Under some additional standard assumptions, set $T = \tilde{\mathcal{O}}(\frac{\mathcal{H}}{\epsilon})$ in Algorithm 2, then $\pi_{w_T}$ is an $\mathcal{O}(\epsilon + \epsilon_c)$-optimal policy, where $\epsilon_c$ is the critic error due to the representation power of the linear function class. The total sample complexity is $\tilde{\mathcal{O}}(\mathcal{H}^2\epsilon^{-3})$.*

Our results demonstrate that, without requiring any addi-

---

[6]$e = (1, 1, ..., 1)$ is the all-one vector.

[7]The detailed algorithms are provided in Section 14.

**Algorithm 2** Reduction Robust Natural Actor-Critic

1: **Input:** $T, \mathcal{H}$, base functions $\{\phi\}$
2: **Initialization:** $\theta_0$ for value function approximation and $w_0$ for policy parametrization
3: $\gamma \leftarrow 1 - \frac{\epsilon}{\mathcal{H}}$
4: **for** $t = 0, 1, \ldots, T - 1$ **do**
5:     Robust critic updates $\theta_t$ with Algorithm 3
6:     Robust natural actor updates $w_{t+1}$ with Algorithm 4
7: **end for**
8: **Output:** $w_T$

tional assumptions on the base functions, our framework enables the optimization of robust average reward with function approximation with a stable convergent algorithm, underscoring its scalability. This represents the first solution for large-scale robust long-term reward optimization, offering both convergence and performance guarantees.

**Remark 5.2.** *In addition to function approximation, designing model-free algorithms for robust AMDPs can also improve scalability. Our framework also facilitates model-free algorithm design, which we discuss in detail in Section 15.*

## 6. Numerical Experiments

We first assess the effectiveness and efficiency of our framework in tabular settings, aiming to verify that with the reduction discount factor in Theorem 3.4, optimizing the robust DMDP approximately optimizes the robust AMDP.

Our experiments are conducted on the Garnet problem (Archibald et al., 1995), where the nominal transition kernel and reward functions are randomly generated from Gaussian distributions. We consider a Garnet problem with 20 states and 8 actions, constructing uncertainty sets using Total Variation (TV) and Chi-Square (CS) divergences, both with radius $R = 0.2$. Additional results for other problems are provided in Section 9.

To verify the effectiveness of our framework, we optimize robust DMDPs with different discount factors and plot the robust average reward $g_{\mathcal{P}}^{\pi_\gamma}$ of the learned policy $\pi_\gamma$ using Algorithm 1 from (Wang et al., 2023d). As a baseline, we plot the optimal robust average reward $g_{\mathcal{P}}^*$, obtained via Algorithm 2 from (Wang et al., 2023d). Additionally, we estimate $\mathcal{H}$ using Algorithm 2 from (Wang et al., 2023e) and compute the corresponding reduction factor for different accuracy levels $\epsilon$, as described in Theorem 3.4. The results, shown in Figure 1, indicate that the robust average reward can be approximately optimized through a robust DMDP with a sufficiently large discount factor. Furthermore, the robust average reward corresponding to each $\epsilon$ falls within the prescribed accuracy level, confirming the effectiveness of our framework and the validity of the reduction factor.

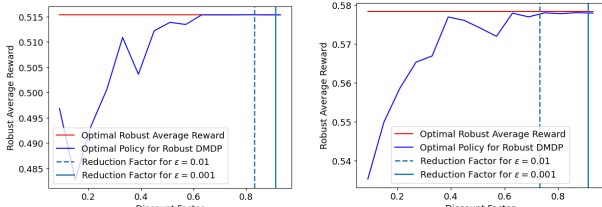

*Figure 1.* Effectiveness under the Garnet problem.

Next, we evaluate the data efficiency of our model-based framework under the same settings by optimizing empirical robust DMDPs with a reduction factor corresponding to $\epsilon = 0.01$ across different dataset sizes. The results, presented in Figure 2, show that our model-based algorithm converges to the optimal policy as dataset size $N$ increases and achieves near-optimal performance with a limited amount of data. These findings highlight the data efficiency of our framework, demonstrating its ability to optimize long-term rewards with fewer samples. Since the computational cost for finding the worst performance (i.e., robust policy evaluation) in TV and CS uncertainty sets are $O\big(S \log(S)\big)$ (Iyengar, 2005) under the tabular setting, their estimation can be tractable.

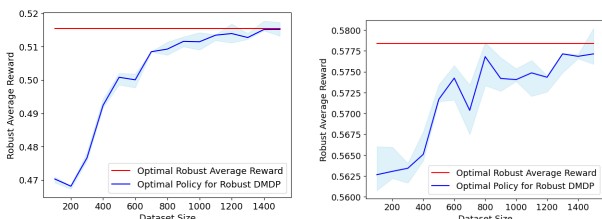

*Figure 2.* Efficiency under the Garnet problem.

We then demonstrate the scalability of our framework in large-scale problems by showing that robust average reward optimization can be equivalently formulated as a robust DMDP with a sufficiently large reduction factor. Specifically, we implement Algorithm 2 in the MuJoCo simulation environments (Todorov et al., 2012) for robust DMDPs with varying discount factors and evaluate the average reward performance under model uncertainty. Implementation details are provided in Section 14.

Our experiments are conducted in two continuous large-scale environments: Walker2d-v3 and Hopper-v3. We first train our algorithms in the nominal environments to learn the optimal policy for the given robust discounted reward. To estimate the robust average reward, we deploy the learned policies in perturbed environments, where at each evaluation epoch, we randomly sample parameters within the perturbation interval and apply them to each joint. The average reward is then computed over 30 independently perturbed environments.

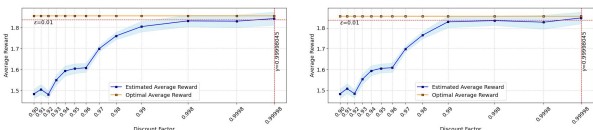

*Figure 3.* Scalability under Walker2d-v3.

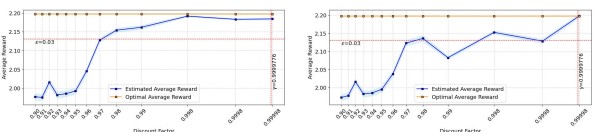

*Figure 4.* Scalability under Hopper-v3.

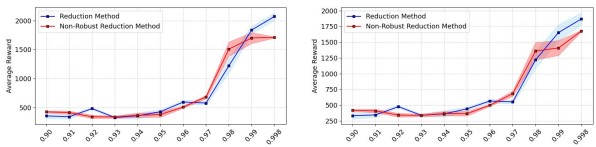

*Figure 5.* Neural network approximation under Walker2d-v3.

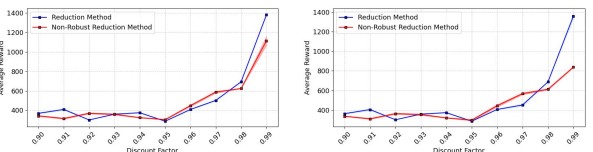

*Figure 6.* Neural network approximation under Hopper-v3.

As noted in Remark 3.3, we assume we have the knowledge of $\mathcal{H}$, however, we would like to note that in practice this is rarely the case. In lieu of a pre-obtained $\mathcal{H}$, we can equivalently use any upper-bound on $\mathcal{H}$ and see that our results still hold. In our experimentation on Walker2d-v3 and Hopper-v3, we opted to estimate the diameter as it could be easily obtained. For each of the independently perturbed environments, we sampled 1,000 randomly obtained trajectories and recorded how many steps it took the given trajectory to terminate. Based on this, we recorded the highest value for the number of steps across each of the 30 environments before averaging these values together to obtain an estimate for the diameter. The results, presented in Figures 3 and 4, show that as the discount factor increases, robust DMDP optimization aligns more closely with robust average reward optimization. In both environments, performance stabilizes when $\gamma$ is large, further validating the effectiveness of our framework. Moreover, our results demonstrate that, when combined with function approximation techniques, our framework effectively scales to large-scale problems.

We emphasize that our reduction framework is independent of any specific discounted algorithm used, thus it is **not** limited to only linear approximation. To verify this claim, we present additional experimentation using neural network approximation in Figures 5 and 6. As the results show, our reduction framework remains valid even with neural network approximation, and it is more robust than the non-robust reduction method. We present additional results for this in Section 14.

## 7. Related Work

### 7.1. Comparison with Prior Art

In this section, we compare our results with the most related work (Grand-Clément & Petrik, 2024), which also explores reduction of robust AMDPs to DMDPs. However, it focuses on bounding the Blackwell discount factor. The Blackwell discount factor, denoted as $\gamma_{bw}$, is defined such that the optimal robust policy for the DMDP with discount factor $\gamma_{bw}$ is also optimal to any robust DMDP with discount factor $\gamma \geq \gamma_{bw}$. This implies that any optimal policy for the robust $\gamma_{bw}$-DMDP also optimizes the robust AMDP.

With this notation, a robust AMDP can similarly be reduced to a robust DMDP with $\gamma_{bw}$. To enable this reduction, (Grand-Clément & Petrik, 2024) derives an upper bound on $\gamma_{bw}$. While this reduction-based framework shares some similarity with our approach, our method offers several advantages compared to their approach.

First, the results in (Grand-Clément & Petrik, 2024) are only applied to robust MDPs that meet two conditions: (1) the uncertainty sets are defined using $l_1$- or $l_\infty$-norms; and (2) the nominal kernels are rational numbers, i.e., $\mathsf{P}_s^a = \frac{n_{s,a}}{m_{s,a}}$, for some $n_{s,a}, m_{s,a} \in \mathbb{N}$. In contrast, we only require the uncertainty set to be compact and the Markov chain induced by each kernel in it to be a unichain. The restrictions in their work are due to the reliance on the separation bounds of algebraic numbers for a rational polynomial in their proofs. In contrast, we developed a more detailed structural characterization of robust AMDPs, allowing us to obtain more general results. More importantly, the resulting sample complexity from (Grand-Clément & Petrik, 2024) is less favorable than ours. The robust Blackwell discount factor is bounded by $\gamma_{bw} \leq 1 - \frac{C}{S^S m^{S^2}}$, where $m$ is the minimal denominator of the nominal kernel. Using this result in our reduction framework leads to an exponentially large sample complexity, making the results impractical. We note that the Blackwell optimality can be overly stringent for merely solving a robust AMDP, and hence it results in significantly worse sample complexity compared to ours.

## 7.2. Other Related Work

**Robust AMDPs.** Robust AMDPs studies are quite limited. Model-based robust AMDPs were first studied in (Tewari & Bartlett, 2007) for a specific finite-interval uncertainty set, which is further extended to more general models in recent works including (Wang et al., 2023d; Grand-Clement et al., 2023). A game-based method is also proposed in (Chatterjee et al., 2023). These works reveal the fundamental structure of robust AMDPs, illustrating their connections to robust DMDPs. However, all of them are model-based with asymptotic convergence, whereas we developed sample complexity analysis.

**Robust DMDPs.** Robust DMDPs were studied in (Iyengar, 2005; Nilim & El Ghaoui, 2004; Bagnell et al., 2001; Wiesemann et al., 2013; Lim et al., 2013), where the uncertainty set is assumed to be fully known. This inspired model-based methods for robust MDPs, where the learner first estimates a model, then solves the estimated model using robust dynamic programming (Zhang et al., 2021c; Panaganti & Kalathil, 2022; Yang et al., 2021; Shi et al., 2023). The studies were also extended to the model-free setting for more practical settings (Roy et al., 2017; Badrinath & Kalathil, 2021; Wang & Zou, 2021; 2022; Liu et al., 2022b; Zhou et al., 2021; Goyal & Grand-Clement, 2018; Kaufman & Schaefer, 2013; Ho et al., 2018; 2021; Wang et al., 2024c). Our work shows that the sample complexity of solving robust AMDPs can be transformed to that of solving DMDPs, enabling us to leverage the extensive prior work on the discounted setting.

**Non-robust AMDPs.** Early contributions to non-robust AMDPs involve fundamental characterizations of the problem and the development of model-based methods (Puterman, 1994; Bertsekas, 2011). Recently, model-free methods in the tabular setting, e.g., (Abounadi et al., 2001; Wan et al., 2021; Wan & Sutton, 2022), have been developed and demonstrated convergence to the optimal average reward. The sample complexity of non-robust AMDPs has been a recent focus (Wang et al., 2022; Zhang & Xie, 2023). Among them, similar reduction-based methods are considered in, e.g. (Wang et al., 2022; 2023b; Zurek & Chen, 2023), achieving the optimal complexity. Extending such frameworks to robust settings is notably challenging, due to the inherent complexity of the robust average reward setting, stemming from the non-linearity of the Bellman operator and a more complicated high-dimensional solution space for the robust Bellman equation (Wang et al., 2023e).

## 8. Conclusion

In this work, we studied the fundamental connection between robust AMDPs and DMDPs. We reveal that obtaining an optimal policy for the robust average reward is equivalent to achieving a near-optimal policy under the discounted reward with a specific reduction discount factor, based on which we constructed a reduction-based framework that solves robust RL with average reward effectively. Our framework is adaptable to any method or oracle and versatile across uncertainty sets. It offers two key benefits: data efficiency and scalability, as illustrated by our design of both tabular and function approximation algorithms, along with their sample complexity analysis and experimental performance. Our results represent the first concrete solutions for robust RL in the average reward setting under the mild assumption (unichain), advancing the understanding of optimizing long-term RL performance under model uncertainty.

## Acknowledgments

This work was supported by DARPA under Agreement No. HR0011-24-9-0427 and NSF under Award CCF-2106339. We would also like to show our appreciation to the reviewers of this work for their impactful insight during the revision of this paper.

## Impact Statement

The work contained in this paper is to advance the field of robust reinforcement learning. As there are many different potential impacts to different fields and that we provide a theory-based general framework, we do not feel that there is a need to delve into specific potential impacts here.

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

# Appendix

## 9. Additional Experiments for Tabular Settings

Building upon the results in 6, we aim to verify our theoretical findings by presenting concrete experimental validation using the true transition kernel as well as the empirical kernel in figures 7 and 8 respectively. In each setting, both the nominal transition kernel and the reward function are generated via a normal distribution for 20 states and 8 actions. Given this, we then construct the uncertainty set with either TV or CS divergence, both with a radius $R = 0.2$. For each fixed value of $\gamma$, we estimate $\pi_\gamma$ for the associated robust DMDP using robust value iteration (Iyengar, 2005). We then use Algorithm 1 in (Wang et al., 2023d) on this policy to obtain its robust average reward $g_{\mathcal{P}}^{\pi_\gamma}$. Additionally, we plot the optimal robust average reward using Algorithm 2 from (Wang et al., 2023d). By iterating through discount factors and obtaining the robust average reward, our results show that as $\gamma \to 1$, the estimated policies obtained from the corresponding robust DMDPs yield an increasingly higher robust average reward, thus we converge to the optimal policy for the robust AMDP. To further solidify this point, given arbitrarily chosen values for $\epsilon$, we can estimate the robust optimal bias span $\mathcal{H}$ using the work of (Wang et al., 2023e). With these values, we know the optimal discount factor necessary to obtain the optimal policy of the robust AMDP.

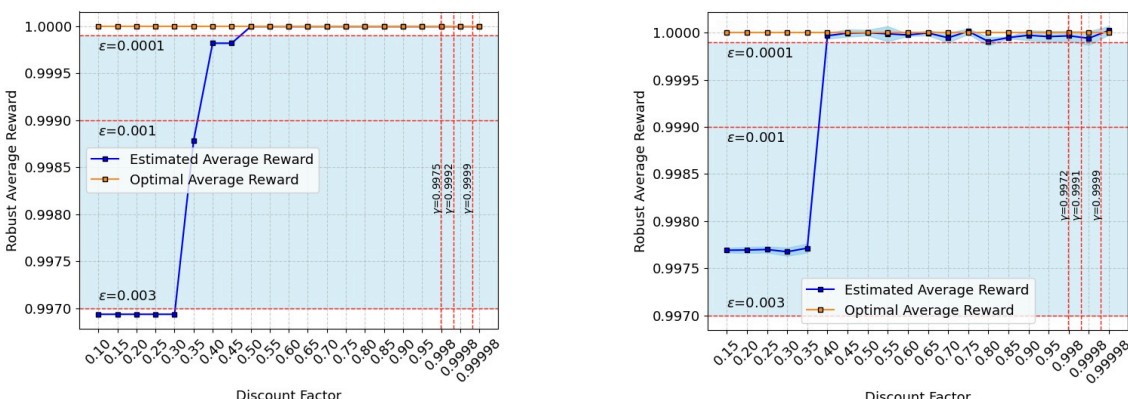

*Figure 7.* Convergence of true kernel for TV (left) and $\chi^2$-divergence (right).

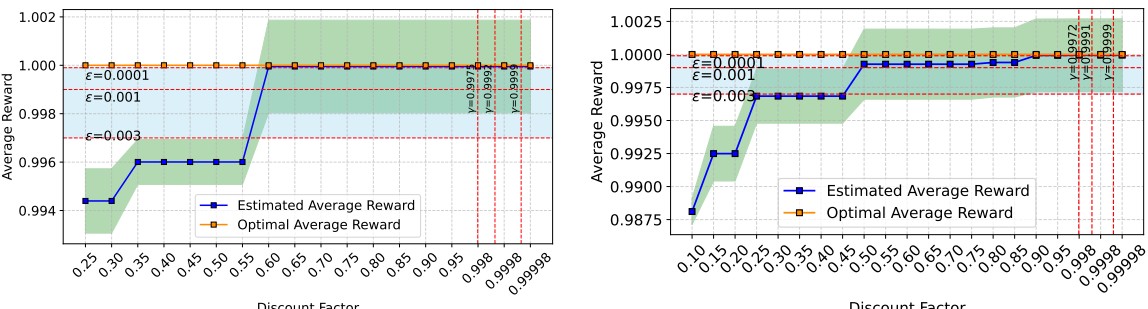

*Figure 8.* Convergence of empirical kernel for TV (left) and $\chi^2$-divergence (right).

We also wish to compare other methods to our reduction framework. Due to the novelty of our work, we opted to compare our method to two baselines for robust AMDPs, robust value iteration (RVI) (Wang et al., 2023d), and robust relative value iteration (RRVI) (Wang et al., 2023e), under the tabular Garnet problem due to the baseline method's asymptotic convergence guarantees. We set the reduction factor to be 0.99 in our framework (corresponds to $\epsilon = 0.001$. As we show in Figure 9, our method obtains a better policy within the same number of steps, thus achieving state-of-the-art performance in robust average reward optimization.

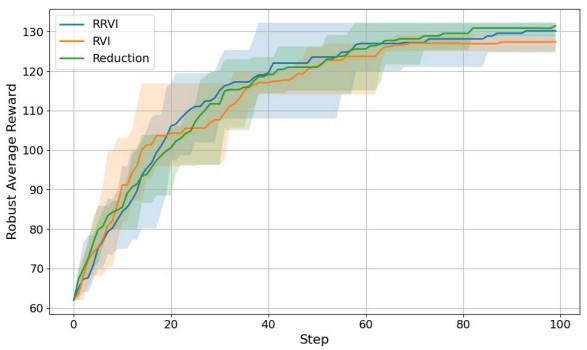

*Figure 9.* Comparison with AMDP baselines

## 10. Discussion on $\mathcal{H}$

For general uncertainty sets, we can obtain an upper bound of $\mathcal{H}$ with an additional assumption.

**Lemma 10.1.** *Assume there exists some positive integer $n > 0$ and some positive value $\rho$, such that*

$$(\mathsf{P}^n)_{s,x} \geq \rho, \forall \mathsf{P} \in \mathcal{P}, (s,x) \in \mathcal{S} \times \mathcal{S}. \tag{11}$$

*Then,*

$$\boldsymbol{Sp}(h_\mathsf{P}^\pi) \leq \frac{1}{1-\rho} \frac{1}{1-(1-\rho)^{\frac{1}{n}}}. \tag{12}$$

*Proof.* Without loss of generality, we prove the case $n = 1$. The results in other cases can be derived in a similar way. First, it holds that

$$\mathsf{P}_{x,y}^\pi \geq \rho \eta_\mathsf{P}(y), \tag{13}$$

where $\mathsf{P}_{\cdot,\cdot}^\pi$ is the reduced transition kernel by $\pi$, and $\eta_\mathsf{P}$ is the stationary distribution. Define a stochastic matrix $Q$ through the equation

$$\mathsf{P} = \rho \mathsf{P}^* + (1-\rho)Q. \tag{14}$$

We claim that

$$(\mathsf{P})^k = (1 - (1-\rho)^k)\mathsf{P}^* + (1-\rho)^k Q^k. \tag{15}$$

To show this, we use induction. Clearly (15) holds when $k = 1$. Assume that (15) holds when $k = n$:

$$(\mathsf{P})^n = (1 - (1-\rho)^n)\mathsf{P}^* + (1-\rho)^n Q^n. \tag{16}$$

Then,

$$
\begin{aligned}
(\mathsf{P})^{1+n} &= (1 - (1-\rho)^n)\mathsf{P}^*\mathsf{P} + (1-\rho)^n Q^n \mathsf{P} \\
&= (1 - (1-\rho)^n)\mathsf{P}^* + (1-\rho)^n Q^n (\rho \mathsf{P}^* + (1-\rho)Q) \\
&= (1 - (1-\rho)^{n+1})\mathsf{P}^* + (1-\rho)^{n+1} Q^{n+1},
\end{aligned}
\tag{17}
$$

which proves the claim (15). Thus, rearranging terms implies

$$\|\mathsf{P}^k - \mathsf{P}^*\| \leq (1-\rho)^k \|Q^k - \mathsf{P}^*\| \leq \frac{1}{1-\rho}(1-\rho)^k. \tag{18}$$

Then, it holds that

$$\|h_{\mathsf{P}}^{\pi}\| = \|\lim_{T\to\infty}\sum_{t=0}^{T}\mathsf{P}^t r - T\mathsf{P}^* r\| \le \lim_{T\to\infty}\|\sum_{t=0}^{T}\mathsf{P}^t r - T\mathsf{P}^* r\| \le \sum_{t=0}^{\infty}(1-\rho)^{t-1} \le \frac{1}{\rho(1-\rho)}, \tag{19}$$

which further implies that

$$\mathbf{Sp}(h_{\mathsf{P}}^{\pi}) \le 2\|h_{\mathsf{P}}^{\pi}\| \le \frac{2}{\rho(1-\rho)}. \tag{20}$$

$\square$

**Remark 10.2.** *As a sufficient condition for the assumption in Lemma 10.1, it is commonly assumed that the Markov chains induced by any deterministic policy and transition kernels in the uncertainty set are aperiodic and irreducible (Levin & Peres, 2017). Since the aperiodicity can be ensured by applying the aperiodicity transformation (Puterman, 1994), the only additional assumption required is the irreducibility of the induced Markov chains.*

**Remark 10.3.** *Another well adopted assumption that can imply an upper bound on $\mathcal{H}$ is to assume uniformly Geometric ergodic chains. Namely, there exists constants $m, \rho$, such that for all deterministic policy and kernel,*

$$\|(\mathsf{P}^{\pi})^n - \eta^{\pi}\| \le m\rho^{-n}. \tag{21}$$

*In this case, it holds that*

$$\mathcal{H} \le \max_{\mathsf{P}\in\mathcal{P}}\boldsymbol{Sp}\left(\lim_{T\to\infty}\sum_{i=1}^{T-1}((P^{\pi})^i - \eta^{\pi})r\right) \le \frac{2m}{1-\rho}. \tag{22}$$

## 11. Proof of Theorem 3.4

We first prove some important lemmas.

**Lemma 11.1.** *Under Assumption 3.1, for any $\gamma \in (0,1)$ and any stationary policy $\pi$, it holds that*

$$\|g_{\mathcal{P}}^{\pi} - (1-\gamma)V_{\gamma,\mathcal{P}}^{\pi}\|_{\infty} \le \boldsymbol{Sp}((1-\gamma)V_{\gamma,\mathcal{P}}^{\pi}).$$

*Proof.* Recalling the definition of $g_{\mathsf{P}}^{\pi}$ in (1), we have that for any kernel $\mathsf{P}$ and policy $\pi$,

$$g_{\mathsf{P}}^{\pi} = \mathsf{P}_{\pi}^* r_{\pi},$$

where $r_{\pi}$ is the induced immediate reward function: $r_{\pi}(s) = \mathbb{E}[r(s,a)|a \sim \pi(s)]$, and $\mathsf{P}_{\pi}^*$ is the Cesàro limit (Puterman, 1994) of the transition kernel $\mathsf{P}$ that follows the policy $\pi$.

On the other hand, we have that

$$V_{\gamma,\mathsf{P}}^{\pi} = (I - \gamma\mathsf{P}_{\pi})^{-1}r_{\pi}. \tag{23}$$

Note that:

$$\mathsf{P}_{\pi}^*(I - \gamma\mathsf{P}_{\pi}) = \mathsf{P}_{\pi}^* - \gamma\mathsf{P}_{\pi}^* = (1-\gamma)\mathsf{P}_{\pi}^*, \tag{24}$$

where the above equation is from the fact that for every policy $\pi$, $\mathsf{P}_{\pi}\mathsf{P}_{\pi}^* = \mathsf{P}_{\pi}^*\mathsf{P}_{\pi} = \mathsf{P}_{\pi}^*\mathsf{P}_{\pi}^* = \mathsf{P}_{\pi}^*$. Thus,

$$\mathsf{P}_{\pi}^* = (1-\gamma)\mathsf{P}_{\pi}^*(I - \gamma\mathsf{P}_{\pi})^{-1}. \tag{25}$$

Hence,

$$g_{\mathsf{P}}^{\pi} = \mathsf{P}_{\pi}^* r_{\pi} = (1-\gamma)\mathsf{P}_{\pi}^*(I - \gamma\mathsf{P}_{\pi})^{-1}r_{\pi} = \mathsf{P}_{\pi}^*\cdot(1-\gamma)V_{\gamma,\mathsf{P}}^{\pi}. \tag{26}$$

Here, the last equation is from (23).

Since each row of $\mathsf{P}_\pi^*$ is a transition kernel of a unichain, $g_\mathsf{P}^\pi(s)$ is constant $\forall s$ and can be bounded as:

$$\min_s (1-\gamma) V_{\gamma,\mathsf{P}}^\pi(s) \le g_\mathsf{P}^\pi \le \max_s (1-\gamma) V_{\gamma,\mathsf{P}}^\pi(s). \tag{27}$$

Taking $\min_{\mathsf{P}\in\mathcal{P}}$ on both sides of inequality (27) implies that

$$\min_{\mathsf{P}\in\mathcal{P}} \min_s (1-\gamma) V_{\gamma,\mathsf{P}}^\pi(s) \le \min_{\mathsf{P}\in\mathcal{P}} g_\mathsf{P}^\pi = g_\mathcal{P}^\pi. \tag{28}$$

By interchanging the order of two $\min$ operators, we have that

$$\min_s (1-\gamma) \min_{\mathsf{P}\in\mathcal{P}} V_{\gamma,\mathsf{P}}^\pi(s) = \min_s (1-\gamma) V_{\gamma,\mathcal{P}}^\pi(s) \le g_\mathcal{P}^\pi. \tag{29}$$

On the other hand, we have

$$g_\mathcal{P}^\pi = \min_{\mathsf{P}\in\mathcal{P}} g_\mathsf{P}^\pi = \min_{\mathsf{P}\in\mathcal{P}} \left[ \mathsf{P}_\pi^* \cdot (1-\gamma) V_{\gamma,\mathsf{P}}^\pi \right]. \tag{30}$$

We denote by $Q_\gamma \in \mathcal{P}$ the worst case transition kernel of $V_{\gamma,\mathcal{P}}^\pi$. Then,

$$g_\mathcal{P}^\pi \le Q_\gamma^* (1-\gamma) V_{\gamma,Q_\gamma}^\pi \tag{31}$$
$$= Q_\gamma^* (1-\gamma) V_{\gamma,\mathcal{P}}^\pi \tag{32}$$
$$\le \max_s (1-\gamma) V_{\gamma,\mathcal{P}}^\pi(s), \tag{33}$$

where (31) is from (30), (32) is from the definition of $Q_\gamma$, and (33) is because for every distribution of $Q_\gamma$ we have $\mathbb{E}_{Q_\gamma}[(1-\gamma) V_{\gamma,Q_\gamma}^\pi] \le \max_s (1-\gamma) V_{\gamma,Q_\gamma}^\pi(s)$.

Combining (29) and (33), we get

$$\min_s (1-\gamma) V_{\gamma,\mathcal{P}}^\pi(s) \le g_\mathcal{P}^\pi \le \max_s (1-\gamma) V_{\gamma,\mathcal{P}}^\pi(s). \tag{34}$$

This implies that

$$\|g_\mathcal{P}^\pi - (1-\gamma) V_{\gamma,\mathcal{P}}^\pi\|_\infty \le \mathbf{Sp}((1-\gamma) V_{\gamma,\mathcal{P}}^\pi), \tag{35}$$

which completes the proof. $\qquad\square$

**Lemma 11.2.** *Under assumption 3.1, for any $\epsilon \in [0, \mathcal{H})$, if we set $\gamma := 1 - \frac{\epsilon}{\mathcal{H}}$, where $\mathcal{H} = \max_{\mathsf{P}\in\mathcal{P}} \mathbf{Sp}(h_\mathsf{P}^{\pi^*})$, then*

$$\mathbf{Sp}((1-\gamma) V_{\gamma,\mathcal{P}}^{\pi_\gamma}) \le \epsilon.$$

*Proof.* Since the MDP is assumed to be a unichain, both $g_\mathsf{P}^{\pi^*}$ and $g_\mathcal{P}^{\pi^*}$ are constant. Moreover, $g_\mathcal{P}^{\pi^*}$ and $h_\mathcal{P}^{\pi^*}$ satisfy the robust Bellman optimality equation (Wang et al., 2023e):

$$(g_\mathcal{P}^{\pi^*} + h_\mathcal{P}^{\pi^*})(s) = \max_a \left\{ r(s,a) + \sigma_{\mathcal{P}_s^a}(h_\mathcal{P}^{\pi^*}) \right\}, \tag{36}$$

where $\sigma_{\mathcal{P}_s^a}(V) \triangleq \min_{p\in\mathcal{P}_s^a} p^\top V$.

It is also known that for any $\gamma \in [0,1)$, the optimal robust discounted value function $V_{\gamma,\mathcal{P}}^{\pi_\gamma}$ satisfies the discounted Bellman equation:

$$V_{\gamma,\mathcal{P}}^{\pi_\gamma}(s) = \max_a \left\{ r(s,a) + \gamma \sigma_{\mathcal{P}_s^a}(V_{\gamma,\mathcal{P}}^{\pi_\gamma}) \right\}. \tag{37}$$

Next, we aim to rewrite the robust discounted Bellman equation to obtain a form similar to the Bellman equation for the average reward setting. First, we define

$$h_{\gamma,\mathcal{P}}^{\pi_\gamma,\pi^*} = V_{\gamma,\mathcal{P}}^{\pi_\gamma} - \frac{1}{1-\gamma}g_{\mathcal{P}}^{\pi^*}. \tag{38}$$

Note the fact that $h_{\gamma,\mathcal{P}}^{\pi_\gamma,\pi^*}$ and $V_{\gamma,\mathcal{P}}^{\pi_\gamma}$ have the same span because $g_{\mathcal{P}}^{\pi^*}$ is a constant:

$$\mathbf{Sp}(h_{\gamma,\mathcal{P}}^{\pi_\gamma,\pi^*}) = \mathbf{Sp}(V_{\gamma,\mathcal{P}}^{\pi^*}). \tag{39}$$

Now, we substitute (38) into (37) to obtain an equation similar to (36):

$$V_{\gamma,\mathcal{P}}^{\pi_\gamma}(s) = \left( h_{\gamma,\mathcal{P}}^{\pi_\gamma,\pi^*} + \frac{1}{1-\gamma}g_{\mathcal{P}}^{\pi^*} \right)(s) = \max_a \left\{ r(s,a) + \gamma\sigma_{\mathcal{P}_s^a}\left( h_{\gamma,\mathcal{P}}^{\pi_\gamma,\pi^*} + \frac{1}{1-\gamma}g_{\mathcal{P}}^{\pi^*} \right) \right\}$$

$$= \max_a \left\{ r(s,a) + \gamma\sigma_{\mathcal{P}_s^a}\left( h_{\gamma,\mathcal{P}}^{\pi_\gamma,\pi^*} \right) + \frac{\gamma}{1-\gamma}g_{\mathcal{P}}^{\pi^*} \right\}, \tag{40}$$

where (40) is because $\frac{\gamma}{1-\gamma}g_{\mathcal{P}}^{\pi^*}$ is a constant and can be taken from the support function. The above equation can be written as

$$\left( h_{\gamma,\mathcal{P}}^{\pi_\gamma,\pi^*} + \frac{1}{1-\gamma}g_{\mathcal{P}}^{\pi^*} \right)(s) = \max_a \left\{ r(s,a) + \gamma\sigma_{\mathcal{P}_s^a}\left( h_{\gamma,\mathcal{P}}^{\pi_\gamma,\pi^*} \right) \right\} + \frac{\gamma}{1-\gamma}g_{\mathcal{P}}^{\pi^*}. \tag{41}$$

This implies that

$$\left( h_{\gamma,\mathcal{P}}^{\pi_\gamma,\pi^*} + g_{\mathcal{P}}^{\pi^*} \right)(s) = \max_a \left\{ r(s,a) + \gamma\sigma_{\mathcal{P}_s^a}\left( h_{\gamma,\mathcal{P}}^{\pi_\gamma,\pi^*} \right) \right\}. \tag{42}$$

Combine (36) and (42) and $\forall s \in S$ we have

$$|h_{\mathcal{P}}^{\pi^*} - h_{\gamma,\mathcal{P}}^{\pi_\gamma,\pi^*}(s)| = |\max_a \left\{ r(s,a) + \sigma_{\mathcal{P}_s^a}(h_{\mathcal{P}}^{\pi^*}) \right\} - \max_a \left\{ r(s,a) + \gamma\sigma_{\mathcal{P}_s^a}\left( h_{\gamma,\mathcal{P}}^{\pi_\gamma,\pi^*} \right) \right\}| \tag{43}$$

$$\leq |\max_a \{ \sigma_{\mathcal{P}_s^a}(h_{\mathcal{P}}^{\pi^*}) - \gamma\sigma_{\mathcal{P}_s^a}\left( h_{\gamma,\mathcal{P}}^{\pi_\gamma,\pi^*} \right) \}| \tag{44}$$

$$= |\max_a \{ \sigma_{\mathcal{P}_s^a}(h_{\mathcal{P}}^{\pi^*}) - \gamma\sigma_{\mathcal{P}_s^a}\left( h_{\mathcal{P}}^{\pi^*} \right) + \gamma\sigma_{\mathcal{P}_s^a}\left( h_{\mathcal{P}}^{\pi^*} \right) - \gamma\sigma_{\mathcal{P}_s^a}\left( h_{\gamma,\mathcal{P}}^{\pi_\gamma,\pi^*} \right) \}| \tag{45}$$

$$\leq |\max_a \{ \gamma\sigma_{\mathcal{P}_s^a}\left( h_{\mathcal{P}}^{\pi^*} \right) - \gamma\sigma_{\mathcal{P}_s^a}\left( h_{\gamma,\mathcal{P}}^{\pi_\gamma,\pi^*} \right) \}| + |\max_a \{ (1-\gamma)\sigma_{\mathcal{P}_s^a}(h_{\mathcal{P}}^{\pi^*}) \}|. \tag{46}$$

And since $|\sigma_{\mathcal{P}_s^a}(V) - \sigma_{\mathcal{P}_s^a}(W)| \leq \|V - W\|_\infty$, and $|\sigma_{\mathcal{P}_s^a}(V)| \leq \|V\|_\infty$:

$$|h_{\mathcal{P}}^{\pi^*} - h_{\gamma,\mathcal{P}}^{\pi_\gamma,\pi^*}(s)| \leq \gamma\|h_{\mathcal{P}}^{\pi^*} - h_{\gamma,\mathcal{P}}^{\pi_\gamma,\pi^*}\|_\infty + (1-\gamma)\|h_{\mathcal{P}}^{\pi^*}\|_\infty. \tag{47}$$

Thus, it follows that,

$$\|h_{\mathcal{P}}^{\pi^*} - h_{\gamma,\mathcal{P}}^{\pi_\gamma,\pi^*}\|_\infty \leq \|h_{\mathcal{P}}^{\pi^*}\|_\infty. \tag{48}$$

Now, we combine (39) and (48):

$$\mathbf{Sp}(V_{\gamma,\mathcal{P}}^{\pi_\gamma}) = \mathbf{Sp}(h_{\gamma,\mathcal{P}}^{\pi_\gamma,\pi^*}) \leq 2\|h_{\gamma,\mathcal{P}}^{\pi_\gamma,\pi^*}\|_\infty \leq 4\|h_{\mathcal{P}}^{\pi^*}\|_\infty. \tag{49}$$

From Theorem 3.1 of (Wang et al., 2023e), for any policy $\pi$, there exists a transition kernel $\mathsf{P}_V \in \mathcal{P}_W$ such that $h_{\mathcal{P}}^\pi = h_{\mathsf{P}_V}^\pi + ce$ for $c \in \mathbb{R}$, where $e$ denotes the vector $(1,1,1,..,1) \in \mathbb{R}^{|S|}$. Hence, we have that

$$\mathbf{Sp}(V_{\gamma,\mathcal{P}}^{\pi_\gamma}) \leq 4\|h_{\mathcal{P}}^{\pi^*}\|_\infty = 4\|h_{\mathsf{P}_V}^{\pi^*}\|_\infty, \tag{50}$$

and

$$\mathbf{Sp}(V_{\gamma,\mathcal{P}}^{\pi_\gamma}) \leq 4\|h_{\mathsf{P}_V}^{\pi^*}\|_\infty \leq 4\mathbf{Sp}(h_{\mathsf{P}_V}^{\pi^{\pi^*}}) \leq 4\mathcal{H}, \tag{51}$$

which completes the proof. $\qquad\square$

Combine the above two results, then we direcly have the following result.

**Corollary 11.3.** *Under Assumption 3.1, for any $\gamma \in (0,1)$ and the policy $\pi_\gamma$, it holds that*

$$\|\frac{g_\mathcal{P}^{\pi_\gamma}}{1-\gamma} - V_{\gamma,\mathcal{P}}^{\pi_\gamma}\|_\infty \leq \mathcal{H}.$$

**Lemma 11.4.** *Under assumption 3.1, for any $\epsilon \in [0, \mathcal{H})$, if we set $\gamma := 1 - \frac{\epsilon}{\mathcal{H}}$, then $\boldsymbol{Sp}((1-\gamma)V_{\gamma,\mathcal{P}}^{\pi^*}) \leq \epsilon$.*

*Proof.* First, we utilize the definitions of the finite time-horizon reward function $V_{T,\mathsf{P}}^\pi(s) \triangleq \mathbb{E}_{\pi,\mathsf{P}}\left[\sum_0^{T-1} r_t | S_0 = s\right]$ and the other definition of bias $h_\mathsf{P}^\pi(s) \triangleq \lim_{T\to\infty}\left[V_{T,\mathsf{P}}^\pi - Tg_\mathsf{P}^\pi\right]$. Note that:

$$V_{\gamma,\mathcal{P}}^{\pi^*} = \min_{\mathsf{P}\in\mathcal{P}} V_{\gamma,\mathsf{P}}^{\pi^*} \tag{52}$$

$$= \min_{\mathsf{P}\in\mathcal{P}}\left(\lim_{T\to\infty}\sum_{t=0}^{T-1}\gamma^t\mathsf{P}_{\pi^*}^t r_{\pi^*}\right) \tag{53}$$

$$= \min_{\mathsf{P}\in\mathcal{P}}\left(\lim_{T\to\infty}V_{T,\mathsf{P}}^{\pi^*} - (1-\gamma)\sum_{t=1}^{T-1}\gamma^{t-1}\mathsf{P}_{\pi^*}^t V_{T-t,\mathsf{P}}^{\pi^*}\right). \tag{54}$$

Recall the worst case transition kernel $Q_\gamma$ where $Q_\gamma \in \mathcal{P}$:

$$V_{\gamma,\mathcal{P}}^{\pi^*} = V_{\gamma,Q_\gamma}^{\pi^*} = \lim_{T\to\infty}V_{T,Q_\gamma}^{\pi^*} - (1-\gamma)\sum_{t=1}^{T-1}\gamma^{t-1}Q_{\gamma,\pi^*}^t V_{T-t,Q_\gamma}^{\pi^*}. \tag{55}$$

We have:

$$\mathbf{Sp}(V_{\gamma,\mathcal{P}}^{\pi^*}) = \mathbf{Sp}(V_{\gamma,Q_\gamma}^{\pi^*}) \leq \lim_{T\to\infty}\sup \mathbf{Sp}(V_{T,Q_\gamma}^{\pi^*}) + (1-\gamma)\sum_{t=1}^{T-1}\gamma^{t-1}\mathbf{Sp}(V_{T-t,Q_\gamma}^{\pi^*}) \tag{56}$$

Now the objective is to find an upper bound for $\mathbf{Sp}(V_{T,Q_\gamma}^{\pi^*})$. From the definition of $h_{Q_\gamma}^\pi$, we have

$$h_{Q_\gamma}^\pi = \lim_{t\to\infty}(V_{t,Q_\gamma}^\pi - tg_{Q_\gamma}^\pi) \tag{57}$$

$$= \lim_{N\to\infty}\frac{1}{N}\sum_{t=1}^N(V_{t,Q_\gamma}^\pi - tg_{Q_\gamma}^\pi) \tag{58}$$

$$= \lim_{N\to\infty}\frac{1}{N}\sum_{t=T+1}^N(V_{t,Q_\gamma}^\pi - tg_{Q_\gamma}^\pi). \tag{59}$$

Thus, we have

$$h_{Q_\gamma}^\pi = \lim_{N\to\infty}\frac{1}{N}\sum_{t=T+1}^N[(V_{T,Q_\gamma}^\pi - Tg_{Q_\gamma}^\pi) + Q_{\gamma,\pi}^T V_{t-T,Q_\gamma}^\pi - (t-T)g_{Q_\gamma}^\pi]$$

$$= (V_{T,Q_\gamma}^\pi - Tg_{Q_\gamma}^\pi) + Q_{\gamma,\pi}^T\lim_{N\to\infty}\frac{1}{N}\sum_{t=T+1}^{N-T}(V_{t-T,Q_\gamma}^\pi - (t-T)g_{Q_\gamma}^\pi) \tag{60}$$

$$= V_{T,Q_\gamma}^\pi - Tg_{Q_\gamma}^\pi + Q_{\gamma,\pi}^T h_{Q_\gamma}^\pi.$$

It follows that

$$V_{T,Q_\gamma}^\pi = Tg_{Q_\gamma}^\pi + h_{Q_\gamma}^\pi - Q_{\gamma,\pi}^T h_{Q_\gamma}^\pi. \tag{61}$$

Thus, we have

$$\mathbf{Sp}(V^{\pi}_{T,Q_\gamma}) \leq 2\mathbf{Sp}(h^{\pi}_{Q_\gamma}). \tag{62}$$

Note that $\mathbf{Sp}(h^{\pi}_{\mathsf{P}})$ is continuous in $\mathsf{P}$ and $\mathcal{P}$ is a compact set, thus $\mathbf{Sp}(h^{\pi}_{\mathsf{P}}) \leq \mathcal{H}, \forall \mathsf{P} \in \mathcal{P}$. Now we can combine the result from (56) and (62):

$$\mathbf{Sp}(V^{\pi^*}_{\gamma,\mathcal{P}}) = \mathbf{Sp}(V^{\pi^*}_{\gamma,Q_\gamma}) \leq \lim_{T\to\infty} \sup \mathbf{Sp}(V^{\pi^*}_{T,Q_\gamma}) + (1-\gamma)\sum_{t=1}^{T-1}\gamma^{t-1}\mathbf{Sp}(V^{\pi^*}_{T-t,Q_\gamma}) \tag{63}$$

$$\leq 2\mathbf{Sp}(h^{\pi^*}_{\mathsf{P}})\left(1 + (1-\gamma)\sum_{t=1}^{\infty}\gamma^{t-1}\right) \tag{64}$$

$$= 4\mathbf{Sp}(h^{\pi^*}_{\mathsf{P}}). \tag{65}$$

Thus,

$$\mathbf{Sp}(V^{\pi^*}_{\gamma,\mathcal{P}}) \leq 4\mathcal{H}. \tag{66}$$

$\square$

As a direct corollary, it holds that

**Corollary 11.5.** *Under Assumption 3.1, for any $\gamma \in (0,1)$, it holds that*

$$\left\|\frac{g^{\pi^*}_{\mathcal{P}}}{1-\gamma} - V^{\pi^*}_{\gamma,\mathcal{P}}\right\|_\infty \leq 4\mathcal{H}.$$

*Proof.* From Lemma 11.1 and Equation (66), it holds that

$$\|g^{\pi^*}_{\mathcal{P}} - (1-\gamma)V^{\pi^*}_{\gamma,\mathcal{P}}\|_\infty \leq \mathbf{Sp}((1-\gamma)V^{\pi^*}_{\gamma,\mathcal{P}}) \leq 4(1-\gamma)\mathcal{H},$$

which completes the proof. $\square$

We are now ready to prove the main theorem.

**Theorem 11.6.** *(Restatement of Theorem 3.4) Under Assumption 3.1, for any $\epsilon$, if we set $\gamma := 1 - \frac{\epsilon}{\mathcal{H}}$, then any $\epsilon_\gamma$-optimal policy $\hat{\pi}_\gamma$ for the robust $\gamma$-DMDP[8] is also an $\mathcal{O}(\epsilon)$-optimal policy for the robust AMDP, i.e.,*

$$g^{\pi^*}_{\mathcal{P}} - g^{\hat{\pi}_\gamma}_{\mathcal{P}} \leq \left(8 + \frac{5\epsilon_\gamma}{\mathcal{H}}\right)\epsilon. \tag{67}$$

*Specifically, an $\mathcal{O}(\mathcal{H})$-optimal robust policy for the robust DMDP is an $\mathcal{O}(\epsilon)$-optimal robust policy under the average reward.*

*Proof.* Under Assumption 3.1, for any $\epsilon \in (0,\mathcal{H}]$ and any $\delta \in (0,1]$, we consider $\gamma = 1 - \frac{\epsilon}{\mathcal{H}}$. Suppose $\pi_\gamma$ is an optimal policy of the robust DMDP and $\hat{\pi}_\gamma$ is an $\epsilon_\gamma$-optimal policy in the robust DMDP. Then,

$$\|(1-\gamma)V^{\pi_\gamma}_{\gamma,\mathcal{P}} - (1-\gamma)V^{\hat{\pi}_\gamma}_{\gamma,\mathcal{P}}\|_\infty \leq (1-\gamma)\epsilon_\gamma. \tag{68}$$

Considering (35), (51), and (66), we have that:

$$\mathbf{Sp}((1-\gamma)V^{\hat{\pi}_\gamma}_{\gamma,\mathcal{P}}) \leq \mathbf{Sp}((1-\gamma)V^{\pi_\gamma}_{\gamma,\mathcal{P}}) + 2\|(1-\gamma)V^{\pi_\gamma}_{\gamma,\mathcal{P}} - (1-\gamma)V^{\hat{\pi}_\gamma}_{\gamma,\mathcal{P}}\|_\infty$$
$$\leq 4\epsilon + 2(1-\gamma)\epsilon_\gamma. \tag{69}$$

---

[8] For a robust DMDP, an $\epsilon$-optimal policy is some policy $\pi$ such that $V^*_{\gamma,\mathcal{P}}(s) - V^{\pi}_{\gamma,\mathcal{P}}(s) \leq \epsilon, \forall s \in \mathcal{S}$.

Moreover, because of the optimality of $\pi_\gamma$,

$$V_{\gamma,\mathcal{P}}^{\pi^*} \le V_{\gamma,\mathcal{P}}^{\pi_\gamma}, \tag{70}$$

recalling that $\pi^*$ is an optimal policy of the robust AMDP.

By merging Lemma 11.1, (68), (69), and (70) we have:

$$g_{\mathcal{P}}^{\pi^*} \le (1-\gamma)V_{\gamma,\mathcal{P}}^{\pi^*} + 4\epsilon + 2(1-\gamma)\epsilon_\gamma \tag{71}$$

$$\le (1-\gamma)V_{\gamma,\mathcal{P}}^{\pi_\gamma} + 4\epsilon + 2(1-\gamma)\epsilon_\gamma \tag{72}$$

$$\le (1-\gamma)V_{\gamma,\mathcal{P}}^{\hat{\pi}_\gamma} + 4\epsilon + 3(1-\gamma)\epsilon_\gamma \tag{73}$$

$$\le g_{\mathcal{P}}^{\hat{\pi}_\gamma} + 8\epsilon + 5(1-\gamma)\epsilon_\gamma , \tag{74}$$

which completes the proof. $\qquad\square$

## 12. Proof of Theorem 4.4 Part 1

We first note that the result under the case $R \ge \frac{1}{\mathcal{H}}$ can be directly obtained from the existing result in (Shi et al., 2023). Specifically, it is shown in (Shi et al., 2023) that learning an $\epsilon_\gamma$-optimal policy for a $\gamma$-discounted robust MDP requires samples of size

$$\mathcal{N}(\mathcal{S}, \mathcal{A}, \hat{\mathcal{P}}, \gamma, \epsilon_\gamma) = \frac{CSA \log \frac{cSA}{\delta}}{(1-\gamma)^2 \max\{R, 1-\gamma\}\epsilon_\gamma^2}. \tag{75}$$

Thus, by setting $\gamma = 1 - \frac{\epsilon}{\mathcal{H}}$ and $\epsilon_\gamma = \mathcal{H}$ as Theorem 4.1, we have that the complexity to learn an $\epsilon$-optimal policy for the robust average reward MDP is

$$\frac{CSA \log \frac{cSA}{\delta}}{(1-\gamma)^2 \max\{R, 1-\gamma\}\epsilon_\gamma^2} = \frac{CSA \log \frac{cSA}{\delta}}{\epsilon^2 R} \le \frac{CSA\mathcal{H} \log \frac{cSA}{\delta}}{\epsilon^2}. \tag{76}$$

We hence mainly focus on the case of $R \le \frac{1}{\mathcal{H}}$. In the following proof, we denote $\epsilon_\gamma$ as $\epsilon$.

We first introduce some notation. Let $\hat{\mathsf{P}}$ and $\hat{\mathcal{P}}$ be the estimated nominal kernel and estimated uncertainty set. For any policy $\pi$, let $\tilde{V}_{\gamma,\hat{\mathcal{P}}}^\pi, V_{\gamma,\hat{\mathcal{P}}}^\pi, V_{\gamma,\mathcal{P}}^\pi$ be the robust value function w.r.t. perturbed reward and estimated uncertainty set, unperturbed reward and estimated uncertainty set, unperturbed reward and the true uncertainty set. The optimal robust policy w.r.t. $\tilde{V}_{\hat{\mathcal{P}}}^\pi$ is denoted by $\tilde{\pi}^*$, and the corresponding optimal robust value functions are denoted by $\tilde{V}_{\gamma,\hat{\mathcal{P}}}^*$ and $\tilde{Q}_{\gamma,\hat{\mathcal{P}}}^*$.

We consider a general policy $\pi$, and we denote the worst-case kernel of any vector $V$ under $\hat{\mathcal{P}}$ and $\mathcal{P}$ by $\hat{\mathsf{P}}_w^{\pi,V}$ and $\mathsf{P}_w^{\pi,V}$. Then, it holds that

$$
\begin{aligned}
V_{\gamma,\hat{\mathcal{P}}}^\pi - V_{\gamma,\mathcal{P}}^\pi &= r_\pi + \gamma\hat{\mathsf{P}}_w^{\pi,\hat{V}}V_{\gamma,\hat{\mathcal{P}}}^\pi - \left(r_\pi + \gamma\mathsf{P}_w^{\pi,V}V_{\gamma,\mathcal{P}}^\pi\right) \\
&= \left(\gamma\hat{\mathsf{P}}_w^{\pi,\hat{V}}V_{\gamma,\hat{\mathcal{P}}}^\pi - \gamma\mathsf{P}_w^{\pi,\hat{V}}V_{\gamma,\hat{\mathcal{P}}}^\pi\right) + \left(\gamma\mathsf{P}_w^{\pi,\hat{V}}V_{\gamma,\hat{\mathcal{P}}}^\pi - \gamma\mathsf{P}_w^{\pi,V}V_{\gamma,\mathcal{P}}^\pi\right) \\
&\overset{(i)}{\le} \gamma\left(\mathsf{P}_w^{\pi,V}V_{\gamma,\hat{\mathcal{P}}}^\pi - \mathsf{P}_w^{\pi,V}V_{\gamma,\mathcal{P}}^\pi\right) + \left(\gamma\hat{\mathsf{P}}_w^{\pi,\hat{V}}V_{\gamma,\hat{\mathcal{P}}}^\pi - \gamma\mathsf{P}_w^{\pi,\hat{V}}V_{\gamma,\hat{\mathcal{P}}}^\pi\right),
\end{aligned}
$$

where (i) holds by observing that

$$\mathsf{P}_w^{\pi,\hat{V}}V_{\gamma,\hat{\mathcal{P}}}^\pi \le \mathsf{P}_w^{\pi,V}V_{\gamma,\hat{\mathcal{P}}}^\pi$$

due to the worst-case kernel. Rearranging terms leads to

$$V_{\gamma,\hat{\mathcal{P}}}^\pi - V_{\gamma,\mathcal{P}}^\pi \le \gamma\left(I - \gamma\mathsf{P}_w^{\pi,V}\right)^{-1}\left(\hat{\mathsf{P}}_w^{\pi,\hat{V}}V_{\gamma,\hat{\mathcal{P}}}^\pi - \mathsf{P}_w^{\pi,\hat{V}}V_{\gamma,\hat{\mathcal{P}}}^\pi\right). \tag{77}$$

Similarly, we can also deduce

$$
\begin{aligned}
V_{\gamma,\hat{\mathcal{P}}}^{\pi} - V_{\gamma,\mathcal{P}}^{\pi} &= r_{\pi} + \gamma\hat{\mathsf{P}}_{w}^{\pi,\widehat{V}}V_{\gamma,\hat{\mathcal{P}}}^{\pi} - \left(r_{\pi} + \gamma\mathsf{P}_{w}^{\pi,V}V_{\gamma,\mathcal{P}}^{\pi}\right) \\
&= \left(\gamma\hat{\mathsf{P}}_{w}^{\pi,\widehat{V}}V_{\gamma,\hat{\mathcal{P}}}^{\pi} - \gamma\mathsf{P}_{w}^{\pi,\widehat{V}}V_{\gamma,\hat{\mathcal{P}}}^{\pi}\right) + \left(\gamma\mathsf{P}_{w}^{\pi,\widehat{V}}V_{\gamma,\hat{\mathcal{P}}}^{\pi} - \gamma\mathsf{P}_{w}^{\pi,V}V_{\gamma,\mathcal{P}}^{\pi}\right) \\
&\geq \gamma\left(\mathsf{P}_{w}^{\pi,\widehat{V}}V_{\gamma,\hat{\mathcal{P}}}^{\pi} - \mathsf{P}_{w}^{\pi,\widehat{V}}V_{\gamma,\mathcal{P}}^{\pi}\right) + \left(\gamma\hat{\mathsf{P}}_{w}^{\pi,\widehat{V}}V_{\gamma,\hat{\mathcal{P}}}^{\pi} - \gamma\mathsf{P}_{w}^{\pi,\widehat{V}}V_{\gamma,\hat{\mathcal{P}}}^{\pi}\right) \\
&\geq \gamma\left(I - \gamma\mathsf{P}_{w}^{\pi,\widehat{V}}\right)^{-1}\left(\hat{\mathsf{P}}_{w}^{\pi,\widehat{V}}V_{\gamma,\hat{\mathcal{P}}}^{\pi} - \mathsf{P}_{w}^{\pi,\widehat{V}}V_{\gamma,\hat{\mathcal{P}}}^{\pi}\right).
\end{aligned}
\tag{78}
$$

Combining (77) and (78), we arrive at

$$
\begin{aligned}
\left\|V_{\gamma,\hat{\mathcal{P}}}^{\pi} - V_{\gamma,\mathcal{P}}^{\pi}\right\|_{\infty} \leq \gamma\max\Big\{ &\left\|\left(I - \gamma\mathsf{P}_{w}^{\pi,V}\right)^{-1}\left(\hat{\mathsf{P}}_{w}^{\pi,\widehat{V}}V_{\gamma,\hat{\mathcal{P}}}^{\pi} - \mathsf{P}_{w}^{\pi,\widehat{V}}V_{\gamma,\hat{\mathcal{P}}}^{\pi}\right)\right\|_{\infty}, \\
&\left\|\left(I - \gamma\mathsf{P}_{w}^{\pi,\widehat{V}}\right)^{-1}\left(\hat{\mathsf{P}}_{w}^{\pi,\widehat{V}}V_{\gamma,\hat{\mathcal{P}}}^{\pi} - \mathsf{P}_{w}^{\pi,\widehat{V}}V_{\gamma,\hat{\mathcal{P}}}^{\pi}\right)\right\|_{\infty}\Big\}.
\end{aligned}
\tag{79}
$$

By decomposing the error in a symmetric way, we can similarly obtain

$$
\begin{aligned}
\left\|V_{\gamma,\hat{\mathcal{P}}}^{\pi} - V_{\gamma,\mathcal{P}}^{\pi}\right\|_{\infty} \leq \gamma\max\Big\{ &\left\|\left(I - \gamma\hat{\mathsf{P}}_{w}^{\pi,\widehat{V}}\right)^{-1}\left(\hat{\mathsf{P}}_{w}^{\pi,V}V_{\gamma,\mathcal{P}}^{\pi} - \mathsf{P}_{w}^{\pi,V}V_{\gamma,\mathcal{P}}^{\pi}\right)\right\|_{\infty}, \\
&\left\|\left(I - \gamma\hat{\mathsf{P}}_{w}^{\pi,V}\right)^{-1}\left(\hat{\mathsf{P}}_{w}^{\pi,V}V_{\gamma,\mathcal{P}}^{\pi} - \mathsf{P}_{w}^{\pi,V}V_{\gamma,\mathcal{P}}^{\pi}\right)\right\|_{\infty}\Big\}.
\end{aligned}
\tag{80}
$$

## 12.1. Part A: $\|\tilde{V}_{\gamma,\hat{\mathcal{P}}}^{\pi^{*}} - V_{\gamma,\mathcal{P}}^{\pi^{*}}\|$

Consider $|\tilde{V}_{\gamma,\hat{\mathcal{P}}}^{\pi^{*}}(s) - V_{\gamma,\hat{\mathcal{P}}}^{\pi^{*}}(s)|$ for some state $s$. If $\tilde{V}_{\gamma,\hat{\mathcal{P}}}^{\pi^{*}}(s) - V_{\gamma,\hat{\mathcal{P}}}^{\pi^{*}}(s) > 0$, then

$$
\begin{aligned}
|\tilde{V}_{\gamma,\hat{\mathcal{P}}}^{\pi^{*}}(s) - V_{\gamma,\hat{\mathcal{P}}}^{\pi^{*}}(s)| &= \tilde{V}_{\gamma,\hat{\mathcal{P}}}^{\pi^{*}}(s) - V_{\gamma,\hat{\mathcal{P}}}^{\pi^{*}}(s) \\
&= (I - \gamma\tilde{\mathsf{P}}_{w}^{\pi^{*}})^{-1}\tilde{r}^{\pi^{*}} - (I - \gamma\mathsf{P}_{w}^{\pi^{*}})^{-1}r^{\pi^{*}} \\
&\leq (I - \gamma\mathsf{P}_{w}^{\pi^{*}})^{-1}\tilde{r}^{\pi^{*}} - (I - \gamma\mathsf{P}_{w}^{\pi^{*}})^{-1}r^{\pi^{*}} \\
&\leq \frac{\epsilon}{6},
\end{aligned}
\tag{81}
$$

where $\tilde{\mathsf{P}}_{w}^{\pi^{*}}$ and $\mathsf{P}_{w}^{\pi^{*}}$ are the corresponding worst-case transition kernel.

A similar result can also be obtained for the other case, hence it holds that

$$
\|\tilde{V}_{\gamma,\hat{\mathcal{P}}}^{\pi^{*}}(s) - V_{\gamma,\hat{\mathcal{P}}}^{\pi^{*}}\| \leq \frac{\epsilon}{6}.
\tag{82}
$$

We thus have that

$$
\|\tilde{V}_{\gamma,\hat{\mathcal{P}}}^{\pi^{*}} - V_{\gamma,\mathcal{P}}^{\pi^{*}}\| \leq \|\tilde{V}_{\gamma,\hat{\mathcal{P}}}^{\pi^{*}} - V_{\gamma,\hat{\mathcal{P}}}^{\pi^{*}}\| + \|V_{\gamma,\hat{\mathcal{P}}}^{\pi^{*}} - V_{\gamma,\mathcal{P}}^{\pi^{*}}\| \leq \frac{\epsilon}{6} + \|V_{\gamma,\hat{\mathcal{P}}}^{\pi^{*}} - V_{\gamma,\mathcal{P}}^{\pi^{*}}\|,
\tag{83}
$$

and it suffices to study the second term $\|V_{\gamma,\hat{\mathcal{P}}}^{\pi^{*}} - V_{\gamma,\mathcal{P}}^{\pi^{*}}\|$. Applying (80) further implies that

$$
\begin{aligned}
\|V_{\gamma,\hat{\mathcal{P}}}^{\pi^{*}} - V_{\gamma,\mathcal{P}}^{\pi^{*}}\| \leq \gamma\max\Big\{ &\left\|\left(I - \gamma\hat{\mathsf{P}}_{w}^{\pi^{*},\widehat{V}}\right)^{-1}\left(\hat{\mathsf{P}}_{w}^{\pi^{*},V}V_{\gamma,\mathcal{P}}^{\pi^{*}} - \mathsf{P}_{w}^{\pi^{*},V}V_{\gamma,\mathcal{P}}^{\pi^{*}}\right)\right\|_{\infty}, \\
&\left\|\left(I - \gamma\hat{\mathsf{P}}_{w}^{\pi^{*},V}\right)^{-1}\left(\hat{\mathsf{P}}_{w}^{\pi^{*},V}V_{\gamma,\mathcal{P}}^{\pi^{*}} - \mathsf{P}_{w}^{\pi^{*},V}V_{\gamma,\mathcal{P}}^{\pi^{*}}\right)\right\|_{\infty}\Big\}.
\end{aligned}
\tag{84}
$$

We first consider the case that $\max\left\{\left\|\left(I - \gamma\hat{\mathsf{P}}_{w}^{\pi^{*},\widehat{V}}\right)^{-1}\left(\hat{\mathsf{P}}_{w}^{\pi^{*},V}V_{\gamma,\mathcal{P}}^{\pi^{*}} - \mathsf{P}_{w}^{\pi^{*},V}V_{\gamma,\mathcal{P}}^{\pi^{*}}\right)\right\|_{\infty}, \left\|\left(I - \gamma\hat{\mathsf{P}}_{w}^{\pi^{*},V}\right)^{-1}\left(\hat{\mathsf{P}}_{w}^{\pi^{*},V}V_{\gamma,\mathcal{P}}^{\pi^{*}} - \mathsf{P}_{w}^{\pi^{*},V}V_{\gamma,\mathcal{P}}^{\pi}\right)\right\|_{\infty}\right\} = \left\|\left(I - \gamma\hat{\mathsf{P}}_{w}^{\pi^{*},\widehat{V}}\right)^{-1}\left(\hat{\mathsf{P}}_{w}^{\pi^{*},V}V_{\gamma,\mathcal{P}}^{\pi^{*}} - \mathsf{P}_{w}^{\pi^{*},V}V_{\gamma,\mathcal{P}}^{\pi^{*}}\right)\right\|_{\infty}$.

We first apply the following lemma.

**Lemma 12.1.** *(Lemma 11 of (Shi et al., 2023)) Consider any $\delta \in (0,1)$. Setting $N \geq \log(\frac{18SAN}{\delta})$, with probability at least $1 - \delta$, one has*

$$\left| \hat{\mathsf{P}}_w^{\pi^*,V} V_{\gamma,\mathcal{P}}^{\pi^*} - \mathsf{P}_w^{\pi^*,V} V_{\gamma,\mathcal{P}}^{\pi^*} \right| \leq 2\sqrt{\frac{\log(\frac{18SAN}{\delta})}{N}} \sqrt{\mathrm{Var}_{\mathsf{P}^{\pi^*}}(V_{\gamma,\mathcal{P}}^{\pi^*})} + \frac{\log(\frac{18SAN}{\delta})}{N(1-\gamma)} 1. \tag{85}$$

Thus, it holds that

$$\left\| \left(I - \gamma\hat{\mathsf{P}}_w^{\pi^*,\hat{V}}\right)^{-1} \left( \hat{\mathsf{P}}_w^{\pi^*,V} V_{\gamma,\mathcal{P}}^{\pi^*} - \mathsf{P}_w^{\pi^*,V} V_{\gamma,\mathcal{P}}^{\pi^*} \right) \right\|_\infty$$

$$\leq 2\sqrt{\frac{\log(\frac{18SAN}{\delta})}{N}} \left\| \left(I - \gamma\hat{\mathsf{P}}_w^{\pi^*,\hat{V}}\right)^{-1} \sqrt{\mathrm{Var}_{\mathsf{P}^{\pi^*}}(V_{\gamma,\mathcal{P}}^{\pi^*})} \right\|_\infty + \frac{\log(\frac{18SAN}{\delta})}{N(1-\gamma)^2}$$

$$\leq \underbrace{2\sqrt{\frac{\log(\frac{18SAN}{\delta})}{N}} \left\| \left(I - \gamma\hat{\mathsf{P}}_w^{\pi^*,\hat{V}}\right)^{-1} \sqrt{\mathrm{Var}_{\hat{\mathsf{P}}_w^{\pi^*},\hat{v}}(V_{\gamma,\hat{\mathcal{P}}}^{\pi^*})} \right\|_\infty}_{A1}$$

$$+ \underbrace{2\sqrt{\frac{\log(\frac{18SAN}{\delta})}{N}} \left\| \left(I - \gamma\hat{\mathsf{P}}_w^{\pi^*,\hat{V}}\right)^{-1} \sqrt{\mathrm{Var}_{\hat{\mathsf{P}}_w^{\pi^*},\hat{v}}(V_{\gamma,\mathcal{P}}^{\pi^*} - V_{\gamma,\hat{\mathcal{P}}}^{\pi^*})} \right\|_\infty}_{A2}$$

$$+ \underbrace{2\sqrt{\frac{\log(\frac{18SAN}{\delta})}{N}} \left\| \left(I - \gamma\hat{\mathsf{P}}_w^{\pi^*,\hat{V}}\right)^{-1} \sqrt{\left| \mathrm{Var}_{\hat{\mathsf{P}}_w^{\pi^*},\hat{v}}(V_{\gamma,\mathcal{P}}^{\pi^*}) - \mathrm{Var}_{\hat{\mathsf{P}}^{\pi^*}}(V_{\gamma,\mathcal{P}}^{\pi^*}) \right|} \right\|_\infty}_{A3}$$

$$+ \underbrace{2\sqrt{\frac{\log(\frac{18SAN}{\delta})}{N}} \left\| \left(I - \gamma\hat{\mathsf{P}}_w^{\pi^*,\hat{V}}\right)^{-1} \left( \sqrt{\mathrm{Var}_{\mathsf{P}^{\pi^*}}(V_{\gamma,\mathcal{P}}^{\pi^*})} - \sqrt{\mathrm{Var}_{\hat{\mathsf{P}}^{\pi^*}}(V_{\gamma,\mathcal{P}}^{\pi^*})} \right) \right\|_\infty}_{A4} + \frac{\log(\frac{18SAN}{\delta})}{N(1-\gamma)^2}. \tag{86}$$

**Term $A1$.** We note that $\mathsf{Q} \triangleq \hat{\mathsf{P}}_w^{\hat{V}} = \arg\min_{\mathsf{P} \in \hat{\mathcal{P}}} \mathsf{P} V_{\gamma,\hat{\mathcal{P}}}^{\pi^*}$, thus we have that

$$V_{\gamma,\hat{\mathcal{P}}}^{\pi^*} = r^{\pi^*} + \gamma\sigma_{\hat{\mathcal{P}}}^{\pi^*}(V_{\gamma,\hat{\mathcal{P}}}^{\pi^*}) = r^{\pi^*} + \gamma\mathsf{Q}^{\pi^*} V_{\gamma,\hat{\mathcal{P}}}^{\pi^*}; \tag{87}$$

On the other hand, we have that

$$V_{\gamma,\mathsf{Q}}^{\pi^*} = r^{\pi^*} + \gamma\mathsf{Q}^{\pi^*} V_{\gamma,\mathsf{Q}}^{\pi^*}, \tag{88}$$

hence both $V_{\gamma,\hat{\mathcal{P}}}^{\pi^*}$ and $V_{\gamma,\mathsf{Q}}^{\pi^*}$ are fixed points of the Bellman operator w.r.t. $\mathsf{Q}$, which implies they are identical $V_{\gamma,\hat{\mathcal{P}}}^{\pi^*} = V_{\gamma,\mathsf{Q}}^{\pi^*}$.

Thus, the term $\left(I - \gamma\hat{\mathsf{P}}_w^{\pi^*,\hat{V}}\right)^{-1} \sqrt{\mathrm{Var}_{\hat{\mathsf{P}}_w^{\pi^*},\hat{v}}(V_{\gamma,\hat{\mathcal{P}}}^{\pi^*})}$ can be rewritten as

$$\left(I - \gamma\hat{\mathsf{P}}_w^{\pi^*,\hat{V}}\right)^{-1} \sqrt{\mathrm{Var}_{\hat{\mathsf{P}}_w^{\pi^*},\hat{v}}(V_{\gamma,\hat{\mathcal{P}}}^{\pi^*})} = \left(I - \gamma\mathsf{Q}^{\pi^*}\right)^{-1} \sqrt{\mathrm{Var}_{\mathsf{Q}^{\pi^*}}(V_{\gamma,\mathsf{Q}}^{\pi^*})}, \tag{89}$$

and the term $A1$ can be rewritten as

$$A1 = 2\sqrt{\frac{\log(\frac{18SAN}{\delta})}{N}} \left\| \left(I - \gamma\mathsf{Q}^{\pi^*}\right)^{-1} \sqrt{\mathrm{Var}_{\mathsf{Q}^{\pi^*}}(V_{\gamma,\mathsf{Q}}^{\pi^*})} \right\|_\infty. \tag{90}$$

We then apply the following lemma to bound $A1$, which is the main result for the complexity improvement.

**Lemma 12.2.** *For any policy $\pi$, if $N \geq \mathcal{O}\left(\frac{\log\frac{SA}{(1-\gamma)\delta\epsilon}}{1-\gamma}\right)$, it holds with probability at least $1 - \delta$ that*

$$\left\| \left(I - \gamma\mathsf{Q}^{\pi^*}\right)^{-1} \sqrt{\mathrm{Var}_{\mathsf{Q}^{\pi^*}}(V_{\gamma,\mathsf{Q}}^{\pi^*})} \right\|_\infty \leq \sqrt{\frac{c_1\mathcal{H}}{(1-\gamma)^2}}. \tag{91}$$

The proof of this lemma can be derived similarly to the ones in (Zurek & Chen, 2023). For completeness, we provide its proof in Section 12.4.

Thus, it holds that

$$A1 \leq \sqrt{\frac{2c_1 \log(\frac{18SAN}{\delta})\mathcal{H}}{(1-\gamma)^2 N}}. \tag{92}$$

**Term $A2$.** It holds that

$$A2 = 2\sqrt{\frac{\log(\frac{18SAN}{\delta})}{N}} \left\| \left(I - \gamma \hat{\mathsf{P}}_w^{\pi,\hat{V}}\right)^{-1} \sqrt{\mathrm{Var}_{\hat{\mathsf{P}}_w^{\pi,\hat{V}}}(V_{\gamma,\mathcal{P}}^{\pi^*} - V_{\gamma,\hat{\mathcal{P}}}^{\pi^*})} \right\|_\infty$$

$$\leq 2\sqrt{\frac{\log(\frac{18SAN}{\delta})}{N(1-\gamma)^2}} \left\| V_{\gamma,\mathcal{P}}^{\pi^*} - V_{\gamma,\hat{\mathcal{P}}}^{\pi^*} \right\|_\infty. \tag{93}$$

**Term $A3$.** It holds that

$$\left(I - \gamma \hat{\mathsf{P}}_w^{\pi,\hat{V}}\right)^{-1} \sqrt{\left| \mathrm{Var}_{\hat{\mathsf{P}}_w^{\pi,\hat{V}}}(V_{\gamma,\mathcal{P}}^{\pi^*}) - \mathrm{Var}_{\hat{\mathsf{P}}^{\pi^*}}(V_{\gamma,\mathcal{P}}^{\pi^*}) \right|}$$

$$= \left(I - \gamma \hat{\mathsf{P}}_w^{\pi,\hat{V}}\right)^{-1} \sqrt{\left| \prod^{\pi^*}(\mathrm{Var}_{\hat{\mathsf{P}}_w^{\pi,\hat{V}}}(V_{\gamma,\mathcal{P}}^{\pi^*}) - \mathrm{Var}_{\hat{\mathsf{P}}}(V_{\gamma,\mathcal{P}}^{\pi^*})) \right|}$$

$$\leq \left(I - \gamma \hat{\mathsf{P}}_w^{\pi,\hat{V}}\right)^{-1} \sqrt{\left\| \mathrm{Var}_{\hat{\mathsf{P}}_w^{\pi,\hat{V}}}(V_{\gamma,\mathcal{P}}^{\pi^*}) - \mathrm{Var}_{\hat{\mathsf{P}}}(V_{\gamma,\mathcal{P}}^{\pi^*}) \right\|}. \tag{94}$$

Note that both $\hat{\mathsf{P}}_w^{\pi,\hat{V}}, \hat{\mathsf{P}}$ belong to the uncertainty set $\hat{\mathcal{P}}$, hence $\|\hat{\mathsf{P}}_w^{\pi,\hat{V}} - \hat{\mathsf{P}}\|_1 \leq 2R$, which further implies that

$$|\mathrm{Var}_{\hat{\mathsf{P}}_w^{\pi,\hat{V}}}(V_{\gamma,\mathcal{P}}^{\pi^*}) - \mathrm{Var}_{\hat{\mathsf{P}}}(V_{\gamma,\mathcal{P}}^{\pi^*}))|_{s,a}$$

$$= |\mathrm{Var}_{\hat{\mathsf{P}}_w^{\pi,\hat{V}}}(V_{\gamma,\mathcal{P}}^{\pi^*} - \frac{1}{1-\gamma}g_{\mathcal{P}}^{\pi^*}) - \mathrm{Var}_{\hat{\mathsf{P}}}(V_{\gamma,\mathcal{P}}^{\pi^*}) - \frac{1}{1-\gamma}g_{\mathcal{P}}^{\pi^*})|_{s,a}$$

$$\leq \|\hat{\mathsf{P}}_w^{\pi,\hat{V}} - \hat{\mathsf{P}}\|_1 \|V_{\gamma,\mathcal{P}}^{\pi^*} - \frac{1}{1-\gamma}g_{\mathcal{P}}^{\pi^*}\|^2$$

$$\leq 2R\mathcal{H}^2, \tag{95}$$

where the last inequality is from Lemma 11.1 and Lemma 11.2.

Since $R \leq \frac{1}{\mathcal{H}}$, it holds that

$$A3 \leq 2\sqrt{\frac{\log(\frac{18SAN}{\delta})}{N}} \left\| \left(I - \gamma \hat{\mathsf{P}}_w^{\pi,\hat{V}}\right)^{-1} \sqrt{2\mathcal{H}} \right\| \leq 2\sqrt{\frac{\log(\frac{18SAN}{\delta})\mathcal{H}}{N(1-\gamma)^2}}. \tag{96}$$

**Term $A4$.** We directly apply Lemma 6 of (Panaganti & Kalathil, 2022) and Lemma 11 of (Shi et al., 2023), and it implies that

$$A4 \leq \frac{4\log(\frac{18SAN}{\delta})}{N(1-\gamma)^2}. \tag{97}$$

We then plug (92), (93), (96) and (97) in (86), and we have that

$$\|V_{\gamma,\hat{\mathcal{P}}}^{\pi^*} - V_{\gamma,\mathcal{P}}^{\pi^*}\|$$

$$\leq \left\| \left(I - \gamma \hat{\mathsf{P}}_w^{\pi,\hat{V}}\right)^{-1} \left(\hat{\mathsf{P}}_w^{\pi,\hat{V}} V_{\gamma,\mathcal{P}}^{\pi} - \mathsf{P}_w^{\pi,V} V_{\gamma,\mathcal{P}}^{\pi}\right) \right\|_\infty$$

$$\leq \frac{\log(\frac{18SAN}{\delta})}{N(1-\gamma)^2} + \sqrt{\frac{2c_1 \log(\frac{18SAN}{\delta})\mathcal{H}}{(1-\gamma)^2 N}} + 2\sqrt{\frac{\log(\frac{18SAN}{\delta})}{N(1-\gamma)^2}} \left\| V_{\gamma,\mathcal{P}}^{\pi^*} - V_{\gamma,\hat{\mathcal{P}}}^{\pi^*} \right\|_\infty + 2\sqrt{\frac{\log(\frac{18SAN}{\delta})\mathcal{H}}{N(1-\gamma)^2}} + \frac{4\log(\frac{18SAN}{\delta})}{N(1-\gamma)^2}. \tag{98}$$

We note that if we set $N \geq \frac{32 \log(\frac{18SAN}{\delta})}{(1-\gamma)^2}$, it holds that

$$\|V_{\gamma,\hat{\mathcal{P}}}^{\pi^*} - V_{\gamma,\mathcal{P}}^{\pi^*}\| \leq \frac{C_1 \log(\frac{18SAN}{\delta})}{N(1-\gamma)^2} + \sqrt{\frac{2C_2 \log(\frac{18SAN}{\delta})\mathcal{H}}{(1-\gamma)^2 N}}, \tag{99}$$

which completes the first term in (80).

To bound the second term in (80), following eq (69) in (Shi et al., 2023), we have that

$$\left\| \left( I - \gamma \hat{\mathsf{P}}_w^{\pi^*,V} \right)^{-1} \left( \hat{\mathsf{P}}_w^{\pi^*,V} V_{\gamma,\mathcal{P}}^{\pi^*} - \mathsf{P}_w^{\pi^*,V} V_{\gamma,\mathcal{P}}^{\pi^*} \right) \right\|_\infty$$
$$\leq 2\sqrt{\frac{\log(\frac{18SAN}{\delta})}{N}} \left\| \left( I - \gamma \hat{\mathsf{P}}_w^{\pi^*,V} \right)^{-1} \sqrt{\mathrm{Var}_{\mathsf{P}^{\pi^*}}(V_{\gamma,\mathcal{P}}^{\pi^*})} \right\|_\infty + \frac{\log(\frac{18SAN}{\delta})}{N(1-\gamma)^2}. \tag{100}$$

Now applying Lemma 12.10,

$$\left\| \left( I - \gamma \hat{\mathsf{P}}_w^{\pi,V} \right)^{-1} \left( \hat{\mathsf{P}}_w^{\pi,V} V_{\gamma,\mathcal{P}}^{\pi} - \mathsf{P}_w^{\pi,V} V_{\gamma,\mathcal{P}}^{\pi} \right) \right\|_\infty \leq \frac{C_3 \log(\frac{18SAN}{\delta})}{N(1-\gamma)^2} + \sqrt{\frac{2C_4 \log(\frac{18SAN}{\delta})\mathcal{H}^2}{(1-\gamma)^2 N}}. \tag{101}$$

We hence obtain the bound on $\|\tilde{V}_{\gamma,\hat{\mathcal{P}}}^{\pi^*} - V_{\gamma,\mathcal{P}}^{\pi^*}\|$ as follows:

$$\|\tilde{V}_{\gamma,\hat{\mathcal{P}}}^{\pi^*} - V_{\gamma,\mathcal{P}}^{\pi^*}\| \leq \frac{a_1 \log(\frac{18SAN}{\delta})}{N(1-\gamma)^2} + \sqrt{\frac{a_2 \log(\frac{18SAN}{\delta})\mathcal{H}^2}{(1-\gamma)^2 N}} + \frac{\epsilon}{6}, \tag{102}$$

when $N \geq \frac{C \log(\frac{18SAN}{\delta})}{(1-\gamma)^2}$.

**12.2. Part B:** $\|\tilde{V}_{\gamma,\hat{\mathcal{P}}}^{\hat{\pi}} - V_{\gamma,\mathcal{P}}^{\hat{\pi}}\|$

Similarly, we have that

$$\|\tilde{V}_{\gamma,\hat{\mathcal{P}}}^{\hat{\pi}} - V_{\gamma,\mathcal{P}}^{\hat{\pi}}\| \leq \|\tilde{V}_{\gamma,\hat{\mathcal{P}}}^{\hat{\pi}} - \tilde{V}_{\gamma,\mathcal{P}}^{\hat{\pi}}\| + \|\tilde{V}_{\gamma,\mathcal{P}}^{\hat{\pi}} - V_{\gamma,\mathcal{P}}^{\hat{\pi}}\|$$
$$\leq \frac{\epsilon}{6} + \|\tilde{V}_{\gamma,\hat{\mathcal{P}}}^{\hat{\pi}} - \tilde{V}_{\gamma,\mathcal{P}}^{\hat{\pi}}\|, \tag{103}$$

hence it suffices to bound the term $\|\tilde{V}_{\gamma,\hat{\mathcal{P}}}^{\hat{\pi}} - \tilde{V}_{\gamma,\mathcal{P}}^{\hat{\pi}}\|$. By setting $\pi = \hat{\pi}$ in (79), we have that

$$\|\tilde{V}_{\gamma,\hat{\mathcal{P}}}^{\hat{\pi}} - \tilde{V}_{\gamma,\mathcal{P}}^{\hat{\pi}}\|_\infty \leq \gamma \max \left\{ \left\| \left( I - \gamma \mathsf{P}_w^{\hat{\pi},\tilde{\hat{V}}} \right)^{-1} \left( \hat{\mathsf{P}}_w^{\hat{\pi},\tilde{\hat{V}}} \tilde{V}_{\gamma,\hat{\mathcal{P}}}^{\hat{\pi}} - \mathsf{P}_w^{\hat{\pi},\tilde{\hat{V}}} \tilde{V}_{\gamma,\hat{\mathcal{P}}}^{\hat{\pi}} \right) \right\|_\infty, \right.$$
$$\left. \left\| \left( I - \gamma \mathsf{P}_w^{\hat{\pi},\tilde{V}} \right)^{-1} \left( \hat{\mathsf{P}}_w^{\hat{\pi},\tilde{V}} \tilde{V}_{\gamma,\hat{\mathcal{P}}}^{\hat{\pi}} - \mathsf{P}_w^{\hat{\pi},\tilde{V}} \tilde{V}_{\gamma,\hat{\mathcal{P}}}^{\hat{\pi}} \right) \right\|_\infty \right\}. \tag{104}$$

We first bound the first term $\left\| \left( I - \gamma \mathsf{P}_w^{\hat{\pi},\tilde{\hat{V}}} \right)^{-1} \left( \hat{\mathsf{P}}_w^{\hat{\pi},\tilde{\hat{V}}} \tilde{V}_{\gamma,\hat{\mathcal{P}}}^{\hat{\pi}} - \mathsf{P}_w^{\hat{\pi},\tilde{\hat{V}}} \tilde{V}_{\gamma,\hat{\mathcal{P}}}^{\hat{\pi}} \right) \right\|_\infty$. To simplify notation, we rewrite $\mathsf{P}_w^{\hat{\pi},\tilde{\hat{V}}}$ as $\mathsf{P}_w^{\hat{\pi},\hat{V}}$ and $\hat{\mathsf{P}}_w^{\hat{\pi},\tilde{\hat{V}}}$ by $\hat{\mathsf{P}}_w^{\hat{\pi},\hat{V}}$.

We first introduce the following separation events:

$$\hat{\Omega}_\omega \triangleq \{\tilde{V}_{\gamma,\hat{\mathcal{P}}}^*(s) - \max_{a \neq \hat{\pi}^*(s)} \tilde{Q}_{\gamma,\hat{\mathcal{P}}}^*(s,a) \geq \omega, \forall s \in \mathcal{S}\}, \tag{105}$$

$$\Omega_\omega \triangleq \{\tilde{V}_{\gamma,\mathcal{P}}^*(s) - \max_{a \neq \pi^*(s)} \tilde{Q}_{\gamma,\mathcal{P}}^*(s,a) \geq \omega, \forall s \in \mathcal{S}\}. \tag{106}$$

These events indicate that there exists some threshold between the value functions of the optimal action and other actions, and there is no tie between the optimal robust value functions. It further implies that the optimal policy $\hat{\pi}^*$ and $\pi^*$ are unique. As we shall show in Lemma 12.3, with a carefully chosen threshold, such events will occur with high probability.

**Lemma 12.3.** *Set $\omega = \frac{\xi\delta(1-\gamma)}{3SA^2}$, then both (105) and (106) occur with probability at least $1 - \delta$.*

We then combine Lemma 14 from (Shi et al., 2023) and Lemma 9 in (Li et al., 2020) together to show the following result. Such a result allows us to decouple the dependence between $\hat{\pi}$ and other terms.

**Lemma 12.4.** *Consider any $\delta \in (0,1)$. Taking $N \geq \mathcal{O}\left(\frac{\log\left(\frac{54SAN^2}{(1-\gamma)\delta}\right)}{1-\gamma}\right)$, with probability at least $1 - 2\delta$, events (105) and (106) occur, and it holds that*

$$\left|\hat{\mathsf{P}}_w^{\hat{\pi},\hat{V}}\tilde{V}_{\gamma,\hat{\mathcal{P}}}^{\hat{\pi}} - \mathsf{P}_w^{\hat{\pi},\hat{V}}\tilde{V}_{\gamma,\hat{\mathcal{P}}}^{\hat{\pi}}\right| \leq 2\sqrt{\frac{\log(\frac{54SAN^2}{(1-\gamma)\delta})}{N}}\sqrt{\mathrm{Var}_{\mathsf{P}_{s,a}}(\tilde{V}_{\gamma,\hat{\mathcal{P}}}^{\hat{\pi}})}\mathbf{1} + \frac{8\log(\frac{54SAN^2}{(1-\gamma)\delta})}{N(1-\gamma)}\mathbf{1}. \tag{107}$$

With Lemma 12.4 in hand, we have

$$\left(I - \gamma\mathsf{P}_w^{\hat{\pi},\hat{V}}\right)^{-1}\left(\hat{\mathsf{P}}_w^{\hat{\pi},\hat{V}}\tilde{V}_{\gamma,\hat{\mathcal{P}}}^{\hat{\pi}} - \mathsf{P}_w^{\hat{\pi},\hat{V}}\tilde{V}_{\gamma,\hat{\mathcal{P}}}^{\hat{\pi}}\right)$$

$$\overset{(i)}{\leq} \left(I - \gamma\mathsf{P}_w^{\hat{\pi},\hat{V}}\right)^{-1}\left|\hat{\mathsf{P}}_w^{\hat{\pi},\hat{V}}\tilde{V}_{\gamma,\hat{\mathcal{P}}}^{\hat{\pi}} - \mathsf{P}_w^{\hat{\pi},\hat{V}}\tilde{V}_{\gamma,\hat{\mathcal{P}}}^{\hat{\pi}}\right|$$

$$\leq 2\sqrt{\frac{\log(\frac{54SAN^2}{(1-\gamma)\delta})}{N}}\left(I - \gamma\mathsf{P}_w^{\hat{\pi},\hat{V}}\right)^{-1}\sqrt{\mathrm{Var}_{P^{\hat{\pi}}}(\tilde{V}_{\gamma,\hat{\mathcal{P}}}^{\hat{\pi}})} + \left(\frac{8\log(\frac{54SAN^2}{(1-\gamma)\delta})}{N(1-\gamma)^2}\right)\mathbf{1}$$

$$\overset{(ii)}{\leq} \left(\frac{8\log(\frac{54SAN^2}{(1-\gamma)\delta})}{N(1-\gamma)^2}\right)\mathbf{1} + \underbrace{2\sqrt{\frac{\log(\frac{54SAN^2}{(1-\gamma)\delta})}{N}}\left(I - \gamma\mathsf{P}_w^{\hat{\pi},\hat{V}}\right)^{-1}\sqrt{\mathrm{Var}_{\mathsf{P}_w^{\hat{\pi},\hat{v}}}(\tilde{V}_{\gamma,\hat{\mathcal{P}}}^{\hat{\pi}})}}_{=:B_1}$$

$$+ \underbrace{2\sqrt{\frac{\log(\frac{54SAN^2}{(1-\gamma)\delta})}{N}}\left(I - \gamma\mathsf{P}_w^{\hat{\pi},\hat{V}}\right)^{-1}\sqrt{\left|\mathrm{Var}_{P^{\hat{\pi}}}(\tilde{V}_{\gamma,\hat{\mathcal{P}}}^{\hat{\pi}}) - \mathrm{Var}_{\mathsf{P}_w^{\hat{\pi},\hat{v}}}(\tilde{V}_{\gamma,\hat{\mathcal{P}}}^{\hat{\pi}})\right|}}_{=:B_2}, \tag{108}$$

where (i) and (ii) hold by the fact that each row of $(1-\gamma)\left(I - \gamma\mathsf{P}_w^{\hat{\pi},\hat{V}}\right)^{-1}$ is a probability vector that falls into $\Delta(\mathcal{S})$.

**Term $B1$.** Similar to term $A1$, term $B1$ is equivalent to $2\sqrt{\frac{\log(\frac{54SAN^2}{(1-\gamma)\delta})}{N}}(I-\gamma\mathsf{P})^{-1}\sqrt{\mathrm{Var}_{\mathsf{P}}(V_{\gamma,\mathsf{P}})}$ with $\mathsf{P} = \mathsf{P}_w^{\hat{\pi},\hat{V}}$. Specifically, $\hat{\pi}$ can be viewed as the optimal policy for $\gamma$ and the empirical uncertainty set. Thus, applying Corollary 11.3 implies that

$$\left\|\tilde{V}_{\gamma,\hat{\mathcal{P}}}^{\hat{\pi}} - \frac{\tilde{g}_{\hat{\mathcal{P}}}^*}{1-\gamma}\right\| \leq \mathcal{H}, \tag{109}$$

and hence

$$B1 \leq \sqrt{\frac{2d_1\log(\frac{18SAN}{\delta})\mathcal{H}^2}{(1-\gamma)^2N}}. \tag{110}$$

**Term $B2$.** Similar to term $A3$, term $B2$ can be bounded by noting that $R \leq \frac{1}{\mathcal{H}}$

$$B2 \leq 2\sqrt{\frac{\log(\frac{18SAN}{\delta})\mathcal{H}^2}{N(1-\gamma)^2}}. \tag{111}$$

Combine both bounds together, and we have that when $N \geq \frac{C\log\frac{SAN}{\delta}}{(1-\gamma)^2}$,

$$\left(I - \gamma\mathsf{P}_w^{\hat{\pi},\hat{V}}\right)^{-1}\left(\hat{\mathsf{P}}_w^{\hat{\pi},\hat{V}}\tilde{V}_{\gamma,\hat{\mathcal{P}}}^{\hat{\pi}} - \mathsf{P}_w^{\hat{\pi},\hat{V}}\tilde{V}_{\gamma,\hat{\mathcal{P}}}^{\hat{\pi}}\right) \leq \frac{D_1\log(\frac{18SAN}{\delta})}{N(1-\gamma)^2} + \sqrt{\frac{2D_2\log(\frac{18SAN}{\delta})\mathcal{H}^2}{(1-\gamma)^2N}} \tag{112}$$

with probability at least $1 - \delta$. Similarly, we can get the bound on the second term of (104), which finally implies that

$$\left\| \tilde{V}_{\gamma,\hat{\mathcal{P}}}^{\hat{\pi}} - \tilde{V}_{\gamma,\mathcal{P}}^{\hat{\pi}} \right\|_\infty \leq \frac{D_1 \log(\frac{18SAN}{\delta})}{N(1-\gamma)^2} + \sqrt{\frac{2D_2 \log(\frac{18SAN}{\delta})\mathcal{H}^2}{(1-\gamma)^2 N}}. \tag{113}$$

### 12.3. Summing Up the Results

Combine the bounds obtained from both Part A and Part B, it holds that with probability at least $1 - 4\delta$,

$$V_{\gamma,\mathcal{P}}^* - V_{\gamma,\mathcal{P}}^{\hat{\pi}} \leq m_1 \epsilon + m_2 \frac{\log(\frac{18SAN}{\delta})}{N(1-\gamma)^2} + m_3 \sqrt{\frac{\log(\frac{18SAN}{\delta})\mathcal{H}^2}{(1-\gamma)^2 N}}, \tag{114}$$

when $N \geq \frac{C \log(\frac{18SAN}{\delta})}{(1-\gamma)^2}$.

Thus, to achieve an $\epsilon$-optimal policy, it requires a total number of samples of

$$NSA = \frac{CSA \log(\frac{18SAN}{\delta})\mathcal{H}^2}{(1-\gamma)^2 \epsilon^2} + \frac{CSA \log(\frac{18SAN}{\delta})}{(1-\gamma)^2}, \tag{115}$$

for some constant $C$. Then setting $\epsilon = \mathcal{H}$ and $1 - \gamma = \frac{\epsilon}{\mathcal{H}}$ implies that

$$NSA \geq \frac{CSA\mathcal{H}^2 \log \frac{SAN}{\delta}}{\epsilon^2} \tag{116}$$

samples are required to find an $\epsilon$-optimal policy for robust average reward.

### 12.4. Proofs of Lemmas

**Lemma 12.5.** *(Lemma 6 of (Zurek & Chen, 2023)) For any deterministic stationary policy $\pi$, we have*

$$\gamma \left\| (I - \gamma \mathsf{P}^\pi)^{-1} \sqrt{\mathbf{Var}_{\mathsf{P}^\pi}\left[ V_{\gamma,\mathsf{P}}^\pi \right]} \right\|_\infty \leq \sqrt{\frac{2}{1-\gamma}} \sqrt{\left\| \mathbf{Var}_{\mathsf{P}^\pi}\left[ \sum_{t=0}^\infty \gamma^t R_t \right] \right\|_\infty} \tag{117}$$

*Proof.* The following variance Bellman equation holds from (Sobel, 1982):

$$\mathbf{Var}_{\mathsf{P}^\pi}\left[ \sum_{t=0}^\infty \gamma^t R_t \right] = \gamma^2 \mathbf{Var}_{\mathsf{P}^\pi}\left[ V_{\gamma,\mathsf{P}}^\pi \right] + \gamma^2 \mathsf{P}^\pi \mathbf{Var}_{\mathsf{P}^\pi}\left[ \sum_{t=0}^\infty \gamma^t R_t \right]. \tag{118}$$

On the other hand, it holds that

$$\left| (1-\gamma)e_s^\top (I - \gamma \mathsf{P}^\pi)^{-1} \sqrt{\mathbf{Var}_{\mathsf{P}^\pi}\left[ V_{\gamma,\mathsf{P}}^\pi \right]} \right| \leq \sqrt{\left| (1-\gamma)e_s^\top (I - \gamma \mathsf{P}^\pi)^{-1} \mathbf{Var}_{\mathsf{P}^\pi}\left[ V_{\gamma,\mathsf{P}}^\pi \right] \right|}.$$

Denote that $v = \mathbf{Var}_{\mathsf{P}^\pi}\left[ V_{\gamma,\mathsf{P}}^\pi \right]$, we then have that

$$\gamma \left\| (I - \gamma \mathsf{P}^\pi)^{-1} \sqrt{v} \right\|_\infty = \gamma \frac{1}{1-\gamma} \left\| (1-\gamma)(I - \gamma \mathsf{P}^\pi)^{-1} \sqrt{v} \right\|_\infty \tag{119}$$

$$\leq \gamma \frac{1}{1-\gamma} \sqrt{\left\| (1-\gamma)(I - \gamma \mathsf{P}^\pi)^{-1} v \right\|_\infty} \tag{120}$$

$$= \gamma \frac{1}{\sqrt{1-\gamma}} \sqrt{\left\| (I - \gamma \mathsf{P}^\pi)^{-1} v \right\|_\infty}. \tag{121}$$

Moreover,

$$\left\|(I - \gamma\mathsf{P}^\pi)^{-1}v\right\|_\infty = \left\|(I - \gamma\mathsf{P}^\pi)^{-1}(I - \gamma^2\mathsf{P}^\pi)(I - \gamma^2\mathsf{P}^\pi)^{-1}v\right\|_\infty \tag{122}$$

$$= \left\|(I - \gamma\mathsf{P}^\pi)^{-1}\left((1-\gamma)I + \gamma(I - \gamma\mathsf{P}^\pi)\right)(I - \gamma^2\mathsf{P}^\pi)^{-1}v\right\|_\infty \tag{123}$$

$$= \left\|\left((1-\gamma)(I - \gamma\mathsf{P}^\pi)^{-1} + \gamma I\right)(I - \gamma^2\mathsf{P}^\pi)^{-1}v\right\|_\infty \tag{124}$$

$$\leq \left\|(1-\gamma)(I - \gamma\mathsf{P}^\pi)^{-1}(I - \gamma^2\mathsf{P}^\pi)^{-1}v\right\|_\infty + \gamma\left\|(I - \gamma^2\mathsf{P}^\pi)^{-1}v\right\|_\infty \tag{125}$$

$$\leq (1-\gamma)\left\|(I - \gamma\mathsf{P}^\pi)^{-1}\right\|_{\infty\to\infty}\left\|(I - \gamma^2\mathsf{P}^\pi)^{-1}v\right\|_\infty + \gamma\left\|(I - \gamma^2\mathsf{P}^\pi)^{-1}v\right\|_\infty \tag{126}$$

$$\leq (1+\gamma)\left\|(I - \gamma^2\mathsf{P}^\pi)^{-1}v\right\|_\infty \tag{127}$$

$$\leq 2\left\|(I - \gamma^2\mathsf{P}^\pi)^{-1}v\right\|_\infty. \tag{128}$$

Combining them with the variance Bellman equation (118), it holds that

$$\gamma\left\|(I - \gamma\mathsf{P}^\pi)^{-1}\sqrt{v}\right\|_\infty \leq \gamma\frac{1}{\sqrt{1-\gamma}}\sqrt{2\left\|(I - \gamma^2\mathsf{P}^\pi)^{-1}v\right\|_\infty} \leq \sqrt{\frac{2}{1-\gamma}}\sqrt{\left\|\mathbf{Var}_{\mathsf{P}^\pi}\left[\sum_{t=0}^\infty \gamma^t R_t\right]\right\|_\infty}. \tag{129}$$

$\square$

**Lemma 12.6.** *(Lemma 7 of (Zurek & Chen, 2023))* *For any integer $T \geq 1$, for any deterministic stationary policy $\pi$, we have*

$$\left\|\mathbf{Var}_{\mathsf{P}^\pi}\left[\sum_{t=0}^\infty \gamma^t R_t\right]\right\|_\infty \leq \frac{\left\|\mathbf{Var}_{\mathsf{P}^\pi}\left[\sum_{t=0}^{T-1} \gamma^t R_t + \gamma^T V_\gamma^\pi(S_T)\right]\right\|_\infty}{1 - \gamma^{2T}}.$$

**Lemma 12.7.** *(Lemma 8 of (Zurek & Chen, 2023))* *If $\gamma \geq 1 - \frac{1}{\mathcal{H}}$ for some integer $\mathcal{H} \geq 1$, then*

$$\frac{1 - \gamma^{2\mathcal{H}}}{1 - \gamma} \geq \left(1 - \frac{1}{e^2}\right)\mathcal{H} \geq \frac{4}{5}\mathcal{H}.$$

**Lemma 12.8.** *Letting $\pi^*$ be the optimal policy for the robust DMDP $(\mathcal{S}, \mathcal{A}, \gamma, r, \mathcal{P})$, we have*

$$\left\|\mathbf{Var}_{\mathsf{P}^{\pi^*}}\left[\sum_{t=0}^\infty \gamma^t R_t\right]\right\|_\infty \leq 5\frac{\mathcal{H}}{1-\gamma}.$$

*Proof.* By using Lemma 12.6, it suffices to bound $\left\|\mathbf{Var}_{\mathsf{P}^{\pi^*}}\left[\sum_{t=0}^{\mathcal{H}-1} \gamma^t R_t + \gamma^{\mathcal{H}} V_{\gamma,\mathsf{P}}^{\pi^*}(S_\mathcal{H})\right]\right\|_\infty$.

Fixing a state $s_0 \in \mathcal{S}$,

$$\mathbf{Var}_{\mathsf{P}_{s_0}^{\pi^*}}\left[\sum_{t=0}^{\mathcal{H}-1} \gamma^t R_t + \gamma^{\mathcal{H}} V_{\gamma,\mathsf{P}}^{\pi^*}(S_\mathcal{H})\right] = \mathbf{Var}_{\mathsf{P}_{s_0}^{\pi^*}}\left[\sum_{t=0}^{\mathcal{H}-1} \gamma^t R_t + \gamma^{\mathcal{H}}\left(V_{\gamma,\mathsf{P}}^{\pi^*}(S_\mathcal{H}) - \frac{1}{1-\gamma}g_\mathsf{P}^{\pi^*}\right)\right]$$

$$\leq \mathbb{E}_{\mathsf{P}_{s_0}^{\pi^*}}\left|\sum_{t=0}^{\mathcal{H}-1} \gamma^t R_t + \gamma^{\mathcal{H}}\left(V_{\gamma,\mathsf{P}}^{\pi^*}(S_\mathcal{H}) - \frac{1}{1-\gamma}g_\mathsf{P}^{\pi^*}\right)\right|^2$$

$$\leq 2\mathbb{E}_{\mathsf{P}_{s_0}^{\pi^*}}\left|\sum_{t=0}^{\mathcal{H}-1} \gamma^t R_t\right|^2 + 2\mathbb{E}_{\mathsf{P}_{s_0}^{\pi^*}}\left|\gamma^{\mathcal{H}}\left(V_{\gamma,\mathsf{P}}^{\pi^*}(S_\mathcal{H}) - \frac{1}{1-\gamma}g_\mathsf{P}^{\pi^*}\right)\right|^2$$

$$\leq 2\mathcal{H}^2 + 2\sup_s\left(V_{\gamma,\mathsf{P}}^{\pi^*}(s) - \frac{1}{1-\gamma}g_\mathsf{P}^{\pi^*}\right)^2$$

$$\leq 4\mathcal{H}^2,$$

where the last inequality can be similarly derived as Lemma 11.1. We thus have that

$$\left\| \mathbf{Var}_{\mathsf{P}^{\pi^*}} \left[ \sum_{t=0}^{\infty} \gamma^t R_t \right] \right\|_{\infty} \leq \frac{4\mathcal{H}^2}{1 - \gamma^{2H}}.$$

Together with Lemma 12.7, this completes the proof. □

**Lemma 12.9.** *(Lemma 12.2) For any policy $\pi$, if $N \geq \mathcal{O}\left( \frac{\log \frac{SA}{(1-\gamma)\delta\epsilon}}{1-\gamma} \right)$, it holds with probability at least $1 - \delta$ that*

$$\left\| \left( I - \gamma \mathsf{Q}^{\pi^*} \right)^{-1} \sqrt{\mathrm{Var}_{\mathsf{Q}^{\pi^*}}(V_{\gamma,\mathsf{Q}}^{\pi^*})} \right\|_{\infty} \leq \sqrt{\frac{c_1 \mathcal{H}}{(1-\gamma)^2}}. \tag{130}$$

*Proof.* We prove a more general result: for any kernel $\mathsf{P}$, it holds that

$$\left\| \left( I - \gamma \mathsf{P}^{\pi^*} \right)^{-1} \sqrt{\mathrm{Var}_{\mathsf{P}^{\pi^*}}(V_{\gamma,\mathsf{P}}^{\pi^*})} \right\|_{\infty} \leq \sqrt{\frac{c_1 \mathcal{H}}{(1-\gamma)^2}}. \tag{131}$$

By Lemma 12.5, we have that

$$\left\| \left( I - \gamma \mathsf{P}^{\pi^*} \right)^{-1} \sqrt{\mathrm{Var}_{\mathsf{P}^{\pi^*}}(V_{\gamma,\mathsf{P}}^{\pi^*})} \right\|_{\infty}$$

$$\leq \sqrt{\frac{2}{1-\gamma}} \sqrt{\left\| \mathbf{Var}_{\mathsf{P}^{\pi^*}} \left[ \sum_{t=0}^{\infty} \gamma^t R_t \right] \right\|_{\infty}}$$

$$\overset{(a)}{\leq} \sqrt{\frac{2}{1-\gamma}} \sqrt{5 \frac{\mathcal{H}}{1-\gamma}}$$

$$= \sqrt{\frac{10\mathcal{H}}{(1-\gamma)^2}}, \tag{132}$$

where $(a)$ is due to Lemma 12.8. □

**Lemma 12.10.** *For any transition kernels $q_1$ and $q_2$, it holds that*

$$\left\| \left( I - \gamma q_1^{\pi^*} \right)^{-1} \sqrt{\mathrm{Var}_{q_2^{\pi^*}}(V_{\gamma,\mathcal{P}}^{\pi^*})} \right\|_{\infty} \leq \sqrt{\frac{c_1 \mathcal{H}^2}{(1-\gamma)^2}}. \tag{133}$$

*Proof.* Note that $\mathbf{Var}_q(V) = \mathbf{Var}_q(V - ke)$ for any $k$ and $e = (1, ..., 1)$. Moreover, from Corollary 11.5, it holds that

$$\left\| V_{\gamma,\mathcal{P}}^{\pi^*} - \frac{1}{1-\gamma} g_{\mathcal{P}}^{\pi^*} \right\| \leq 4\mathcal{H}. \tag{134}$$

Thus

$$\mathrm{Var}_{q_2^{\pi^*}}(V_{\gamma,\mathcal{P}}^{\pi^*}) = \mathrm{Var}_{q_2^{\pi^*}}\left(V_{\gamma,\mathcal{P}}^{\pi^*} - \frac{1}{1-\gamma} g_{\mathcal{P}}^{\pi^*}\right) \leq \left\| V_{\gamma,\mathcal{P}}^{\pi^*} - \frac{1}{1-\gamma} g_{\mathcal{P}}^{\pi^*} \right\|^2 \leq 16\mathcal{H}^2. \tag{135}$$

The proof is then completed. □

**Lemma 12.11.** *(Lemma 12.3) Set $\omega = \frac{\xi\delta(1-\gamma)}{3SA^2}$, then both (105) and (106) occur with probability at least $1 - \delta$.*

*Proof.* The proof is similar for both events, hence we only present the proof for (106). We show a more general result, namely, with probability at least $1 - \delta$, for any $s$ and $a_1 \neq a_2$,

$$\left| Q_{\gamma,\mathcal{P}}^*(s, a_1) - Q_{\gamma,\mathcal{P}}^*(s, a_2) \right| > \frac{\xi\delta(1-\gamma)}{3SA^2}. \tag{136}$$

We further introduce the following notation:

$$r_\tau(s, a_1) = \tau, \tag{137}$$
$$r_\tau(s', a') = \tilde{r}(s', a'), \forall (s', a') \neq (s, a_1). \tag{138}$$

We denote the optimal robust value functions and the optimal policy w.r.t. $r_\tau$ as $Q_\tau^*, V_\tau^*$ and $\pi_\tau^*$.

We first prove the following claim: there exists some $\tau'$, such that

$$\pi_\tau^*(s) \neq a_1, \text{ for all } \tau < \tau', \tag{139}$$
$$\pi_\tau^*(s) = a_1, \text{ for all } \tau > \tau'. \tag{140}$$

Define

$$\tau' = \sup\{u : \pi_\tau^*(s) \neq a_1, \forall \tau < u\}, \tag{141}$$

then it suffices to show (140) for our choice, which exactly follows as the proofs of eq (95) in (Li et al., 2020). We then prove the lemma as follows.

First, define the following sets:

$$I_{0,\omega} \triangleq \{\tau : |Q_\tau^*(s, a_1) - Q_\tau^*(s, a_2)| < \omega\}, \tag{142}$$
$$I_{1,\omega} \triangleq \{\tau : \tau < \tau', |Q_\tau^*(s, a_1) - Q_\tau^*(s, a_2)| < \omega\}, \tag{143}$$
$$I_{2,\omega} \triangleq \{\tau : \tau \geq \tau', |Q_\tau^*(s, a_1) - Q_\tau^*(s, a_2)| < \omega\}. \tag{144}$$

Clearly, $I_{0,\omega} = I_{1,\omega} \cup I_{2,\omega}$, and we will show that the probability of these events is small.

**Step 1.** For $\tau \in I_{1,\omega}$, note that $V_\tau^*$ does not depend on $\tau$, since $\pi_\tau^*(s) \neq a_1$, and $\tau$ is never active when calculating $V_\tau^*$. Thus, the robust Bellman equation becomes

$$Q_\tau^*(s, a_1) = \tau + \gamma \sigma_{\mathcal{P}_s^{a_1}}(V_\tau^*),$$
$$Q^*(s, a_2) = \tilde{r}(s, a_2) + \gamma \sigma_{\mathcal{P}_s^{a_2}}(V_\tau^*). \tag{145}$$

Thus, it holds that

$$I_{1,\omega} \subset \{\tau : |\tau + \gamma \sigma_{\mathcal{P}_s^{a_1}}(V_\tau^*) - Q^*(s, a_2)| < \omega\}. \tag{146}$$

Since both terms $\gamma \sigma_{\mathcal{P}_s^{a_1}}(V_\tau^*)$ and $Q^*(s, a_2)$ are independent from $\tau$, the Lebesgue measure of $I_{1,\omega}$ is at most $2\omega$.

**Step 2.** We now consider $I_{2,\omega}$. First, note that

$$0 \leq Q_{\tau_2}^* - Q_{\tau_1}^* \leq r_{\tau_2} - r_{\tau_1} + \gamma(\sigma(V_{\tau_2}^*) - \sigma(V_{\tau_1}^*))$$
$$\leq r_{\tau_2} - r_{\tau_1} + \gamma \|V_{\tau_2}^* - V_{\tau_1}^*\|, \tag{147}$$

for any $\tau_2 > \tau_1 > \tau'$, which is from the 1-Lipschitz of the support functions. Moreover, for any $(x, b) \neq (s, a_1)$, since $r_{\tau_2}(x, b) = r_{\tau_1}(x, b)$, it holds that

$$0 \leq Q_{\tau_2}^*(x, b) - Q_{\tau_1}^*(x, b) \leq \gamma \|V_{\tau_2}^* - V_{\tau_1}^*\|. \tag{148}$$

On the other hand, note that

$$0 \leq V_{\tau_2}^* - V_{\tau_1}^* = \max_a Q_{\tau_2}^* - \max_a Q_{\tau_1}^* \leq \|Q_{\tau_2}^* - Q_{\tau_1}^*\|, \tag{149}$$

and thus

$$\|V_{\tau_2}^* - V_{\tau_1}^*\| \leq \|Q_{\tau_2}^* - Q_{\tau_1}^*\|. \tag{150}$$

Note that (148) implies that

$$Q^*_{\tau_2}(x,b) - Q^*_{\tau_1}(x,b) \le \gamma\|V^*_{\tau_2} - V^*_{\tau_1}\| < \|V^*_{\tau_2} - V^*_{\tau_1}\|, \forall(x,b) \ne (s,a_1), \tag{151}$$

thus

$$\|Q^*_{\tau_2} - Q^*_{\tau_1}\| = |Q^*_{\tau_2}(s,a_1) - Q^*_{\tau_1}(s,a_1)| \ge \|V^*_{\tau_2} - V^*_{\tau_1}\|. \tag{152}$$

Since $\tau_1, \tau_2 \ge \tau'$, $V^*_{\tau_2}(s) = Q^*\tau_2(s,a_1)$ and $V^*_{\tau_1}(s) = Q^*\tau_1(s,a_1)$, and we further have that

$$V^*_{\tau_2}(s) - V^*_{\tau_1}(s) = Q^*_{\tau_2}(s,a_1) - Q^*_{\tau_1}(s,a_1) \ge \|V^*_{\tau_2} - V^*_{\tau_1}\|, \tag{153}$$

and hence

$$\|V^*_{\tau_2} - V^*_{\tau_1}\| = Q^*_{\tau_2}(s,a_1) - Q^*_{\tau_1}(s,a_1). \tag{154}$$

Now from the robust Bellman equation, it holds that

$$\begin{aligned}
&Q^*_{\tau_2}(s,a_1) - Q^*_{\tau_1}(s,a_1)\\
&= \|V^*_{\tau_2} - V^*_{\tau_1}\|\\
&= \|r_{\tau_2} - r_{\tau_1} + \gamma(\sigma(V^*_{\tau_2}) - \sigma(V^*_{\tau_1}))\|\\
&\ge r_{\tau_2} - r_{\tau_1},
\end{aligned} \tag{155}$$

due to the monotonicity properties of the support functions.

We note that (148) and (155) exactly match eqs. (99) and (102) in (Li et al., 2020), and hence the rest of the proof follows similarly. $\qquad\square$

**Lemma 12.12.** *(Lemma 12.4) Consider any $\delta \in (0,1)$. Taking $N \ge \mathcal{O}\left(\frac{\log\left(\frac{54SAN^2}{(1-\gamma)\delta}\right)}{1-\gamma}\right)$, with probability at least $1 - 2\delta$, events* (105) *and* (106) *occur, and it holds that*

$$\left|\hat{\mathsf{P}}^{\hat{\pi},\hat{V}}_w \tilde{V}^{\hat{\pi}}_{\gamma,\hat{\mathcal{P}}} - \mathsf{P}^{\hat{\pi},\hat{V}}_w \tilde{V}^{\hat{\pi}}_{\gamma,\hat{\mathcal{P}}}\right| \le 2\sqrt{\frac{\log(\frac{54SAN^2}{(1-\gamma)\delta})}{N}}\sqrt{\mathrm{Var}_{\mathsf{P}_{s,a}}(\tilde{V}^{\hat{\pi}}_{\gamma,\hat{\mathcal{P}}})}1 + \frac{8\log(\frac{54SAN^2}{(1-\gamma)\delta})}{N(1-\gamma)}1. \tag{156}$$

*Proof.* The proof is obtained similarly as Lemma 14 from (Shi et al., 2023), by only replace $r(s,a)$ therein by $\mathbb{E}[\tilde{r}(s,a)]$.

For any $(s,a)$, by the duality we have that

$$\left|(\hat{\mathsf{P}}^{\hat{\pi},\hat{V}})_{s,a}\tilde{V}^{\hat{\pi}}_{\gamma,\hat{\mathcal{P}}} - (\mathsf{P}^{\hat{\pi},\hat{V}})_{s,a}\tilde{V}^{\hat{\pi}}_{\gamma,\hat{\mathcal{P}}}\right| \le \max_{\alpha\in[\min_s \tilde{V}^{\hat{\pi}}_{\gamma,\hat{\mathcal{P}}}(s),\max_s \tilde{V}^{\hat{\pi}}_{\gamma,\hat{\mathcal{P}}}(s)]} \left|\left(\mathsf{P}_{s,a} - \hat{\mathsf{P}}_{s,a}\right)\left[\tilde{V}^{\hat{\pi}}_{\gamma,\hat{\mathcal{P}}}\right]_\alpha\right|. \tag{157}$$

**Construction of auxiliary RMDPs with deterministic empirical nominal transitions.** Recall that we target the empirical infinite-horizon robust MDP with the nominal transition kernel $\hat{\mathsf{P}}$. We define the nominal transition kernel and reward function as $P^{s,u}$ and $r^{s,u}$, which are expressed as follows

$$\begin{cases} P^{s,u}(s'|s,a) = \mathbf{1}(s' = s) & \text{for all } (s',a) \in \mathcal{S} \times \mathcal{A},\\ P^{s,u}(\cdot|\tilde{s},a) = \hat{\mathsf{P}}(\cdot|\tilde{s},a) & \text{for all } (\tilde{s},a) \in \mathcal{S} \times \mathcal{A} \text{ and } \tilde{s} \ne s, \end{cases} \tag{158}$$

and

$$\begin{cases} r^{s,u}(s,a) = u & \text{for all } a \in \mathcal{A},\\ r^{s,u}(\tilde{s},a) = \mathbb{E}[\tilde{r}(\tilde{s},a)] & \text{for all } (\tilde{s},a) \in \mathcal{S} \times \mathcal{A} \text{ and } \tilde{s} \ne s. \end{cases} \tag{159}$$

Correspondingly, the associated robust Bellman operator is then

$$\forall(\tilde{s},a) \in \mathcal{S} \times \mathcal{A}: \quad \mathbf{T}_{s,u}(Q)(\tilde{s},a) = r^{s,u}(\tilde{s},a) + \gamma\inf_{\mathsf{P}\in\mathcal{P}(P^{s,u}_{\tilde{s},a})}\mathsf{P}V, \quad \text{with } V(\tilde{s}) = \max_a Q(\tilde{s},a). \tag{160}$$

**Fixed-point equivalence.** Recall that $\tilde{Q}^{\hat{\pi}}_{\gamma,\hat{\mathcal{P}}}$ is the unique fixed point of the Bellman operator with the corresponding robust value $\tilde{V}^{\hat{\pi}}_{\gamma,\hat{\mathcal{P}}}$. We assert that the corresponding robust value function $(\tilde{V}^{\hat{\pi}}_{\gamma,\hat{\mathcal{P}}})_{s,u^*}$ obtained from the fixed point of $\mathbf{T}_{s,u}(\cdot)$ aligns with the robust value function $\tilde{V}^{\hat{\pi}}_{\gamma,\hat{\mathcal{P}}}$, as long as we choose $u$ in the following manner:

$$u^* = u^*(s) = \tilde{V}^{\hat{\pi}}_{\gamma,\hat{\mathcal{P}}}(s) - \gamma \inf_{\mathsf{P}\in\mathcal{P}(e_s)} \mathsf{P}\tilde{V}^{\hat{\pi}}_{\gamma,\hat{\mathcal{P}}}. \tag{161}$$

where $e_s$ is the $s$-th standard basis vector in $\mathbb{R}^S$. Towards verifying this, we shall break our arguments in two different cases.

- **For state $s$:** One has for any $a \in \mathcal{A}$:

$$
\begin{aligned}
r^{s,u^*}(s,a) + \gamma \inf_{\mathsf{P}\in\mathcal{P}(P^{s,u^*}_{s,a})} \mathsf{P}\tilde{V}^{\hat{\pi}}_{\gamma,\hat{\mathcal{P}}} &= u^* + \gamma \inf_{\mathsf{P}\in\mathcal{P}(e_s)} \mathsf{P}\tilde{V}^{\hat{\pi}}_{\gamma,\hat{\mathcal{P}}} \\
&= \tilde{V}^{\hat{\pi}}_{\gamma,\hat{\mathcal{P}}}(s) - \gamma \inf_{\mathsf{P}\in\mathcal{P}(e_s)} \mathsf{P}\tilde{V}^{\hat{\pi}}_{\gamma,\hat{\mathcal{P}}} + \gamma \inf_{\mathsf{P}\in\mathcal{P}(e_s)} \mathsf{P}\tilde{V}^{\hat{\pi}}_{\gamma,\hat{\mathcal{P}}} = \tilde{V}^{\hat{\pi}}_{\gamma,\hat{\mathcal{P}}}(s),
\end{aligned} \tag{162}
$$

  where the first equality follows from the definition of $P^{s,u^*}_{s,a}$ in (158), and the second equality follows from plugging in the definition of $u^*$ in (161).

- **For state $s' \neq s$:** It is easily verified that for all $a \in \mathcal{A}$,

$$
\begin{aligned}
r^{s,u^*}(s',a) + \gamma \inf_{\mathsf{P}\in\mathcal{P}(P^{s,u^*}_{s',a})} \mathsf{P}\tilde{V}^{\hat{\pi}}_{\gamma,\hat{\mathcal{P}}} &= r(s',a) + \gamma \inf_{\mathsf{P}\in\mathcal{P}(\hat{\mathsf{P}}_{s',a})} \mathsf{P}\tilde{V}^{\hat{\pi}}_{\gamma,\hat{\mathcal{P}}} \\
&= \mathbf{T}(\tilde{Q}^{\hat{\pi}}_{\gamma,\hat{\mathcal{P}}})(s',a) = \tilde{Q}^{\hat{\pi}}_{\gamma,\hat{\mathcal{P}}}(s',a),
\end{aligned} \tag{163}
$$

  where the first equality follows from the definitions in (159) and (158), and the last line arises from the definition of the robust Bellman operator, and that $\tilde{Q}^{\hat{\pi}}_{\gamma,\hat{\mathcal{P}}}$ is the fixed point of $\mathbf{T}(\cdot)$.

Combining the facts in the above two cases, we establish that there exists a fixed point $(\tilde{Q}^{\hat{\pi}}_{\gamma,\hat{\mathcal{P}}})_{s,u^*}$ of the operator $\mathbf{T}_{s,u^*}(\cdot)$ by taking

$$
\begin{cases}
(\tilde{Q}^{\hat{\pi}}_{\gamma,\hat{\mathcal{P}}})_{s,u^*}(s,a) = \tilde{V}^{\hat{\pi}}_{\gamma,\hat{\mathcal{P}}}(s) & \text{for all } a \in \mathcal{A}, \\
(\tilde{Q}^{\hat{\pi}}_{\gamma,\hat{\mathcal{P}}})_{s,u^*}(s',a) = \tilde{Q}^{\hat{\pi}}_{\gamma,\hat{\mathcal{P}}}(s',a) & \text{for all } s' \neq s \text{ and } a \in \mathcal{A}.
\end{cases} \tag{164}
$$

Consequently, we confirm the existence of a fixed point of the operator $\mathbf{T}_{s,u^*}(\cdot)$. In addition, its corresponding value function $(\tilde{V}^{\hat{\pi}}_{\gamma,\hat{\mathcal{P}}})_{s,u^*}$ also coincides with $\tilde{V}^{\hat{\pi}}_{\gamma,\hat{\mathcal{P}}}$.

This equivalence exactly matches with Step 1 and Step 2 in Lemma 14 of (Shi et al., 2023), and hence the remaining part directly follows. $\qquad\square$

## 13. Proof of Theorem 4.4 Part 2

The proof of Theorem 4.4 mainly follows a similar structure. We note that it is equivalent to show that

$$\left\|V^{\pi^*}_{\gamma,\mathcal{P}} - V^{\hat{\pi}}_{\gamma,\mathcal{P}}\right\|_\infty \leq 16\sqrt{\frac{2(1+R)\mathcal{H}^2 \log(\frac{36SAN^2}{\delta})}{(1-\gamma)^2 N}}. \tag{165}$$

In order to control the performance gap $\left\|V^{\pi^*}_{\gamma,\mathcal{P}} - V^{\hat{\pi}}_{\gamma,\mathcal{P}}\right\|_\infty$, note that

$$V^{\pi^*}_{\gamma,\mathcal{P}} - V^{\hat{\pi}}_{\gamma,\mathcal{P}} \leq V^{\pi^*}_{\gamma,\mathcal{P}} - V^{\pi^*}_{\gamma,\hat{\mathcal{P}}} + V^{\pi^*}_{\gamma,\hat{\mathcal{P}}} - V^{\hat{\pi}}_{\gamma,\hat{\mathcal{P}}} + V^{\hat{\pi}}_{\gamma,\hat{\mathcal{P}}} - V^{\hat{\pi}}_{\gamma,\mathcal{P}} \leq V^{\pi^*}_{\gamma,\mathcal{P}} - V^{\pi^*}_{\gamma,\hat{\mathcal{P}}} + V^{\hat{\pi}}_{\gamma,\hat{\mathcal{P}}} - V^{\hat{\pi}}_{\gamma,\mathcal{P}}. \tag{166}$$

It is hence sufficient to bound the two terms on the RHS.

**Part A:** $\left\|V_{\gamma,\hat{\mathcal{P}}}^{\pi^*} - V_{\gamma,\mathcal{P}}^{\pi^*}\right\|_\infty$. Towards this, recall the bound in (80):

$$\left\|V_{\gamma,\hat{\mathcal{P}}}^{\pi^*} - V_{\gamma,\mathcal{P}}^{\pi^*}\right\|_\infty \leq \gamma \max \left\{ \left\| \left(I - \gamma\hat{\mathsf{P}}_w^{\pi^\star,\widehat{V}}\right)^{-1} \left(\hat{\mathsf{P}}_w^{\pi^\star,V}V_{\gamma,\mathcal{P}}^{\pi^*} - \mathsf{P}_w^{\pi^\star,V}V_{\gamma,\mathcal{P}}^{\pi^*}\right)\right\|_\infty, \right.$$
$$\left. \left\| \left(I - \gamma\hat{\mathsf{P}}_w^{\pi^\star,V}\right)^{-1} \left(\hat{\mathsf{P}}_w^{\pi^\star,V}V_{\gamma,\mathcal{P}}^{\pi^*} - \mathsf{P}_w^{\pi^\star,V}V_{\gamma,\mathcal{P}}^{\pi^*}\right)\right\|_\infty \right\}. \tag{167}$$

To control the main term $\hat{\mathsf{P}}_w^{\pi^\star,V}V_{\gamma,\mathcal{P}}^{\pi^*} - \mathsf{P}_w^{\pi^\star,V}V_{\gamma,\mathcal{P}}^{\pi^*}$ in (167), we first introduce the following lemma.

**Lemma 13.1.** *For any $\delta \in (0,1)$ and any fixed policy $\pi$, one has with probability at least $1 - \delta$,*

$$\left\|\hat{\mathsf{P}}_w^{\pi,V}V_{\gamma,\mathcal{P}}^{\pi} - \mathsf{P}_w^{\pi,V}V_{\gamma,\mathcal{P}}^{\pi}\right\|_\infty \leq 4\sqrt{\frac{2(1+R)\mathcal{H}^2 \log(\frac{24SAN}{\delta})}{N}}. \tag{168}$$

Applying Lemma 13.1 by taking $\pi = \pi^\star$ gives

$$\left\|\hat{\mathsf{P}}_w^{\pi^\star,V}V_{\gamma,\mathcal{P}}^{\pi^*} - \mathsf{P}_w^{\pi^\star,V}V_{\gamma,\mathcal{P}}^{\pi^*}\right\|_\infty \leq 4\sqrt{\frac{2(1+R)\mathcal{H}^2 \log(\frac{24SAN}{\delta})}{N}}, \tag{169}$$

which directly leads to

$$\left\| \left(I - \gamma\hat{\mathsf{P}}_w^{\pi^\star,\widehat{V}}\right)^{-1} \left(\hat{\mathsf{P}}_w^{\pi^\star,V}V_{\gamma,\mathcal{P}}^{\pi^*} - \mathsf{P}_w^{\pi^\star,V}V_{\gamma,\mathcal{P}}^{\pi^*}\right)\right\|_\infty$$
$$\leq \left\|\hat{\mathsf{P}}_w^{\pi^\star,V}V_{\gamma,\mathcal{P}}^{\pi^*} - \mathsf{P}_w^{\pi^\star,V}V_{\gamma,\mathcal{P}}^{\pi^*}\right\|_\infty \cdot \left\|\left(I - \gamma\hat{\mathsf{P}}_w^{\pi^\star,\widehat{V}}\right)^{-1}1\right\|_\infty \leq 4\sqrt{\frac{2(1+R)\mathcal{H}^2 \log(\frac{24SAN}{\delta})}{(1-\gamma)^2 N}}. \tag{170}$$

Similarly, we have

$$\left\| \left(I - \gamma\hat{\mathsf{P}}_w^{\pi^\star,V}\right)^{-1} \left(\hat{\mathsf{P}}_w^{\pi^\star,V}V_{\gamma,\mathcal{P}}^{\pi^*} - \mathsf{P}_w^{\pi^\star,V}V_{\gamma,\mathcal{P}}^{\pi^*}\right)\right\|_\infty \leq 4\sqrt{\frac{2(1+R)\mathcal{H}^2 \log(\frac{24SAN}{\delta})}{(1-\gamma)^2 N}}. \tag{171}$$

Inserting (170) and (171) back to (167) yields

$$\left\|V_{\gamma,\hat{\mathcal{P}}}^{\pi^*} - V_{\gamma,\mathcal{P}}^{\pi^*}\right\|_\infty \leq 4\sqrt{\frac{2(1+R)\mathcal{H}^2 \log(\frac{24SAN}{\delta})}{(1-\gamma)^2 N}}. \tag{172}$$

**Part B: controlling** $\left\|V_{\gamma,\hat{\mathcal{P}}}^{\hat{\pi}} - V_{\gamma,\mathcal{P}}^{\hat{\pi}}\right\|_\infty$. Similarly, we have that

$$\left\|V_{\gamma,\hat{\mathcal{P}}}^{\hat{\pi}} - V_{\gamma,\mathcal{P}}^{\hat{\pi}}\right\|_\infty \leq \gamma \max \left\{ \left\| \left(I - \gamma\mathsf{P}_w^{\hat{\pi},V}\right)^{-1} \left(\hat{\mathsf{P}}_w^{\hat{\pi},\widehat{V}}V_{\gamma,\hat{\mathcal{P}}}^{\hat{\pi}} - \mathsf{P}_w^{\hat{\pi},\widehat{V}}V_{\gamma,\hat{\mathcal{P}}}^{\hat{\pi}}\right)\right\|_\infty, \right.$$
$$\left. \left\| \left(I - \gamma\mathsf{P}_w^{\hat{\pi},\widehat{V}}\right)^{-1} \left(\hat{\mathsf{P}}_w^{\hat{\pi},\widehat{V}}V_{\gamma,\hat{\mathcal{P}}}^{\hat{\pi}} - \mathsf{P}_w^{\hat{\pi},\widehat{V}}V_{\gamma,\hat{\mathcal{P}}}^{\hat{\pi}}\right)\right\|_\infty \right\}. \tag{173}$$

We introduce the following lemma which controls $\hat{\mathsf{P}}_w^{\hat{\pi},\widehat{V}}V_{\gamma,\hat{\mathcal{P}}}^{\hat{\pi}} - \mathsf{P}_w^{\hat{\pi},\widehat{V}}V_{\gamma,\hat{\mathcal{P}}}^{\hat{\pi}}$ in (173);

**Lemma 13.2.** *With probability at least $1 - \delta$, one has*

$$\left\|\hat{\mathsf{P}}_w^{\hat{\pi},\widehat{V}}V_{\gamma,\hat{\mathcal{P}}}^{\hat{\pi}} - \mathsf{P}_w^{\hat{\pi},\widehat{V}}V_{\gamma,\hat{\mathcal{P}}}^{\hat{\pi}}\right\|_\infty \leq 12\sqrt{\frac{2(1+R)\mathcal{H}^2 \log(\frac{36SAN^2}{\delta})}{N}}. \tag{174}$$

Repeating the arguments from (169) to (172) yields

$$\left\|V_{\gamma,\hat{\mathcal{P}}}^{\hat{\pi}} - V_{\gamma,\mathcal{P}}^{\hat{\pi}}\right\|_\infty \leq 12\sqrt{\frac{2(1+R)\mathcal{H}^2 \log(\frac{36SAN^2}{\delta})}{(1-\gamma)^2 N}}. \tag{175}$$

Finally, combining all bounds together implies that

$$
\begin{aligned}
&\left\| V_{\gamma,\mathcal{P}}^{\pi^*} - V_{\gamma,\mathcal{P}}^{\hat{\pi}} \right\|_\infty \\
&\leq 4\sqrt{\frac{2(1+R)\mathcal{H}^2 \log(\frac{24SAN}{\delta})}{(1-\gamma)^2 N}} + 12\sqrt{\frac{2(1+R)\mathcal{H}^2 \log(\frac{36SAN^2}{\delta})}{(1-\gamma)^2 N}} \\
&\leq 16\sqrt{\frac{2(1+R)\mathcal{H}^2 \log(\frac{36SAN^2}{\delta})}{(1-\gamma)^2 N}}.
\end{aligned}
\tag{176}
$$

### 13.1. Proofs of Lemmas

**Lemma 13.3.** *(Lemma 13.1) For any $\delta \in (0,1)$ and any fixed policy $\pi$, one has with probability at least $1 - \delta$,*

$$
\left\| \hat{\mathsf{P}}_w^{\pi,V} V_{\gamma,\mathcal{P}}^\pi - \mathsf{P}_w^{\pi,V} V_{\gamma,\mathcal{P}}^\pi \right\|_\infty \leq 4\sqrt{\frac{2(1+R)\mathcal{H}^2 \log(\frac{24SAN}{\delta})}{N}}.
\tag{177}
$$

*Proof.* **Step 1: controlling the point-wise concentration.** Consider any fixed policy $\pi$ and the corresponding robust value vector $V \triangleq V_{\gamma,\mathcal{P}}^\pi - \frac{g_{\mathcal{P}}^\pi}{1-\gamma}$ (independent from $\hat{\mathsf{P}}$). We note that $\|V\| \leq \mathcal{H}$ as showed before. By the duality of CS sets (Shi et al., 2023) it holds that

$$
\begin{aligned}
&\left| (\hat{\mathsf{P}}_w^{\pi,V})_{s,a} V_{\gamma,\mathcal{P}}^\pi - (\mathsf{P}_w^{\pi,V})_{s,a} V_{\gamma,\mathcal{P}}^\pi \right| \\
&= \left| \max_{\alpha\in[\min_s V(s),\max_s V(s)]} \left\{ \mathsf{P}_{s,a}[V]_\alpha - \sqrt{R\mathsf{Var}_{\mathsf{P}_{s,a}}([V]_\alpha)} \right\} \right. \\
&\quad \left. - \max_{\alpha\in[\min_s V(s),\max_s V(s)]} \left\{ \hat{\mathsf{P}}_{s,a}[V]_\alpha - \sqrt{R\mathsf{Var}_{\hat{\mathsf{P}}_{s,a}}([V]_\alpha)} \right\} \right| \\
&\leq \max_{\alpha\in[\min_s V(s),\max_s V(s)]} \left| \left( \mathsf{P}_{s,a} - \hat{\mathsf{P}}_{s,a} \right)[V]_\alpha + \sqrt{R\mathsf{Var}_{\hat{\mathsf{P}}_{s,a}}([V]_\alpha)} - \sqrt{R\mathsf{Var}_{\mathsf{P}_{s,a}}([V]_\alpha)} \right| \\
&\leq \max_{\alpha\in[\min_s V(s),\max_s V(s)]} \left| \left( \mathsf{P}_{s,a} - \hat{\mathsf{P}}_{s,a} \right)[V]_\alpha \right| + \\
&\quad + \max_{\alpha\in[\min_s V(s),\max_s V(s)]} \sqrt{R} \left| \sqrt{\mathsf{Var}_{\hat{\mathsf{P}}_{s,a}}([V]_\alpha)} - \sqrt{\mathsf{Var}_{\mathsf{P}_{s,a}}([V]_\alpha)} \right|,
\end{aligned}
\tag{178}
$$

where the first inequality follows by the maximum operator being 1-Lipschitz, and the second inequality follows from the triangle inequality.

The first term in (178) can be directly bounded through an $\epsilon$-net technique and Hoeffding's inequality, which implies that with probability at least $1 - \delta$,

$$
\max_{\alpha\in[\min_s V(s),\max_s V(s)]} \left| \left( \mathsf{P}_{s,a} - \hat{\mathsf{P}}_{s,a} \right)[V]_\alpha \right| \leq 2\sqrt{\frac{\log(\frac{2SAN}{\delta})\mathcal{H}^2}{N}},
\tag{179}
$$

holds for all $(s,a)$.

**Step 2: controlling the second term in** (178). Consider a fixed $\alpha \in [0, \frac{1}{1-\gamma}]$, applying Lemma 6 of (Panaganti & Kalathil, 2021) with $\|[V]_\alpha\|_\infty \leq \mathcal{H}$, we get that

$$
\left| \sqrt{\mathsf{Var}_{\hat{\mathsf{P}}_{s,a}}([V]_\alpha)} - \sqrt{\mathsf{Var}_{\mathsf{P}_{s,a}}([V]_\alpha)} \right| \leq \sqrt{\frac{2\log(\frac{2}{\delta})\mathcal{H}}{N}}
\tag{180}
$$

holds with probability at least $1 - \delta$. We then introduce the following lemma, whose proof can be similarly derived as Lemma 18 of (Shi et al., 2023).

**Lemma 13.4.** *For any $V$ obeying $\|V\|_\infty \leq \mathcal{H}$, the function $J_{s,a}(\alpha, V) := \left| \sqrt{\mathsf{Var}_{\hat{\mathsf{P}}_{s,a}}([V]_\alpha)} - \sqrt{\mathsf{Var}_{\mathsf{P}_{s,a}}([V]_\alpha)} \right|$ w.r.t. $\alpha$ obeys*

$$
|J_{s,a}(\alpha_1, V) - J_{s,a}(\alpha_2, V)| \leq 4\sqrt{|\alpha_1 - \alpha_2|\mathcal{H}}.
$$

We then construct an $\epsilon$-net $\mathcal{N}$ over $[0, \mathcal{H}]$ with size $N_n \leq 3\epsilon \mathcal{H}$ (Vershynin, 2018), so that with probability at least $1 - \frac{\delta}{SA}$, it holds that for any $(s, a)$,

$$
\max_{\alpha \in [\min_s V(s), \max_s V(s)]} \left| \sqrt{\mathsf{Var}_{\hat{\mathsf{P}}_{s,a}}([V]_\alpha)} - \sqrt{\mathsf{Var}_{\mathsf{P}_{s,a}}([V]_\alpha)} \right|
$$

$$
\leq \max_{\alpha \in [0, 1/(1-\gamma)]} \left| \sqrt{\mathsf{Var}_{\hat{\mathsf{P}}_{s,a}}([V]_\alpha)} - \sqrt{\mathsf{Var}_{\mathsf{P}_{s,a}}([V]_\alpha)} \right|
$$

$$
\overset{(i)}{\leq} 4\sqrt{\frac{\epsilon}{1-\gamma}} + \sup_{\alpha \in \mathcal{N}} \left| \sqrt{\mathsf{Var}_{\hat{\mathsf{P}}_{s,a}}([V]_\alpha)} - \sqrt{\mathsf{Var}_{\mathsf{P}_{s,a}}([V]_\alpha)} \right|
$$

$$
\overset{(ii)}{\leq} 4\sqrt{\frac{\epsilon}{1-\gamma}} + \sqrt{\frac{2\mathcal{H}^2 \log(\frac{2SA|N_\epsilon|}{\delta})}{N}}
$$

$$
\overset{(iii)}{\leq} 2\sqrt{\frac{2\mathcal{H}^2 \log(\frac{2SA|N_\epsilon|}{\delta})}{N}}
$$

$$
\leq 2\sqrt{\frac{2\mathcal{H}^2 \log(\frac{24SAN}{\delta})}{N}}, \tag{181}
$$

where (i) holds by the property of the $\epsilon$-net, (ii) follows from (180), (iii) follows from taking $\epsilon = \frac{\mathcal{H} \log(\frac{2SA|\mathcal{N}|}{\delta})}{8N}$. Inserting (179) and (181) back to (178) and taking the union bound over $(s, a)$, with probability at least $1 - \delta$,

$$
\left| (\hat{\mathsf{P}}_w^{\pi, V})_{s,a} V - (\mathsf{P}_w^{\pi, V})_{s,a} V \right| \leq \max_{\alpha \in [\min_s V(s), \max_s V(s)]} \left| \left( \mathsf{P}_{s,a} - \hat{\mathsf{P}}_{s,a} \right) [V]_\alpha \right| +
$$

$$
+ \max_{\alpha \in [\min_s V(s), \max_s V(s)]} \left| \sqrt{R\mathsf{Var}_{\hat{\mathsf{P}}_{s,a}}([V]_\alpha)} - \sqrt{R\mathsf{Var}_{\mathsf{P}_{s,a}}([V]_\alpha)} \right|
$$

$$
\leq 4\sqrt{\frac{2(1+R)\mathcal{H}^2 \log(\frac{24SAN}{\delta})}{N}}.
$$

Finally, we complete the proof by recalling the matrix form as below:

$$
\left\| \hat{\mathsf{P}}_w^{\pi, V} V_{\gamma, \mathcal{P}}^\pi - \mathsf{P}_w^{\pi, V} V_{\gamma, \mathcal{P}}^\pi \right\|_\infty \leq \max_{(s,a)} \left| \hat{\mathsf{P}}_{s,a}^{\pi, V} V - (\mathsf{P}_w^{\pi, V})_{s,a} V \right| \leq 4\sqrt{\frac{2(1+R)\mathcal{H}^2 \log(\frac{24SAN}{\delta})}{N}}.
$$

$\square$

**Lemma 13.5.** *(Lemma 13.2) With probability at least $1 - \delta$, one has*

$$
\left\| \hat{\mathsf{P}}_w^{\hat{\pi}, \hat{V}} V_{\gamma, \hat{\mathcal{P}}}^{\hat{\pi}} - \mathsf{P}_w^{\hat{\pi}, \hat{V}} V_{\gamma, \hat{\mathcal{P}}}^{\hat{\pi}} \right\|_\infty \leq 12\sqrt{\frac{2(1+R)\mathcal{H}^2 \log(\frac{36SAN^2}{\delta})}{N}}. \tag{182}
$$

*Proof.* For any $(s, a)$, following the same arguments of (178) yields

$$
\left| (\hat{\mathsf{P}}_w^{\hat{\pi}, \hat{V}})_{s,a} V_{\gamma, \hat{\mathcal{P}}}^{\hat{\pi}} - (\mathsf{P}_w^{\hat{\pi}, \hat{V}})_{s,a} V_{\gamma, \hat{\mathcal{P}}}^{\hat{\pi}} \right|
$$

$$
= \left| (\hat{\mathsf{P}}_w^{\hat{\pi}, \hat{V}})_{s,a} (V_{\gamma, \hat{\mathcal{P}}}^{\hat{\pi}} - g_{\hat{\mathcal{P}}}^{\hat{\pi}}) - (\mathsf{P}_w^{\hat{\pi}, \hat{V}})_{s,a} (V_{\gamma, \hat{\mathcal{P}}}^{\hat{\pi}} - g_{\hat{\mathcal{P}}}^{\hat{\pi}}) \right|
$$

$$
\leq \max_{\alpha \in \left[\min_s V_{\gamma, \hat{\mathcal{P}}}^{\hat{\pi}}(s) - g_{\hat{\mathcal{P}}}^{\hat{\pi}}, \max_s V_{\gamma, \hat{\mathcal{P}}}^{\hat{\pi}}(s) - g_{\hat{\mathcal{P}}}^{\hat{\pi}}\right]} \left| \left( \mathsf{P}_{s,a} - \hat{\mathsf{P}}_{s,a} \right) [V_{\gamma, \hat{\mathcal{P}}}^{\hat{\pi}} - g_{\hat{\mathcal{P}}}^{\hat{\pi}}]_\alpha \right| +
$$

$$
+ \max_{\alpha \in \left[\min_s V_{\gamma, \hat{\mathcal{P}}}^{\hat{\pi}}(s) - g_{\hat{\mathcal{P}}}^{\hat{\pi}}, \max_s V_{\gamma, \hat{\mathcal{P}}}^{\hat{\pi}}(s) - g_{\hat{\mathcal{P}}}^{\hat{\pi}}\right]} \sqrt{R} \left| \sqrt{\mathsf{Var}_{\hat{\mathsf{P}}_{s,a}}\left([V_{\gamma, \hat{\mathcal{P}}}^{\hat{\pi}} - g_{\hat{\mathcal{P}}}^{\hat{\pi}}]_\alpha\right)} - \sqrt{\mathsf{Var}_{\mathsf{P}_{s,a}}\left([V_{\gamma, \hat{\mathcal{P}}}^{\hat{\pi}} - g_{\hat{\mathcal{P}}}^{\hat{\pi}}]_\alpha\right)} \right|. \tag{183}
$$

The first term in (183) can be bounded through Hoeffding's inequality as

$$
\max_{\alpha \in \left[\min_s V^{\hat{\pi}}_{\gamma,\hat{\mathcal{P}}}(s) - g^{\hat{\pi}}_{\hat{\mathcal{P}}}, \max_s V^{\hat{\pi}}_{\gamma,\hat{\mathcal{P}}}(s) - g^{\hat{\pi}}_{\hat{\mathcal{P}}}\right]} \left| \left( \mathsf{P}_{s,a} - \hat{\mathsf{P}}_{s,a} \right) \left[ V^{\hat{\pi}}_{\gamma,\hat{\mathcal{P}}} \right]_\alpha \right|
$$

$$
\leq \max_{\alpha \in [-\mathcal{H}, \mathcal{H}]} \left| \left( \mathsf{P}_{s,a} - \hat{\mathsf{P}}_{s,a} \right) \left[ V^{\hat{\pi}}_{\gamma,\hat{\mathcal{P}}} \right]_\alpha \right|
$$

$$
\leq 4 \sqrt{ \mathcal{H}^2 \frac{\log\left( \frac{3SAN^{3/2}}{(1-\gamma)\delta} \right)}{N} }. \tag{184}
$$

We then consider the second term of (183). Towards this, we can construct an auxiliary robust MDP $(\mathcal{S}, \mathcal{A}, \mathcal{P}^{s,u}, r_{s,u}, \gamma)$ as in Section D.2.2. in (Shi et al., 2023), so that $V^{\hat{\pi}}_{\gamma,\hat{\mathcal{P}}} - g^{\hat{\pi}}_{\hat{\mathcal{P}}} = V_{s,u^*} - g_{s,u^*}$ for some $u^* \in [-\mathcal{H}, \mathcal{H}]$. We then construct an $\epsilon$-net $\mathcal{N}$ over $[-\mathcal{H}, \mathcal{H}]$, so that $|u^* - u| \leq \epsilon$ for some $u \in \mathcal{N}$. Following (Shi et al., 2023), it holds that

$$
\max_{\alpha \in \left[\min_s V^{\hat{\pi}}_{\gamma,\hat{\mathcal{P}}}(s) - g^{\hat{\pi}}_{\hat{\mathcal{P}}}, \max_s V^{\hat{\pi}}_{\gamma,\hat{\mathcal{P}}}(s) - g^{\hat{\pi}}_{\hat{\mathcal{P}}}\right]} \left| \sqrt{\mathsf{Var}_{\hat{\mathsf{P}}_{s,a}} \left( \left[ V^{\hat{\pi}}_{\gamma,\hat{\mathcal{P}}} - g^{\hat{\pi}}_{\hat{\mathcal{P}}} \right]_\alpha \right)} - \sqrt{\mathsf{Var}_{\mathsf{P}_{s,a}} \left( \left[ V^{\hat{\pi}}_{\gamma,\hat{\mathcal{P}}} - g^{\hat{\pi}}_{\hat{\mathcal{P}}} \right]_\alpha \right)} \right|
$$

$$
\leq 6 \sqrt{ \frac{2R\mathcal{H}^2 \log\left( \frac{36SAN^2|\mathcal{N}|}{\delta} \right)}{N} }, \tag{185}
$$

with probability at least $1 - \delta$.

Inserting (185) and (184) back to (183), we have that with probability at least $1 - \delta$,

$$
\left\| \hat{\mathsf{P}}^{\hat{\pi}, \widehat{V}}_w V^{\hat{\pi}}_{\gamma,\hat{\mathcal{P}}} - \mathsf{P}^{\hat{\pi}, \widehat{V}}_w V^{\hat{\pi}}_{\gamma,\hat{\mathcal{P}}} \right\|_\infty \leq 12 \sqrt{ \frac{2(1+R)\mathcal{H}^2 \log\left( \frac{36SAN^2}{\delta} \right)}{N} }. \tag{186}
$$

$\square$

## 14. Function Approximation for Robust AMDPs

The algorithm design part is almost identical to (Zhou et al., 2024). For completeness we provide a brief discussion.

Following (Zhou et al., 2024), we consider a class of Integral Probability Metric (IPM) Uncertainty Set. Given some function class $\mathcal{F} \subset \mathbb{R}^S$ including the zero function, the integral probability metric (IPM) is defined by $d_{\mathcal{F}}(p,q) := \sup_{f \in \mathcal{F}} \{ p^\top f - q^\top f \} \geq 0$ (Müller, 1997). Many metrics such as Kantorovich metric, total variation, etc., are special cases of IPM under different function classes (Müller, 1997). The IPM uncertainty set is defined as $\mathcal{P}^a_s = \{ q : d_{\mathcal{F}}(q, \mathsf{P}^a_s) \leq R \}$.

We also consider the linear function class as (Zhou et al., 2024). Denote $\Phi \in \mathbb{R}^{Sd}$ the feature matrix with rows $\phi(s)$, we set

$$
\mathcal{F} := \{ s \mapsto \psi(s)^\top \xi : \xi \in \mathbb{R}^d, \|\xi\| \leq 1 \}. \tag{187}
$$

Without loss of generality, assume $\Phi$ has full column rank, and let the first coordinate of $\phi(s)$ be 1 for any $s$.

We then detail the algorithm designs of the two updates steps in Algorithm 2.

---

**Algorithm 3 Robust Linear Temporal Difference (RLTD)**

1: **Input:** $\pi, K$
2: **Initialization:** $\theta_0, s_0$
3: **for** $k = 0, 1, \ldots, K-1$ **do**
4:    Sample $a_k \sim \pi(\cdot|s_k)$, $y_{k+1}$ according to $\mathsf{P}_{s_k, a_k}$, and $s_{k+1}$ from $y_{k+1}$
5:    Update $w_{k+1} = \theta_k + \alpha_k \phi(s_k) \left[ (\hat{\mathbf{T}}^\pi V_{\theta_k})(s_k, a_k, y_{k+1}) - \phi(s_k)^\top \theta_k \right]$
6: **end for**
7: **Output:** $\theta_K$

---

We adopt the following standard assumption from robust RL studies.

---

**Algorithm 4 Robust Q-Natural Policy Gradient (RQNPG)**

---

1: **Input:** $\theta, \eta, w, N$
2: **Initialization:** $u_0, s_0$
3: **for** $n = 0, 1, \ldots, N-1$ **do**
4:     Sample $a_n \sim \pi_w(\cdot|s_n)$, $y_{n+1}$ according to $\mathsf{P}_{s_k, a_k}$ and determine $s_{n+1}$ from $y_{n+1}$
5:     Update $u_{n+1} = u_n + \zeta_n \phi(s_n, a_n) \left[ (\hat{\mathbf{T}}^\pi V_\theta)(s_n, a_n, y_{n+1}) - \phi(s_n, a_n)^\top u_n \right]$
6: **end for**
7: **Output:** $w + \eta u_N$

---

**Assumption 14.1.** There exists $\beta < 1$ such that

$$\gamma q_{s,s'}^a \leq \beta \mathsf{P}_{s,s'}^a, \forall q_s^a \in \mathcal{P}_s^a. \tag{188}$$

This assumption is widely adopted in function approximation studies in robust RL, e.g., (Tamar et al., 2014; Xu & Mannor, 2010; Zhou et al., 2024), to ensure the solvability of function approximation.

We then provide the formal statement of Theorem 5.1.

**Theorem 14.2.** *Set geometrically increasing step sizes $\eta^t \geq \frac{S^2 \mathcal{H}^2}{\epsilon S \mathcal{H} - \epsilon^2} \eta^{t-1}$ for each $t = 1, 2, \ldots, T$. Set $N, K = \mathcal{O}(\mathcal{H}\epsilon^{-2})$ in Algorithm 3 and Algorithm 4, and $T \geq \frac{\mathcal{H} \log \mathcal{H}}{\epsilon}$, then*

$$g_\mathcal{P}^* - \mathbb{E}[g_\mathcal{P}^{\pi_T}] \leq C \left( 1 - \frac{\epsilon}{\mathcal{H}S} \right)^{T-1} + \frac{M\mathcal{H}^2}{\epsilon^2} \epsilon_e^2,$$

*where $C$ is some constant, and $\epsilon_e = \epsilon_{stat} + \epsilon_{bias}$ measures the approximation error .*

*Proof.* To apply Theorem 1 of (Zhou et al., 2024), it is assumed that for initial state distribution $\rho$, there exists $M$ such that $\sup_{\kappa \in \mathcal{P}} \| \frac{d_\rho^{*,\kappa}}{\rho} \|_\infty \leq M < \infty$. We note that since $g_\mathcal{P}^\pi$ is a constant and does not depend on the initial distribution, we can simply set uniform initial distribution $\rho = (\frac{1}{S}, \ldots, \frac{1}{S})$, in which case $M = S$.

By applying Theorem 1 of (Zhou et al., 2024) with $\gamma = 1 - \frac{\epsilon}{\mathcal{H}}$, it holds that

$$V_{\gamma,\mathcal{P}}^* - \mathbb{E}[V_{\gamma,\mathcal{P}}^{\pi_T}] \leq \left( 1 - \frac{\epsilon}{\mathcal{H}S} \right)^{T-1} C + \frac{S\mathcal{H}^2}{\epsilon} \epsilon_e^2. \tag{189}$$

Note that $\epsilon_e = \tilde{O}(\frac{1}{\sqrt{K}} + \frac{1}{\sqrt{N}})$ when omitting the critic error, thus setting $N, K = \mathcal{O}(S\mathcal{H}\epsilon^{-2})$ implies that

$$V_{\gamma,\mathcal{P}}^* - \mathbb{E}[V_{\gamma,\mathcal{P}}^{\pi_T}] \leq \left( 1 - \frac{\epsilon}{\mathcal{H}S} \right)^{T-1} C + S\mathcal{H}. \tag{190}$$

Note that if we set $T$ large enough so that $\left( 1 - \frac{\epsilon}{\mathcal{H}S} \right)^{T-1} \leq \mathcal{H}$, which can be satisfied if $T \geq \frac{\mathcal{H} \log \mathcal{H}}{\epsilon}$, then $\pi_T$ is an $\mathcal{H}$-optimal policy for the robust DMDP. Combining the above with Theorem 3.4 completes the proof. $\square$

### 14.1. Details for Experiments

To expand on our results in Section 6, we provide additional detail here. In our experiments, for different discount factors $\gamma$, we run the corresponding RNAC algorithm (Zhou et al., 2024) to learn a robust policy, and then we estimate the robust average reward of the learned policy by calculating the average reward under environment perturbations. Experiments were performed in two custom perturbed MuJoCo environments, Walker2d-v3 and Hopper-v3, using the integral probability metric (IPM) uncertainty set (Zhou et al., 2024). We evaluate the performance of our algorithm with different perturbation levels, and the results, including one standard deviation from the mean, are shown below in Figure 10 for varying perturbation levels. These experimental results align with our theoretical results, and thus verify the scalability of combining our framework with function approximation.

To show that our framework does not result in high computational costs regarding varying reduction factors, we show the execution time of our method using these factors. As can be seen in Tables 1 and 2, even with a very large value for $\gamma$, the

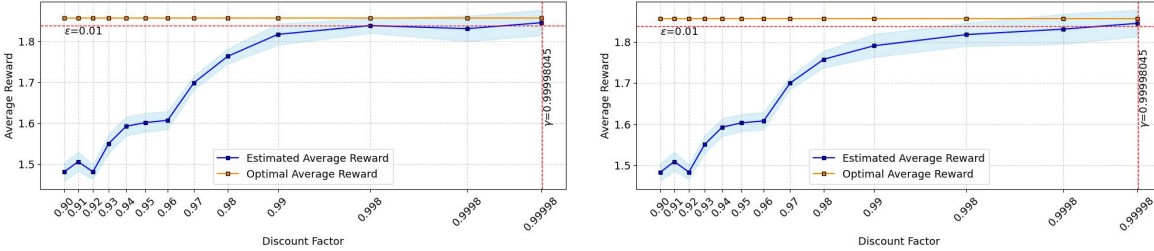

*Figure 10.* Scalability under Walker2d-v3.

execution time is similar, thus showing that our method is also computationally efficient. Similar to how we showed in Section 6, we show additional results comparing our robust reduction method to that of the non-robust reduction method in Figures 11 and 12 for neural network approximation in the Walker2d-v3 and Hopper-v3 environments, respectively. Additionally, while the focus of this work is on distributional robustness, we hypothesize that our reduction framework *should* work under the adversarial robustness formulation. Due to this idea and inspired by (Zhang et al., 2020), we conducted a preliminary experiment on (discounted) adversarial robust RL in the Humanoid-v4 environment using increasing factors of $\gamma$. As our result in Figure 13 shows, the reward under attack increases as $\gamma$ increases, indicating the potential to develop a similar reduction framework for this setting.

*Table 1.* Computational efficiency under the Walker2d-v3 environment.

| PHASE | $\gamma$=[0.9, 0.99, 0.999, 0.9999, 0.99998, 0.999980448383733] |
|---|---|
| TRAINING | 603.92, 630.72, 690.33, 706.32, 711.70, 676.36 |
| EVALUATION | 5.93, 25.60, 10.03, 9.16, 10.44, 6.05 |

*Table 2.* Computational efficiency under the Hopper-v3 environment.

| PHASE | $\gamma$=[0.9, 0.99, 0.999, 0.9999, 0.99998, 0.9999925319260162] |
|---|---|
| TRAINING | 579.41, 634.98, 678.29, 682.53, 686.92, 683.56 |
| EVALUATION | 6.42, 37.80, 7.15, 5.80, 7.10, 6.84 |

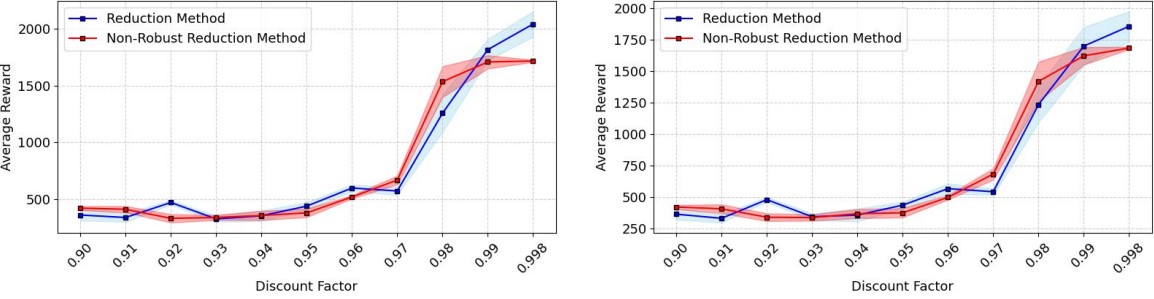

*Figure 11.* Neural network approximation under Walker2d-v3.

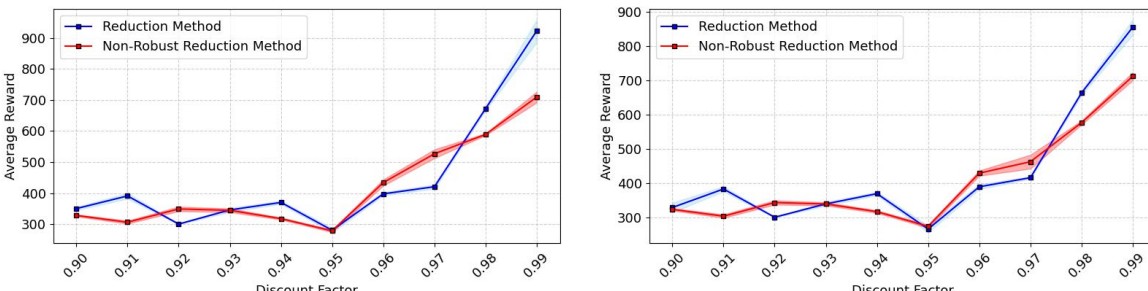

*Figure 12.* Neural network approximation under Hopper-v3.

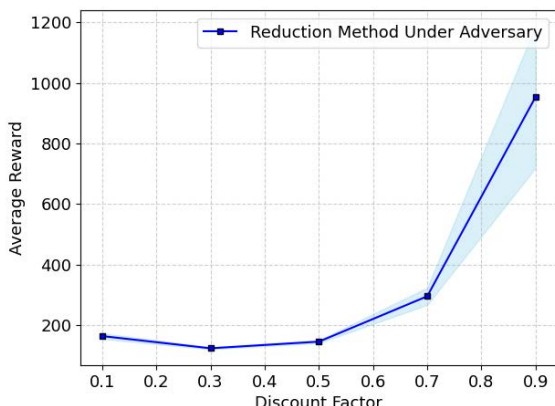

*Figure 13.* Preliminary discounted adversarial robust RL under Humanoid-v4.

## 15. Model-Free RL for Robust AMDPs

### 15.1. Multi-Level Monte Carlo Robust Q-Learning

One potential way to improve scalability is to design model-free algorithms. In contrast to the model-based method in the last section, model-free methods do not store the transition kernels and aim to learn the optimal policy directly. To illustrate the applicability of our framework and ensure it's scalability, we develop two model-free algorithms that can be applied efficiently for large-scale problems. First, we introduce an algorithm that combines our reduction framework with the multi-level Monte Carlo (MLMC) technique (Liu et al., 2022b; Wang et al., 2023e; Blanchet & Glynn, 2015; Blanchet et al., 2019; Wang & Wang, 2022), and provide a theoretical analysis of its sample complexity. We further design another mini-batch model-free robust Q-learning algorithm for robust AMDP in Section 15.2. Both algorithms do not require model estimation and storage, making them scalable for robust RL with average reward.

The MLMC approach is widely used to construct an unbiased estimator of the robust Bellman operator, which is challenging due to its non-linear dependence on the nominal kernel[9] (Wang et al., 2023e). It relies on a geometric distribution with parameter $\Psi \in (0, 1)$, and requires $2^{N+1}$ samples at each step, with $N \sim \mathbf{Geom}(\Psi)$, to construct an unbiased estimator of $\sigma_{\mathcal{P}_s^a}(V)$.

By adapting the MLMC estimator with our reduction-based framework, we propose the MLMC robust Q-learning algorithm for robust AMDPs, given in Algorithm 5.

Our algorithm is hence the first model-free one for robust RL under average reward, along with finite sample analysis. These results demonstrate the broad applicability of our reduction-based framework, enabling the direct integration of any model-free algorithm designed for robust DMDPs to yield algorithms for robust AMDPs with sample complexity guarantees.

---

[9]Specifically, the support function $\sigma_{\mathcal{P}_s^a}(V)$ is not linear in $\mathrm{P}_s^a$.

---

**Algorithm 5** MLMC robust Q-learning for robust AMDPs

---

1: **Input:** A generative model of $(\mathcal{S}, \mathcal{A}, \mathsf{P}, r)$, uncertainty radius $\sigma$, robust bias span $\mathcal{H}$, accuracy $\epsilon$, $\Psi = 0.5$, threshold $N_m$
2: **Initialization:** $Q_1 \leftarrow 0, s_0$
3: $\gamma \leftarrow 1 - \frac{\epsilon}{\mathcal{H}}, T \leftarrow \frac{1}{\mathcal{H}^2(1-\gamma)^5}$
4: **for** $t < T$ **do**
5:     **for** all $s \in \mathcal{S}, a \in \mathcal{A}$ **do**
6:         $V(s) \leftarrow \max_a Q(s, a)$
7:         Sample $N \sim \text{Geom}(\Psi)$
8:         $N \leftarrow \min\{N_m, N\}$
9:         Sample $2^{N+1}$ samples following $\mathsf{P}_s^a$
10:        Obtain the multi-level estimator according to (191)
11:        $Q(s, a) \leftarrow r(s, a) + \gamma \hat{\sigma}_{\mathcal{P}_s^a}(V)$
12:     **end for**
13: **end for**
14: $\hat{\pi}_\gamma(s) \leftarrow \arg\max_a Q(s, a), \forall s$
15: **Output:** $\hat{\pi}_\gamma$

---

As a result, our framework is scalable to large-scale problems while maintaining high data efficiency.

The MLMC operator is based on a geometric distribution with parameter $\Psi \in (0, 1)$. For any $s, a$, we first generate a number $N$ from a geometric distribution with parameter $\Psi \in (0, 1)$. Then, we take action $a$ at state $s$ for $2^{N+1}$ times, and observe $r(s, a)$ and the subsequent state $\{s_i'\}, i = 1, \ldots, 2^{N+1}$. We divide these $2^{N+1}$ samples into two groups: samples with odd indices, and samples with even indices. We then individually calculate the empirical distribution of $s'$ using the even-index samples, the odd-index samples, all the samples, and the first sample: $\hat{\mathsf{P}}_{s,N+1}^{a,E} = \frac{1}{2^N} \sum_{i=1}^{2^N} \mathbf{1}_{s_{2i}'}$, $\hat{\mathsf{P}}_{s,N+1}^{a,O} = \frac{1}{2^N} \sum_{i=1}^{2^N} \mathbf{1}_{s_{2i-1}'}$, $\hat{\mathsf{P}}_{s,N+1}^{a} = \frac{1}{2^{N+1}} \sum_{i=1}^{2^{N+1}} \mathbf{1}_{s_i'}$, $\hat{\mathsf{P}}_{s,N+1}^{a,1} = \mathbf{1}_{s_1'}$. Then, we use these estimated transition kernels as nominal kernels to construct four estimated uncertainty sets (with the same uncertainty radius): $\hat{\mathcal{P}}_{s,N+1}^{a,E}, \hat{\mathcal{P}}_{s,N+1}^{a,O}, \hat{\mathcal{P}}_{s,N+1}^{a}, \hat{\mathcal{P}}_{s,N+1}^{a,1}$. The multi-level estimator is then defined as

$$\hat{\sigma}_{\mathcal{P}_s^a}(V) \triangleq \sigma_{\hat{\mathcal{P}}_{s,N+1}^{a,1}}(V) + \frac{\Delta_N(V)}{p_N}, \tag{191}$$

where $p_N = \Psi(1 - \Psi)^N$ and

$$\Delta_N(V) \triangleq \sigma_{\hat{\mathcal{P}}_{s,N+1}^a}(V) - \frac{\sigma_{\hat{\mathcal{P}}_{s,N+1}^{a,E}}(V) + \sigma_{\hat{\mathcal{P}}_{s,N+1}^{a,O}}(V)}{2}.$$

Our threshold-MLMC estimator is constructed as follows

$$\hat{\sigma}_{\mathcal{P}_s^a}(V) \triangleq \sigma_{\hat{\mathcal{P}}_{s,\min\{N_m+1,N+1\}}^{a,1}}(V) + \frac{\Delta_{\min\{N,N_m\}}(V)}{p_{\min\{N,N_m\}}}.$$

The formal statements on the sample complexity are as follows. The proofs are straightforward by combining our framework with the results in (Wang et al., 2024c).

**Theorem 15.1.** *(1). For the TV-defined uncertainty set, set* $N_m = \frac{2 \log T}{\log 2}$ *and the step size as* $\beta_t = \frac{2 \log T}{(1-\gamma)T}$. *Then, the output of Algorithm 5 is an $\epsilon$-optimal policy for the robust AMDP if*

$$N \geq \widetilde{\mathcal{O}}\left(\frac{SA\mathcal{H}^3}{\epsilon^5}\right). \tag{192}$$

*(2). For the Chi-Square-divergence-defined uncertainty set, set* $N_m = \frac{2 \log T}{\log 2}$ *and step size as* $\beta_t = \frac{2 \log T}{(1-\gamma)T}$. *Then the output of Algorithm 5 is an $\epsilon$-optimal policy for the robust AMDP if*

$$N \geq \widetilde{\mathcal{O}}\left(\frac{SA\mathcal{H}^3}{\epsilon^5}\right). \tag{193}$$

*(3). For the KL-divergence-defined uncertainty set, set*

$$N_m = \max\left\{\frac{2\log T}{\log 2}, \frac{\log(1 + p_\wedge^2 \log(2S)\log T)}{\log 2}\right\},$$

*where $p_\wedge$ is the minimal positive entry of the nominal kernel* P. *Then, the output of Algorithm 5 is an $\epsilon$-optimal policy for the robust AMDP if*

$$N \geq \widetilde{\mathcal{O}}\left(\frac{SA\mathcal{H}^3}{p_\wedge^2 \epsilon^5}\right). \tag{194}$$

## 15.2. Mini-Batch Robust Q-Learning with Variance Reduction

In this section, we present a model-free mini-batch Q-learning algorithm. The algorithm is derived from the robust DMDP algorithm (Wang et al., 2023c), which employs a variance reduction technique to improve the sample complexity. When combined with our framework, we can also achieve improved sample complexity for robust AMDPs. The details can be found in Algorithm 6.

---

**Algorithm 6** Mini-batch robust Q-learning for robust AMDPs

---

1: **Input:** A generative model of $(\mathcal{S}, \mathcal{A}, \mathsf{P}, r)$, uncertainty radius $\sigma$, robust bias span $\mathcal{H}$, accuracy $\epsilon$, batch size $n$, and behavior policy $\pi$
2: **Initialization:** $Q_1 \leftarrow 0$, $s_0$
3: Set $\gamma \leftarrow 1 - \frac{\epsilon}{\mathcal{H}}$
4: **for** $t < T$ **do**
5:     $V_t(s) \leftarrow \max_a Q_t(s, a), \forall s$
6:     Sample $a_t \sim \pi(\cdot|s_t)$
7:     Sample $n$ samples from $\mathsf{P}_{s_t}^{a_t}$ and obtain empirical uncertainty set $\hat{\mathcal{P}}_{s_t}^{a_t}$
8:     $\lambda_t \leftarrow \frac{1}{1+(1-\gamma)t}$
9:     $Q_{t+1}(s_t, a_t) \leftarrow (1 - \lambda_t)Q_t(s_t, a_t) + \lambda_t(r(s_t, a_t) + \gamma\sigma_{\hat{\mathcal{P}}_{s_t}^{a_t}}(V_t))$
10:     Sample the next state $s_{t+1} \sim \mathsf{P}_{s_t}^{a_t}$ from the generative model
11: **end for**
12: $\hat{\pi}_\gamma(s) \leftarrow \arg\max_a Q_T(s, a), \forall s$
13: **Output:** $\hat{\pi}_\gamma$

---

We then derive the sample complexity of the mini-batch robust Q-learning algorithm, as stated in the following theorem.

**Theorem 15.2.** *Consider the uncertainty set defined by the KL divergence. If we set*[10]

$$T = \tilde{\mathcal{O}}\left(\frac{SA\mathcal{H}^2}{\epsilon^2}\right), \ n = \tilde{\mathcal{O}}\left(\frac{\mathcal{H}}{\epsilon}\right) \tag{195}$$

*in Algorithm 6, then the output policy $\hat{\pi}_\gamma$ of Algorithm 6 is an $\epsilon$-optimal policy for robust AMDP.*

The result shows that the mini-batch robust Q-learning requires $\tilde{\mathcal{O}}\left(\frac{SA\mathcal{H}^3}{\epsilon^3}\right)$ samples to obtain an $\epsilon$-optimal policy for the robust AMDP. This improves the sample complexity for our MLMC robust Q-learning algorithm and represents the state-of-the-art in robust AMDP model-free sample complexity.

## 15.3. Numerical Experiments

To exemplify the scalability of our framework for robust AMDPs, we present concrete proof of convergence of our model-free *MLMC robust Q-learning* algorithm in figure 14. We first created the uncertainty set using total variation. Then for each $\gamma$ value we wanted to test, we generate a number $N$ from a geometric distribution with $\Psi \in (0, 1)$ and take action $a$ at state

---

[10]We omit the complex dependencies of $T$ and $n$ on other parameters, including the uncertainty set radius, to highlight the potential of our framework.

$s$ for $2^{N+1}$ times in order to learn the optimal policy under the robust discounted DMDP. Once we have the $\epsilon_\gamma$-optimal policy, $\hat{\pi}_\gamma$, we then applied algorithm 1 from (Wang et al., 2023d) to evaluate the robust average reward for each policy. This experiment was independently conducted 5 times, where we plot the mean of the estimated robust average reward for every $\gamma$ factor along with 1 standard deviation above and below the mean in figure 14. Following this, we then implemented algorithm 2 from (Wang et al., 2023d) in order to find the optimal robust average reward. By combining our reduction-based framework with the multi-level Monte Carlo (MLMC) technique (Liu et al., 2022b; Wang et al., 2023e; Blanchet & Glynn, 2015; Blanchet et al., 2019; Wang & Wang, 2022), we are able to prove our theoretical results in showing that our model-free algorithm converges to the optimal robust average reward.

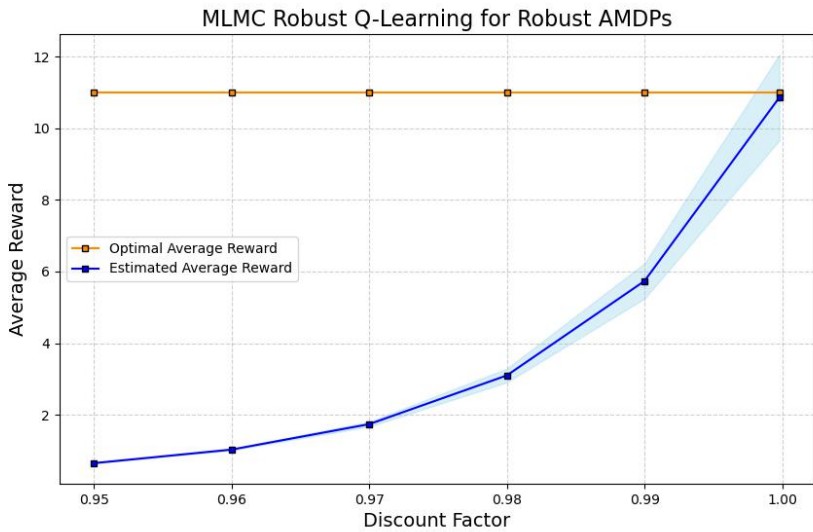

*Figure 14.* Model-Free Experimental Results Under Total Variation

