# OpenReview forum: "A Reduction Framework for Distributionally Robust Reinforcement Learning under Average Reward"
_ICML.cc/2025/Conference — ICML 2025 poster_

### Official Review · Reviewer_chdA · 2025-03-11

**Overall Recommendation:** 3

**Summary:**

The paper "Efficient and Scalable Reinforcement Learning for Average Reward under Model Uncertainty" focuses on solving robust reinforcement learning (RL) with the average reward criterion, which is crucial for optimizing long-term performance in uncertain environments. The key challenge is that most robust RL methods focus on discounted rewards, while average reward RL is more suitable for real-world applications like queuing systems, supply chain management, and communication networks. However, robust average reward RL is difficult due to non-contractive Bellman operators, high-dimensional solution spaces, and model mismatches between training and deployment environments.

**Claims And Evidence:**

The paper claims that their framework scales well with large problems through function approximation, but it only demonstrates scalability using linear function approximation. They do not provide theoretical guarantees for non-linear function approximation, making the robustness of their approach questionable in high-dimensional problems.

**Essential References Not Discussed:**

No.

**Experimental Designs Or Analyses:**

1. The experiments focus only on synthetic perturbations in MuJoCo but do not test adversarial robustness in real-world deployment settings (e.g., changes in environment dynamics, sensor noise).

**Methods And Evaluation Criteria:**

1. The uncertainty estimation seems not practical, since it is computed primarily under total variation, chi-squared, and KL divergence. Although those methods are commonly used in theory, but may not computationally efficient. Extending the framework to more realistic non-parametric uncertainty sets would be better.

2. Robust policy evaluation requires solving a min-max problem over an uncertainty set, which may not be tractable for large-scale problems.

**Other Comments Or Suggestions:**

No.

**Other Strengths And Weaknesses:**

See comments above.

**Questions For Authors:**

Please answer the questions mentioned above.

**Relation To Broader Scientific Literature:**

This paper formalizes a reduction framework with a concrete discount factor selection, making the connection practical for real-world use. It also extends sample complexity analysis for robust RL, building on Agarwal et al. (2020) for model-based RL and Panaganti & Kalathil (2022) for robust RL under uncertainty sets (e.g., TV, Chi-squared, KL divergence).

**Theoretical Claims:**

Yes.

---

> ### Author Rebuttal · Authors · 2025-04-01
>
> We sincerely thank the reviewer for your time and thoughtful feedback, and we are glad to hear that our work is appreciated.
>
> We conducted additional experiments, at the link: https://anonymous.4open.science/r/ICML-2662-4C1E/README.md
>
> **W1: Scalability is demonstrated using linear function. No theoretical guarantees for non-linear function approximation.**
>
> We use the commonly adapted linear function approximation to illustrate scalability, as it has better mathematic properties and enables rigorous theoretical studies. Under it, we develop both empirical and theoretical studies.
>
> However, our major contribution is the reduction of the challenging robust average reward, which is independent from algorithms used. Thus we can adopt any algorithms for the discounted reward, no matter what function approximation is used. As an empirical evidence, we develop additional experiments with neural network approximation, shown in E2 in the link above. As the results show, our robust reduction method enjoys enhanced robustness against non-robust baselines, and the increasing robust average reward as $\gamma$ increasing further verifies the effectiveness of our reduction approach, even with neural network approximation. However, even standard non-robust RL with neural network approximation generally lacks theoretical guarantees, thus it can be challenging to derive theoretical results. We leave this as a future direction.
>
>
>
>
> **W2 \& W3: The uncertainty estimation is not practical, and thus extending the framework to a non-parametric uncertainty set would be beneficial. Robust policy evaluation requires solving a min-max problem over an uncertainty set, which may not be tractable for large-scale problems.**
>
>
> We first emphasize that the computational cost for finding the worst performance (i.e., robust policy evaluation) in TV and $\chi^2$ sets are $\mathcal{O}(S\log S)$ (Iyengar, 2005) under the tabular setting, thus their estimation can be tractable.
>
> On the other hand, developing efficient methods for distributionally robust RL with large scale is still an open question. One potential way is to relax the uncertainty set constraints, which can result in an efficient solution to the support function, even for large or continuous spaces [1,2] (Zhou et al., 2024). Adversarial training techniques can also be applied to approximate the robust value functions [3-5].
>
>
> We agree that the framework for robust RL with a non-parametric uncertainty set is important, but it generally lacks a concrete theoretical formulation. However, we expect that our intuition of reducing the average reward to the discounted reward still remains valid under different uncertainty sets. We conduct a preliminary experiment under adversarial attacks (please see below) to illustrate this.
>
>
> [1] Kumar, N. et al. Policy gradient for rectangular robust Markov decision processes. 2023.
> [2] Zhang, Y. et al. Robust reinforcement learning in continuous control tasks with uncertainty set regularization. 2023.
> [3] Huang, S., et al.  Adversarial attacks on neural network policies. 2017.
> [4] Pinto, L., et al.  Robust adversarial reinforcement learning. 2017.
> [5] Yu, T., et al. Combo: Conservative offline model-based policy optimization. 2021.
>
>
>
> **W4: The experiments do not test adversarial robustness in real-world deployment settings.**
> We emphasize that we consider the distributionally robust RL formulation, which aims to enhance robustness under dynamic uncertainty, and is more tractable for theoretical studies than other formulations of robustness. Since we focus more on the theoretical aspect, we only use synthetic experiments to verify our results. In our experiments, we applied perturbations to the joints and other parameters to simulate dynamic uncertainties, which provides evidence for our theoretical claims.
>
> We acknowledge that adversarial robustness is also important, and we believe that our reduction framework can also be extended to the adversarial robustness setting. We conduct a preliminarily experiment inspired by [1] in the Humanoid-v4 environment, shown in E5 in the link above. We implement (discounted) adversarial robust RL under increasing factors, and plot the average reward under attack. As the results show, the performance improves as $\gamma$ increases, indicating the potential of developing a similar reduction framework. We will conduct more comprehensive experiments and include them in the final version.
>
>
> [1] Zhang, H. et al,  Robust deep reinforcement learning against adversarial perturbations on state observations, 2020.

---

### Official Review · Reviewer_PFDp · 2025-03-14

**Overall Recommendation:** 4

**Summary:**

This work builds a framework to aid in the reduction of average reward MDPs to robust discounted reward MDPs. Previous work has shown that as discount factor approaches 1, a policies value function in a robust discount reward MDP approaches the return found in an average reward MDP. This work builds a framework that helps choose the correct discount factor in a robust DMDP that achieves the desired approximation to a robust AMDP.

**Claims And Evidence:**

The main claim in this work is that solving a DMDP with a discount factor of $1- \frac{\epsilon}{\mathcal{H}}$ within $\epsilon$ of the optimal policy will correspondingly produce a policy in the ADMP where the suboptimality is bounded by $8 + \frac{5\epsilon_\gamma}{\mathcal{H}}$.  This claim is proven in the appendix.

Further, the work introduces a model-based algorithm for solving robust AMDPs, under strict assumptions that the transition kernel can be arbitrarily queried. The work also claims that this method can scale to the function approximation setting, specifically with linear and neural network function approximaters.

**Essential References Not Discussed:**

N/a

**Experimental Designs Or Analyses:**

The experiments in this work are minimal and discussed in the Methods and Evaluations Criteria section. This work presents a mostly theoretical framework.

**Methods And Evaluation Criteria:**

To verify empirically the choice of discount factor necessary for reduction, the authors solve the Garnet problem tabularly under different discount factors. They demonstrate (in Figure 1) that the predicted discount factor produces the $\epsilon$-optimal policy in the DMDP setting and the equivalent in the AMDP setting.

Further, Figure 2 demonstrates that the performance of the proposed methods scales with the size of the dataset, eventually reaching optimal performance. Lastly, the Figures 3 and 4 demosntrate the ability to find the optimal robust AMDP policy on common RL benchmark tasks in Mujoco.

**Other Comments Or Suggestions:**

n/a

**Other Strengths And Weaknesses:**

Beyond more extensive empirical evaluation, this paper would benefit greatly from including some intuition for Theorem 3.4 in the main paper, instead of relinquishing the proof to the appendix.

**Questions For Authors:**

n/a

**Relation To Broader Scientific Literature:**

The main result is the reduction of the robust AMDP to a robust DMDP. This provides practitioners a tractable way to solve both types of MDPs interchangeably.

**Theoretical Claims:**

Not verified.

---

> ### Author Rebuttal · Authors · 2025-04-01
>
> We sincerely thank the reviewer for your time and thoughtful feedback, and we are glad to hear that our work is appreciated.
>
> We conducted additional experiments, at the link: https://anonymous.4open.science/r/ICML-2662-4C1E/README.md
>
> **W1: More extensive empirical evaluation.**
> Since we focus mainly on theoretical studies, we only use experiments as illustrations of our claims. We first want to mention that have some additional experiments in the appendix. We additionally develop the following experiments.
>
> 1. To verify the effectiveness of our reduction framework, we conduct experiments with reduction factor estimation and the optimal robust average reward in E1 in the link above. As in Remark 3.3, our framework and subsequent results are still effective if $\mathcal{H}$ is replaced by its upper bound. We adopt the worst-case diameter, which is an upper bound and is easier to estimate [Wang et al., 2022], as an alternative of $\mathcal{H}$. We estimate the worst-case diameter for the Walker ($\approx 511$) and the the Hopper environment ($\approx 1339$), and plot the robust average reward of the policies learned from corresponding discount factors. We also calculate the reduction discount factor for different optimality level $\epsilon$. As the results show, our reduction framework (with the reduction factor) obtains a policy with a suboptimality gap within the error range, which verifies our theoretical results and illustrate the effectiveness of our framework, even in large scale setting.
>
> 2. To verify the scalability, we conduct experiments with neural network approximation in E2 in the link above. As the results showed, our method is more robust to the non-robust baseline, while also maintain scalable.
>
> 3. To enhance our empirical evaluation, we develop a primarily experiment under more practical adversarial robustness, under the humanoid environment. Although adversarial robustness has a different formulation from the distributional robustness we study, we believe that the reduction framework should also work. We conduct a preliminarily experiment inspired by [1] in the Humanoid-v4 environment. We implement (discounted) adversarial robust RL under increasing factors, and plot the average reward under attack. As the results (E5 of the link above) show, the performance improves as $\gamma$ increases, hence the reduction framework should also remain effective, i.e., there exists a reduction factor to reduce the average reward to discounted reward. It is also promising to develop a similar reduction framework for adversarial robustness. We leave the development of theoretical analysis and the identification of the reduction factor for future work.
>
> 4. We compare our method to other baselines in (Wang et al., 2023c, Wang et al., 2023d). These methods only have asymptotic convergence guarantees, and can only be applied for tabular settings, whereas our method has finite sample complexity analysis and can be applied with function approximation. We compare our method to the two baselines, robust Value iteration (RVI) (Wang et al., 2023c) and robust relative value iteration (RRVI) (Wang et al., 2023d), under the tabular Garnet problem. Results in E3 in the link show that our method finds a better policy with the same number of steps. Thus, our method achieves state-of-the-art performance in robust average reward optimization
>
> We will also include more experiments in the final version.
>
> [1] Zhang, H. et al,  Robust deep reinforcement learning against adversarial perturbations on state observations, 2020.
>
> **W2: The intuition for Theorem 3.4 should be in the main paper instead of leaving everything to the proof.**
> We would like to thank you for your suggestion. We provide a brief sketch of the proof for our reduction framework.
>
> The key step is the study of the convergence error of the robust discounted value function to average reward. The result (Lemma 11.1) shows that the error is upper bounded by the Span semi-norm of the robust discounted vector $\|(1-\gamma)V^\pi_\gamma-g^\pi \|\leq Sp ((1-\gamma)V^\pi_\gamma)$, for any policy $\pi$. We can thus control the choice of $\gamma$ so that the RHS is smaller than $\epsilon$. We can then decompose the error between the optimal robust average reward and the one under the learned $\epsilon_\gamma$-optimal policy $\pi$ (we omit $\gamma$ from $V^\pi_\gamma$): $g^{\pi^*}-g^\pi= g^{\pi^*}-(1-\gamma) V^{\pi^*}+(1-\gamma) V^{\pi^*}-(1-\gamma) V^\pi+(1-\gamma) V^\pi-g^\pi\leq \epsilon+\epsilon_\gamma+\epsilon$, which completes the proof of the reduction.
>
> The improvement of sample complexity for model-based methods is also based on this key result. Using the fact that $\|(1-\gamma)V^\pi_\gamma-g^\pi \|\leq Sp ((1-\gamma)V^\pi_\gamma)$, we can bound the variance terms (the main challenging term in the complexity analysis) in terms of $\mathcal{H}$ instead of $(1-\gamma)$, thereby improving the results.
>
> We will provide a more detailed sketch in the final version.

---

> > ### Comment · Reviewer_PFDp · 2025-04-03
> >
> > Authors,
> >
> > Thank you for addressing my concerns and answering my questions. I maintain my score.

---

### Official Review · Reviewer_JZQS · 2025-03-20

**Overall Recommendation:** 3

**Summary:**

This work proposes a reduction-based framework that converts robust average reward optimization into robust discounted reward optimization by selecting an appropriate discount factor. The framework focuses on total variation (TV) and $\chi^2$ divergence and introduces a model-based algorithm with near-optimal sample complexity. It also applies function approximation techniques to robust average reward and proposes a robust natural actor-critic (NAC) algorithm with linear function approximation. The experimental results demonstrate the effectiveness of the proposed approach.

**Claims And Evidence:**

Yes. Claims made in the submission supported by clear and convincing evidence.

**Essential References Not Discussed:**

No.

**Experimental Designs Or Analyses:**

The author presents two sets of experiments. In the first part, Garnet MDP is used to evaluate the data efficiency of the algorithm. In the second part, two classic examples from GYM are used to demonstrate scalability.

**Methods And Evaluation Criteria:**

Yes.

**Other Comments Or Suggestions:**

1. Figure $4$ is fuzzy.

2. What is the computational time of the algorithm as the state size $S$ and action size $A$ increase?

3. In Algorithm $2$, the main part of the algorithm is similar to the structure of Robust Natural Actor-Critic. The author should provide more explanation about the contributions of the proposed algorithm, or compare the computational time of the tailored algorithm $2$ with the original Robust Natural Actor-Critic.

**Other Strengths And Weaknesses:**

Strengths:
The paper is well-structured and clearly presents its ideas. The authors provide a near-optimal complexity supported by strong proofs. In addition to proposing a model-based algorithm for robust AMDPs, they also introduce the Reduction Robust Natural Actor-Critic approach to address problems with function approximation. The experimental results are convincing and demonstrate the efficiency and scalability of the proposed method.

Weaknesses:
1. This work primarily focuses on total variation (TV) and $\chi^2$ divergence. Citing relevant references would help justify that these divergences are commonly used. Incorporating KL-divergence [1,2] or Wasserstein distance [1,2] could further improve the generality and applicability of the approach.

2. In the experiment, Figure 1 shows the robust average reward with different discount factors. According to the definition

\begin{equation}
V_{\gamma, \mathcal{P}}^\pi(s) \triangleq \min _{\kappa \in \bigotimes_{t \geq 0} \mathcal{P}} \mathbb{E}_{\pi, \kappa}\left[\sum_{t=0}^{\infty} \gamma^t r_t \mid S_0=s\right],
\end{equation}

a larger $\gamma$ should imply a larger reward. However, in Figure $1$, there are cases where a smaller $\gamma$ performs better. This observation is also different from Figure $5$ in Appendix $9$.

3. In the experimental section, it would be beneficial to perform out-of-sample testing and use the total reward to compare AMDP and DMDP(different $\gamma$). This would better demonstrate the advantage of the proposed method.

Reference:

[1] J. Grand-Clément and M. Petrik. Reducing Blackwell and average optimality to discounted MDPs via the Blackwell discount factor. Advances in Neural Information Processing Systems, 36, 2024.

[2] Y. Wang, A. Velasquez, G. K. Atia, A. Prater-Bennette, and S. Zou. Model-free robust average-reward reinforcement learning. In Proceedings of the International Conference on Machine Learning (ICML), pp. 36431–36469. PMLR, 2023d.

**Questions For Authors:**

No.

**Relation To Broader Scientific Literature:**

Compared to previous work, this approach applies to a more general setting and achieves better sample complexity.

**Theoretical Claims:**

Yes.

---

> ### Author Rebuttal · Authors · 2025-04-01
>
> We sincerely thank the reviewer for your time and thoughtful feedback, and we are glad to hear that our work is appreciated.
>
> We conducted additional experiments, at the link: https://anonymous.4open.science/r/ICML-2662-4C1E/README.md
>
> **W1: Results for other uncertainty sets**
> We first want to highlight that the TV and $\chi^2$ are the most commonly studied in robust RL, e.g., (Shi et al., 2023; Panaganti \& Kalathil, 2022; Yang et al., 2021), and we develop our studies for them to compare with existing results and illustrate the advantages of ours.
>
> However, our framework is independent of the uncertainty set used, hence results under other uncertainty sets can be directly obtained. For example, the results under KL divergence are discussed in Line 237 and Theorem 15.1 and 15.2 in Appendix. Results for Wasserstein distance can be similarly obtained based on [1]. However, as we discussed in Section 4, results directly obtained are generally suboptimal due to the loose bound associated with the discounted reward  setting. Tightening this bound and improving the sample complexity, however, can be challenging and may require tailored efforts for each individual set. We therefore leave these investigations for future work.
>
> [1] Xu, Z., et al. Improved sample complexity bounds for distributionally robust reinforcement learning. 2023.
>
> **W2: Results in Fig 1**
> In Fig 1 we plot the robust average (not discounted) reward $g_{\mathcal{P}}$ of the policies learned under different discount factors. We want to clarify that increasing $\gamma$ does not imply the increase of the average reward. It holds that $(1-\gamma)V_{\gamma,\mathcal{P}} \to g_{\gamma,\mathcal{P}}$ (eq 6), however, there may not exist a monotonic dependence of the average reward on the discount factor, and hence a larger discount factor can still result in a smaller average reward. This instability is one of the motivations of our work: rather than selecting a large factor that may still lead to suboptimal performance, we aim to identify a specific factor with a performance guarantee.
>
> **W3: Out-of-sample testing**
> We first want to clarify that the experiments are performed with out-of-sample testing. We trained our algorithms under the nominal kernel and tested them under the worst-case kernel, which is generally different from the raining environment. Thus, the testing environments are never seen during training, and hence they are out-of-sample testing.
>
> As for the advantages of our method v.s. the discounted one, as our results showed, with a small discount factor (less than the reduction factor we choose for our method), the robust average reward (reflects the total reward by tracking the total time steps) is higher, thereby demonstrating the benefit of our approach. Additional experiments are showed in
> E1 in the link, where we plot the optimal value and our reduction factor. As the results show, our method (with the reduction factor) results in a near-optimal policy, whereas the DMDP are worse.
>
> **Q1: Figure 4 is fuzzy** Thank you for letting us know. We will fix this issue in the final version.
>
> **Q2: Computational time**
> Since the algorithms are implemented with fixed time steps, the computational cost is then determined by the cost for each step. In our Algorithm 1, we use robust value iteration in each step, whose computational cost is $\mathcal{O}(S^2A\log (S))$ for both TV and $\chi^2$ sets (Iyengar, 2005). This cost is typical for any model-based robust RL method.
>
> **Q3: Algorithm 2 v.s. original Robust Natural Actor-Critic**
> We would like to clarify that our major contribution is the development of a reduction framework to reduce the challenging robust average reward problem to the easier discounted one, and the framework also enables methods for large scale optimization with function approximation.  We thus adopt the NAC algorithm in Algorithm 2 to illustrate that our framework can also be applied with function approximation techniques. Hence, the major part of the algorithm will be similar to RNAC, but the major contribution is the choice of the reduction factor in Line 3, which provides the performance guarantee under the average reward. As we showed numerically in E1 in the link, the choice of the reduction factor is necessary, otherwise RNAC with an randomly chosen factor can result in suboptimal performance. We also provide the first theoretical convergence guarantee for robust average reward with function approximation in Thm 5.1.
>
> As for the computational time, it will be identical to the NAC if the discount factors are the same. To illustrate that our framework will not result in high computation cost (with the reduction factor), we show the execution time for different factors in E4 in the link. Clearly, even with a large gamma, the execution time is similar, and hence our method is also computational efficient. More importantly, we provide solution to the average reward optimization, and RNAC is for discounted.

---

### Official Review · Reviewer_4tjq · 2025-03-24

**Overall Recommendation:** 3

**Summary:**

The paper introduces a reduction-based framework for solving robust average Markov Decision Processes (AMDPs) by transforming them into robust discounted MDPs (DMDPs). This framework allows existing methods for discounted reinforcement learning to be applied effectively to the average reward setting, ensuring data efficiency and scalability. The authors validate their approach through theoretical analysis and numerical experiments, demonstrating its robustness across uncertainty sets.

**Claims And Evidence:**

Yes.

**Essential References Not Discussed:**

N/A.

**Experimental Designs Or Analyses:**

Yes, the experiments are reasonable.

**Methods And Evaluation Criteria:**

Yes.

**Other Comments Or Suggestions:**

N/A.

**Other Strengths And Weaknesses:**

N/A.

**Questions For Authors:**

1. Are there any existing empirical methods for robust AMDPs? If so, the authors should compare their performance with the proposed methods in the experiments.
2. Algorithm 2 is indeed a DMDP algorithm without the discount factor selection part?
3. What are the used $\phi$s in the experiments when you implement Algorithm 2?

**Relation To Broader Scientific Literature:**

This paper is theoretically sounded and empirical meaningful.
1. Theoretically, the authors propose a provable framework to convert robust AMDPs to DMDPs by selecting an appropriate discount factor. The proof of Theorem 4.4 also has technical novelty to achieve tightness in the sample complexity under TV/CS constrained set.
2. Empirically, the authors validate the theoretical findings by observing the average reward attained from different discount factors.

Limitation:
1. The estimation of $\mathcal{H}$ seems hard when the environment is complicated, and thus empirically the selection of the discount factor might need hyperparameter fine-tuning.
2. Algorithm 2 only deals with linear approximation and needs to know the feature vectors in advance.

**Theoretical Claims:**

Yes, the theorems and the proofs are correct.

---

> ### Author Rebuttal · Authors · 2025-04-01
>
> We sincerely thank the reviewer for your time and thoughtful feedback, and we are glad to hear that our work is appreciated.
>
> We conducted additional experiments, at the link: https://anonymous.4open.science/r/ICML-2662-4C1E/README.md
>
> **W1: The estimation of $\mathcal{H}$.**
> Since we mainly focus on developing the theoretical foundation of our reduction framework, we assume the knowledge of $\mathcal{H}$, which is extensively adopted in previous theoretical studies, (see line 167, and additionally in [1,2]). Also we discussed several estimation methods in Appendix when problems have additional structures.
>
> On the other hand, while we acknowledge that in practice estimation of $\mathcal{H}$ can be challenging, we emphasize that our framework and subsequent results are still effective if $\mathcal{H}$ is replaced by its upper bound (see Remark 3.3). For example, the worst-case diameter is an upper bound and is easier to estimate [Wang et al., 2022]. We conduct additional experiments to verify our framework's effectiveness. We estimate the worst-case diameter for the Walker ($\approx 511$) and the the Hopper environment ($\approx 1339$), and plot the robust average reward of the policies learned from corresponding discount factors. We also specify the reduction factor and the optimal value. As the results in E1 in the link show, with our reduction factor, the reduction framework obtains a policy with the suboptimality gap within the error range, which verifies our theoretical results and illustrate the effectiveness of our framework with alternative upper bounds of $\mathcal{H}$.
>
> [1] Zhang, Z. et al. Sharper model-free reinforcement learning for average-reward markov decision processes. 2023.
> [2] Wang, S et al. Optimal Sample Complexity for Average Reward Markov Decision Processes. 2023.
>
> **W2: Algorithm 2 only deals with linear approximation and needs to know the feature vectors in advance.**
> Firstly, we want to clarify that the feature vectors in linear function approximation are pre-set by the learner before learning, instead of some unknown parameters. Hence, these features are known by the learner and can be set through different ways, e.g., tile coding [1], Fourier basis [2], or randomly generated [3].
>
> We also want to highlight that, our reduction framework is independent of any specific algorithm used, thus it is not restricted to the linear approximation. After setting the reduction discount factor, any discounted algorithm can be applied to optimize the average reward. We present Algorithm 2 with linear function mainly to develop a theoretical guarantee, but it can combined with any algorithm. To verify this, we provide an additional experiment with neural network approximation in E2 in the link. As the results showed, our reduction framework remains valid even with neural network approximation, and is more robust than the non-robust reduction method.
>
> [1]Sutton, Richard  Generalization in reinforcement learning: Successful examples using sparse coarse coding. 1995.
> [2] Konidaris, G. et al. Value function approximation in reinforcement learning using the Fourier basis. 2011.
> [3] Ghavamzadeh, M. et al.  LSTD with random projections. 2010.
>
> **Q1: Are there any existing empirical methods for robust AMDPs? If so, the authors should compare their performance with the proposed methods in the experiments.**
> We first acknowledge that there exist other methods for robust AMDPs (Wang et al., 2023c, Wang et al., 2023d). However, these methods only have asymptotic convergence guarantees, and can only be applied for tabular settings, whereas our method has finite sample complexity analysis and can be applied with function approximation.
>
> Since there is no straightforward implementations of these methods for large-scale problems, we compare our method to the two baselines, robust Value iteration (RVI) (Wang et al., 2023c) and robust relative value iteration (RRVI) (Wang et al., 2023d), under the tabular Garnet problem. We set the reduction factor to be $0.99$ in our framework (corresponds to $\epsilon=0.001$).   Results in E3 in the link show that our method finds a better policy with the same number of steps. Thus, our method achieves state-of-the-art performance in robust average reward optimization.
>
> **Q2: Algorithm 2 is indeed a DMDP algorithm without the discount factor selection part?**
> We clarify that we select the discount factor in Line 3. Based on our framework, robust average reward can be optimized through any algorithm for discounted MDP with the reduction factor. Thus, the remaining steps in algorithm 2 after selecting the factor is for the discounted reward. We also want to highlight that the developing of the reduction framework (instead of concrete algorithm design) is our major contribution.
>
> **Q3: What are the used $\phi$s in the experiments when you implement algorithm 2?**
> For each state-action pair, we generate a random vector and then normalize it to $(0,1)$ as the feature vector.

---

> > ### Comment · Reviewer_4tjq · 2025-04-03
> >
> > I appreciate the authors' response and will maintain my score.

---

### Decision · Program_Chairs · 2025-05-01

**Decision:**

Accept (poster)

**Comment:**

The paper studies robust RL with average reward by building a reduction to robust discounted MDP. With the reduction framework, any robust discounted MDP algorithm can be used to solve the original problem in the average reward setting.

While the reviewers in general are positive on the paper, there are still a few concerns about the paper. First the proposed approach needs to estimate the worst case diameter. The authors mentioned that they can estimate the diameter in practice but more details should be provided in terms of how and when one can ensure the estimated diameter is accurate. Another conern is on the computation side. Beyond very special settings such as tabular, it is in general computationally intractable to solve the robust MDP problem. Authors probably should fully discuss these points in their final version of the work.